# Chronos: Learning the Language of Time Series

**Abdul Fatir Ansari**[1][*], **Lorenzo Stella**[1][*], **Caner Turkmen**[1], **Xiyuan Zhang**[3][†], **Pedro Mercado**[1], **Huibin Shen**[1], **Oleksandr Shchur**[1], **Syama Sundar Rangapuram**[1], **Sebastian Pineda Arango**[4][†], **Shubham Kapoor**[1], **Jasper Zschiegner**[†], **Danielle C. Maddix**[1], **Hao Wang**[1,5][†], **Michael W. Mahoney**[2,6][†], **Kari Torkkola**[2], **Andrew Gordon Wilson**[2,7][†], **Michael Bohlke-Schneider**[1], **Yuyang Wang**[1]

*{ansarnd, stellalo}@amazon.com*

[1]*AWS AI Labs*, [2]*Amazon Supply Chain Optimization Technologies*, [3]*UC San Diego*, [4]*University of Freiburg*, [5]*Rutgers University*, [6]*UC Berkeley*, [7]*New York University*

Reviewed on OpenReview: https://openreview.net/forum?id=gerNCVqqtR
Code and Pretrained Models: https://github.com/amazon-science/chronos-forecasting

## Abstract

We introduce CHRONOS, a simple yet effective framework for pretrained probabilistic time series models. CHRONOS tokenizes time series values using scaling and quantization into a fixed vocabulary and trains existing transformer-based language model architectures on these tokenized time series via the cross-entropy loss. We pretrained CHRONOS models based on the T5 family (ranging from 20M to 710M parameters) on a large collection of publicly available datasets, complemented by a synthetic dataset that we generated via Gaussian processes to improve generalization. In a comprehensive benchmark consisting of 42 datasets, and comprising both classical local models and deep learning methods, we show that CHRONOS models: (a) significantly outperform other methods on datasets that were part of the training corpus; and (b) have comparable and occasionally superior *zero-shot* performance on new datasets, relative to methods that were *trained specifically on them*. Our results demonstrate that CHRONOS models can leverage time series data from diverse domains to improve zero-shot accuracy on unseen forecasting tasks, positioning pretrained models as a viable tool to greatly simplify forecasting pipelines.

## 1 Introduction

Time series forecasting is an essential component of decision-making across various domains, including retail, energy, finance, healthcare, climate science, among others. Traditionally, forecasting has been dominated by statistical models such as ARIMA and ETS. These have served as reliable tools, at least until the recent shift towards deep learning techniques (Hyndman & Athanasopoulos, 2018; Benidis et al., 2022). This shift can be attributed to the availability of large and diverse time series data sources, and the emergence of *operational forecasting* problems (Kolassa & Januschowski, 2019) that play to the strengths of deep forecasters, i.e., the ability to extract patterns out of a large collection of time series. Despite their impressive performance, deep forecasters still operate in the standard regime of training and prediction on the same dataset. While there have been works dedicated to transfer learning (Ye & Dai, 2018) and domain adaptation (Jin et al., 2022) for forecasting, the field has yet to converge on a unified, general-purpose forecasting model, a goal that remains a beacon for time series researchers.

The emergence of large language models (LLMs) with zero-shot learning capabilities has ignited interest in developing "foundation models" for time series. In the context of LLMs, this interest has been pursued through two main avenues: directly prompting pretrained LLMs in natural language (Gruver et al., 2023;

---

[*]Equal contribution.
[†]Xiyuan Zhang and Sebastian Pineda Arango contributed to this work during their internships at AWS. Hao Wang, Michael W. Mahoney, and Andrew Gordon Wilson hold concurrent appointments at Amazon and their corresponding universities, and this paper describes work performed at Amazon.

Xue & Salim, 2023) and fine-tuning LLMs for time series tasks (Zhou et al., 2023a; Jin et al., 2024). However, these methods face significant limitations, notably the need for prompt engineering or fine-tuning for each new task, or reliance on large-scale models (GPT-3 (Brown et al., 2020), Llama 2 (Touvron et al., 2023), etc.) that demand substantial computational resources and time for inference. Recent concurrent work (Dooley et al., 2023; Das et al., 2023; Rasul et al., 2023; Woo et al., 2024) also explores pretraining transformer-based models with sophisticated time-series-specific designs on a large corpus of real and (or) synthetic time series data.

In this work, we take a step back and ask: what are the fundamental differences between a language model that predicts the next token, and a time series forecasting model that predicts the next values? Despite the apparent distinction — tokens from a finite dictionary versus values from an unbounded, usually continuous domain — both endeavors fundamentally aim to model the sequential structure of the data to predict future patterns. Shouldn't good language models "just work" on time series? This naive question prompts us to challenge the necessity of time-series-specific modifications, and answering it led us to develop CHRONOS, a language modeling framework minimally adapted for time series forecasting. CHRONOS tokenizes time series into discrete bins through simple scaling and quantization of real values. In this way, we can train off-the-shelf language models on this "language of time series," with no changes to the model architecture (see Figure 1 for a high-level depiction of CHRONOS). Remarkably, this straightforward approach proves to be effective and efficient, underscoring the potential for language model architectures to address a broad range of time series problems with minimal modifications.

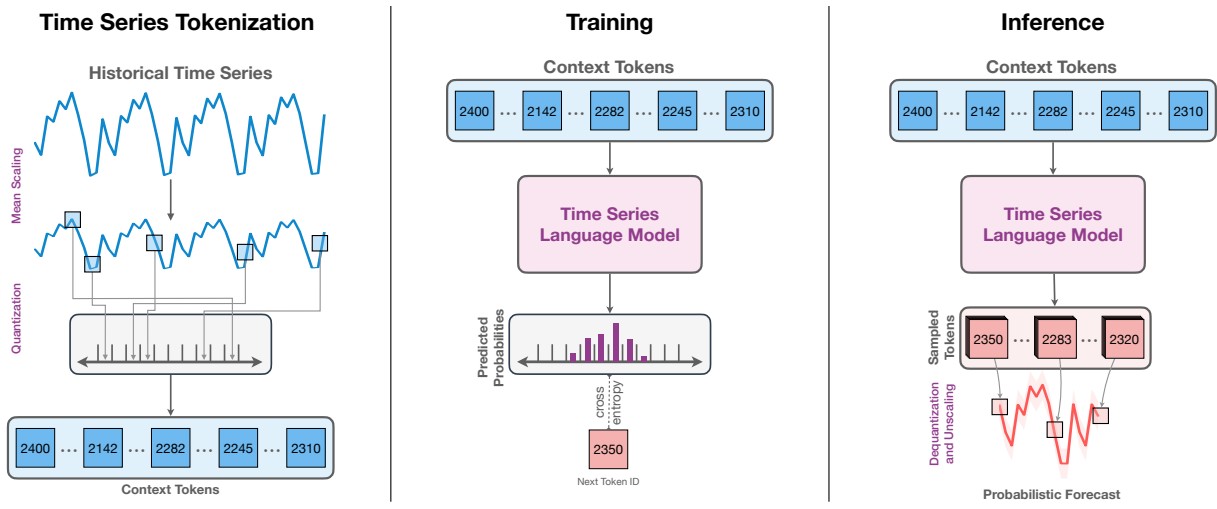

Figure 1: High-level depiction of CHRONOS. (**Left**) The input time series is scaled and quantized to obtain a sequence of tokens. (**Center**) The tokens are fed into a language model which may either be an encoder-decoder or a decoder-only model. The model is trained using the cross-entropy loss. (**Right**) During inference, we autoregressively sample tokens from the model and map them back to numerical values. Multiple trajectories are sampled to obtain a predictive distribution.

For the development of a useful general-purpose time series forecasting model, the scarcity of publicly available time series datasets, both in quantity and quality, is arguably more critical than the modeling framework. In addition to the comprehensive collection of public datasets we used to train CHRONOS, a central aspect of our approach is the integration of data augmentation strategies, including TSMixup and KernelSynth. TSMixup randomly samples a set of base time series from different training datasets, and generates new time series based on a convex combination of them; KernelSynth uses Gaussian processes to generate synthetic time series by randomly composing kernel functions. These techniques address the inherent limitations of small training datasets in time series forecasting, enhancing model robustness and generalization.

Our comprehensive evaluation across 42 datasets establishes CHRONOS as a benchmark for both in-domain and zero-shot forecasting, surpassing both traditional models and task-specific deep learning approaches.

Notably, Chronos achieves impressive zero-shot forecasting performance out of the box, without necessitating task-specific adjustments. Its accuracy, coupled with its relatively modest model size, positions it as a preferable alternative to larger, more computationally demanding models for zero-shot forecasting applications. By its very nature as a language model operating over a fixed vocabulary, Chronos can seamlessly integrate with future advancements in LLMs, making it an ideal candidate for further development as a generalist time series model.

The rest of the paper is organized as follows. Section 2 introduces the background on time series forecasting and language models, and discusses related work. In Section 3, we describe Chronos, our proposed language modeling framework for time series. Section 4 discusses our data augmentation technique and synthetic time series generation process. In Section 5, we present our main results and a rigorous analysis of different design choices. We discuss future directions in Section 6, and conclude the paper in Section 7. Additional material is presented in the appendices.

## 2 Background and Related Work

**Time series forecasting** concerns using historical data from a quantity of interest (typically real-valued) to predict their future values. Formally, given a uniformly-spaced time series $\boldsymbol{x}_{1:C} = [x_1, \ldots, x_C]$, we are interested in predicting the joint distribution of the next $H$ steps, $p(\boldsymbol{x}_{C+1:C+H}|\boldsymbol{x}_{1:C})$. In this work, we focus on *univariate* forecasting, where the observations are scalars, i.e., $x_i \in \mathbb{R}$ for all $i$.

Time series forecasting can be addressed with a variety of different methods which can be broadly categorized into classical forecasting methods and deep learning methods. Classical forecasting methods such as ETS, ARIMA (Hyndman et al., 2008), Theta (Assimakopoulos & Nikolopoulos, 2000) fit a separate model to each time series independently (hence referred to as *local* models). In contrast, deep learning forecasting models learn across time series in a given dataset (and are called *global* models). These methods leverage advances in deep learning, such as RNNs which are used by DeepState (Rangapuram et al., 2018), DeepFactor (Wang et al., 2019), DeepAR (Salinas et al., 2020), TimeGrad (Rasul et al., 2021), and transformers which are used by TFT (Lim et al., 2021) and PatchTST (Nie et al., 2023). Apart from the choice of architecture, these approaches differ in the way they model the target, with some modeling the density function while others directly predicting a set of quantiles (Wen et al., 2017; Gasthaus et al., 2019; Park et al., 2022). Nevertheless, not all models produce probabilistic forecasts: notably, models such as Informer (Zhou et al., 2021) and DLinear (Zeng et al., 2023) only produce point forecasts.

**Large language models (LLMs)** have demonstrated impressive performance on various natural language processing tasks (Brown et al., 2020; Chung et al., 2022; Touvron et al., 2023). Given a sequence of input tokens, $\boldsymbol{w}_{1:k} = [w_1, \ldots, w_k]$, language models aim to predict the next token, $w_{k+1}$, by modeling the conditional distribution, $p(w_{k+1}|\boldsymbol{w}_{1:k})$. The tokens belong to a vocabulary, $\mathcal{V}$, and may be characters, subwords (Sennrich et al., 2015), or words, depending on the tokenization scheme used.

Most modern LLMs (Brown et al., 2020; Chung et al., 2022; Touvron et al., 2023) are based on the transformer architecture (Vaswani et al., 2017). The original transformer architecture is an encoder-decoder model designed for machine translation. The encoder maps an input sentence of some language to a continuous representation, and the decoder generates the translation token-by-token using the input representation and previously decoded tokens. Many popular language models, such as BART (Lewis et al., 2019) and T5 (Raffel et al., 2020; Chung et al., 2022), belong to this family. Another popular architecture for LLMs is decoder-only, used in GPT-3 (Brown et al., 2020) and Llama 2 (Touvron et al., 2023), where the model only attends to tokens up to the current token. LLMs are typically trained on a very large corpus of text with their number of parameters ranging from millions (Raffel et al., 2020) to hundreds of billions (Chowdhery et al., 2023). We refer the reader to Zhao et al. (2023) for a recent survey on this area of research.

**LLM-based forecasters.** Inspired by the success of pretrained LLMs, recent work has shown that LLMs are general pattern recognizers (Mirchandani et al., 2023) and several methods adapting LLMs to the time series domain have been developed. One line of work treats numerical time series data as raw text and directly uses the pretrained LLMs with minimal or no fine tuning to forecast unseen time series. PromptCast (Xue & Salim, 2023) leverages pretrained LLMs for forecasting by transforming the time series data into text-based

input and output pairs and reformulating the forecasting problem as a question answering task. However, PromptCast requires dataset-specific templates for converting numerical data to text prompts. Perhaps the most straightforward LLM-based forecasting model is LLMTime (Gruver et al., 2023), which shows clear evidence for zero-shot forecasting ability of pretrained LLMs on a variety of benchmark time series datasets. LLMTime proposes a new tokenization scheme that encodes real-valued data as a string of digits after fixing the numerical precision and scaling the data appropriately. Once encoded as strings, forecasts are obtained in a zero-shot setting from pretrained LLMs such as GPT-3 (Brown et al., 2020) and Llama 2 (Touvron et al., 2023). Nevertheless, the use of such compute-hungry models hampers the scalability and practical utility of LLMTime.

Zhou et al. (2023a) propose a unified one-fits-all model (GPT4TS) for different time series analysis tasks by using a pretrained GPT-2 model (Radford et al., 2019) as a backbone and only fine-tune the positional embeddings and the parameters of the layer normalization for each individual task. Instead of using tokenized input, they directly feed the model with patch embeddings, similar to PatchTST (Nie et al., 2023). Recent concurrent work, Time-LLM (Jin et al., 2024), repurposes LLMs for time series forecasting by aligning embeddings of time series patches with text prototypes, and prompting the (frozen) LLM with these aligned embeddings and a natural language prefix describing the task. Unlike CHRONOS, both GPT4TS and Time-LLM require in-domain training or fine-tuning, i.e., they are fine-tuned and tested on each dataset separately. Furthermore, the aforementioned methods are based on prompting or fine-tuning *pretrained* LLMs. In contrast, CHRONOS trains language models *from scratch* on a large collection of time series, tokenized via scaling and quantization.

**Zero-shot forecasting.** Zero-shot forecasting is the ability of models to generate forecasts for time series from unseen datasets. Some early work (Orozco & Roberts, 2020; Oreshkin et al., 2021; Jin et al., 2022) in zero-shot forecasting considers training on a single time series dataset and testing on a different dataset. ForecastPFN (Dooley et al., 2023) tackles the problem of zero-shot forecasting by training a transformer-based model purely on synthetic data generated according to predefined trend, seasonalities (daily, monthly, yearly). The trained transformer model is then used to forecast real-world time series in a zero-shot setting. In this work, we also propose a method to generate synthetic time series data from Gaussian processes (Section 4.2); however, we use the synthetic data in combination with real data to train CHRONOS models, which improves the overall zero-shot performance. Furthermore, CHRONOS models are probabilistic, whereas ForecastPFN can only generate point forecasts.

Recent concurrent works (Rasul et al., 2023; Goswami et al., 2024; Das et al., 2023; Woo et al., 2024) also develop zero-shot forecasting models by pretraining transformer-based architectures on a large corpus of time series data. These works operate on the real values of the time series and include time-series-specific designs such as time features, lags, patching, and real-valued distribution heads, among others. In contrast, CHRONOS follows a minimalist approach by tokenizing time series values into a fixed vocabulary and training existing language model architectures on these tokens without any time-series-specific design or features. That is, CHRONOS uses a categorical distribution to model the observations, performing regression via classification.

**Other time series tasks.** Similar to Zhou et al. (2023a), recent works have studied general purpose models applicable across time series tasks including imputation, forecasting, classification and anomaly detection. Wu et al. (2023) develop a task-generic backbone based on the Inception model (Szegedy et al., 2015). In order to use the CNN-based Inception model, one dimensional time series is transformed into a two dimensional image-like representation by essentially segmenting the time series based on the periodicity and stacking the segments. SimMTM (Dong et al., 2023) is a masked pretraining framework for time series which learns general time series representations that are then used for forecasting and classification via fine-tuning. Although we focus on univariate time series forecasting in this work, based on its excellent performance on unseen time series datasets, we hypothesize that CHRONOS learns general representations that can potentially be deployed for tasks beyond forecasting.

# 3 Chronos: A Language Modeling Framework for Time Series

In this section we introduce CHRONOS, a framework adapting existing language model architectures and training procedures to probabilistic time series forecasting. While both language and time series are sequential in nature, they differ in terms of their representation — natural language consists of words from a finite vocabulary, while time series are real-valued. This distinction necessitates specific modifications to existing language modeling frameworks, especially concerning tokenization, to make them applicable to time series data. Nevertheless, since existing transformer models have excelled on language tasks, our design philosophy involves making minimal changes to the model architectures and training procedure.

## 3.1 Time Series Tokenization

Consider a time series $\boldsymbol{x}_{1:C+H} = [x_1, \ldots, x_{C+H}]$, where the first $C$ time steps constitute the historical context, and the remaining $H$ represent the forecast horizon. Language models operate on tokens from a finite vocabulary, so using them for time series data requires mapping the observations $x_i \in \mathbb{R}$ to a finite set of tokens. To this end, we first scale and then quantize observations into a fixed number of bins.

**Scaling.**   The scale of time series can differ significantly even within a single dataset. This poses optimization challenges for deep learning models. Therefore, individual time series are normalized to facilitate better optimization. In the case of CHRONOS, the goal of normalization is to map the time series values into a suitable range for quantization. A common normalization technique involves applying an affine transformation to the time series, i.e., $\tilde{x}_i = (x_i - m)/s$. Several popular normalization schemes, such as mean scaling, standard scaling and min-max scaling, can be obtained by appropriately choosing $m$ and $s$. We opt for mean scaling, a method that has proven effective in deep learning models commonly used for practical time series applications (Salinas et al., 2020; Rabanser et al., 2020), but other approaches are viable and only require minimal changes. An attractive feature of mean scaling is that it preserves zero values in the time series, which are often semantically meaningful, such as zero sales for a product or zero solar energy generation at night. Mean scaling normalizes individual entries of the time series by the mean of the absolute values in the historical context. Specifically, this involves setting $m = 0$ and $s = \frac{1}{C} \sum_{i=1}^{C} |x_i|$.

**Quantization.**   The scaled time series $\tilde{\boldsymbol{x}}_{1:C+H} = [\tilde{x}_1, \ldots, \tilde{x}_C, \ldots, \tilde{x}_{C+H}]$, is still real-valued and cannot be processed directly by language models. To convert these real values into discrete tokens, we employ quantization. Formally, we select $B$ bin centers $c_1 < \ldots < c_B$ on the real line, and $B-1$ edges $b_i$ separating them, $c_i < b_i < c_{i+1}$, for $i \in \{1, \ldots, B-1\}$. The quantization function $q : \mathbb{R} \to \{1, 2, \ldots, B\}$, and dequantization $d : \{1, 2, \ldots, B\} \to \mathbb{R}$, are then defined as

$$q(x) = \begin{cases} 1 & \text{if } -\infty \le x < b_1, \\ 2 & \text{if } b_1 \le x < b_2, \\ \vdots & \\ B & \text{if } b_{B-1} \le x < \infty, \end{cases} \quad \text{and} \quad d(j) = c_j, \tag{1}$$

respectively. The positioning of bin centers and edges can either be data-dependent or uniform (Rabanser et al., 2020). Quantile binning, a type of data-dependent binning, exploits the cumulative distribution function (CDF) of the training datapoints to construct bins such that approximately equal number of datapoints are assigned to each bin. In contrast, uniform binning selects uniformly-spaced bin centers within the interval $[c_1, c_B]$ and the bin edges fall mid-way between the successive bin centers, i.e., $b_i = \frac{c_i + c_{i+1}}{2}$ for $i \in \{1, \ldots, B-1\}$. Since the distribution of values for unseen downstream datasets can differ significantly from the training distribution, we opt for uniform binning in our experiments, but other quantization techniques can be used. We refer the reader to Rabanser et al. (2020) for a detailed discussion on quantization schemes for time series. A potential limitation of this approach is that the prediction range is restricted between $[c_1, c_B]$, making it *theoretically* infeasible to model time series with a strong trend. We explore this further in a practical setting in Section 5.7.

Apart from the time series tokens $\{1, 2, \ldots, B\}$, we include two special tokens, commonly used in language models, into the time series vocabulary, $\mathcal{V}_{\text{ts}}$: PAD and EOS. The PAD token is used to pad time series of different

lengths to a fixed length for batch construction and to replace missing values. The EOS token is appended to the quantized and padded time series to denote the end of the sequence. While the use of an EOS token is not strictly necessary in the case of time series, it makes training and inference using popular language modeling libraries convenient. The sequences of tokens from $\mathcal{V}_{ts}$ can readily be processed by language models (both encoder-decoder and decoder only models), to train them as usual. A common approach in time series modeling is to incorporate time and frequency information, through features such as day-of-week, week-of-year, and so on. Perhaps counter-intuitively, in CHRONOS, we ignore time and frequency information, treating the "time series" simply as a sequence.

We primarily focus on the variants of the encoder-decoder T5 model (Raffel et al., 2020). Additionally, we conduct an experiment with the GPT-2 (Radford et al., 2019) model to demonstrate that our approach can be straightforwardly extended to decoder-only models. No modifications are required to the language model architecture, except adjusting the vocabulary size to $|\mathcal{V}_{ts}|$, which depends on the number of bins used for quantization and may be different from the vocabulary size of the original language model. Concretely, adjusting the vocabulary size entails truncating (or extending) the input and output embedding layers of the language model.

### 3.2  Objective Function

As typical in language models, we use the categorical distribution over the elements of $\mathcal{V}_{ts}$ as the output distribution, $p(z_{C+h+1}|\boldsymbol{z}_{1:C+h})$ where $\boldsymbol{z}_{1:C+h}$ is the tokenized time series. CHRONOS is trained to minimize the cross entropy between the distribution of the quantized ground truth label and the predicted distribution. Formally, the loss function for a single tokenized time series (also accounting for EOS tokens) is given by,

$$\ell(\boldsymbol{\theta}) = -\sum_{h=1}^{H+1}\sum_{i=1}^{|\mathcal{V}_{ts}|}\mathbf{1}_{(z_{C+h+1}=i)}\log p_{\boldsymbol{\theta}}(z_{C+h+1}=i|\boldsymbol{z}_{1:C+h}), \tag{2}$$

where $p_{\boldsymbol{\theta}}(z_{C+h+1}=i|\boldsymbol{z}_{1:C+h})$ denotes the categorical distribution predicted by the model parameterized by $\boldsymbol{\theta}$. In practice, the loss is averaged over a batch of time series during training.

Note that the categorical cross entropy loss (Eq. 2) is not a distance-aware objective function, i.e., it does not explicitly recognize that bin $i$ is closer to bin $i+1$ than to $i+2$. Instead, the model is expected to associate nearby bins together, based on the distribution of bin indices in the training dataset. In other words, CHRONOS performs regression via classification (Torgo & Gama, 1997; Stewart et al., 2023). This is unlike typical probabilistic time series forecasting models, which either use parametric continuous distributions such as Gaussian and Student's-t (Salinas et al., 2020) or perform quantile regression (Wen et al., 2017; Lim et al., 2021).

Opting for a categorical output distribution offers two key advantages. Firstly, it requires no modification to the language model architecture or training objective, enabling the use of popular language modeling libraries and the utilities they provide out of the box (Wolf et al., 2020). Secondly, it imposes no restrictions on the structure of the output distribution, allowing the model to learn arbitrary distributions, including multimodal ones. This flexibility proves especially valuable for a pretrained model, as time series datasets from diverse domains may follow distinct output distribution patterns.

Arguably, modeling the output as an ordinal variable would be more appropriate, since the output domain is obtained by discretizing the real line. In fact, regression models for ordinal variables have been extensively studied in the literature (McCullagh, 1980; Winship & Mare, 1984), including for neural networks and transformer models (Cheng et al., 2008; Hu et al., 2021). Imposing the ordinal nature of the classes on top of the models, in similar ways to the mentioned literature, could be an interesting extension of this work.

### 3.3  Forecasting

CHRONOS models are probabilistic by design and multiple realizations of the future can be obtained by autoregressively sampling from the predicted distribution, $p_{\theta}(z_{C+h+1}|\boldsymbol{z}_{1:C+h})$, for $h \in \{1, 2, \dots, H\}$. These sample paths come in the form of token IDs that need to be mapped back to real values and then unscaled

to obtain the actual forecast. The dequantization function $d$ from Eq. (1) maps the predicted tokens to real values: these are then unscaled by applying the inverse scaling transformation, which in the case of mean scaling involves multiplying the values by the scale $s$.

## 4 Data Augmentation

The quality and quantity of public time series data pales in comparison to the natural language processing (NLP) domain, which benefits from ample high-quality text datasets such as WikiText-103 (Merity et al., 2016), C4 (Raffel et al., 2020), and The Pile (Gao et al., 2020). This poses challenges for training models intended for zero-shot forecasting, which rely on large-scale time series data with diverse patterns. To address this issue, we propose enhancing the diversity of training data by generating mixup augmentations from real datasets and supplementing training with synthetic data.

### 4.1 TSMixup: Time Series Mixup

Mixup (Zhang et al., 2017) is a data augmentation scheme proposed in the context of image classification. It generates convex combinations of random image pairs and their labels from the training dataset, which alleviates issues such as memorization and overfitting in deep learning models. Existing works (Carmona et al., 2021; Zhou et al., 2023b) have extended Mixup to the time series domain.

Building upon these works, we propose TSMixup, which generalizes the idea of Mixup to more than two datapoints. Concretely, TSMixup randomly samples $k \sim \mathcal{U}\{1, K\}$ time series of a specific length, $l \sim \mathcal{U}\{l_{\min}, l_{\max}\}$, from the training datasets, scales them, and takes their convex combination,

$$\tilde{\boldsymbol{x}}_{1:l}^{\text{TSMixup}} = \sum_{i=1}^{k} \lambda_i \tilde{\boldsymbol{x}}_{1:l}^{(i)}, \qquad (3)$$

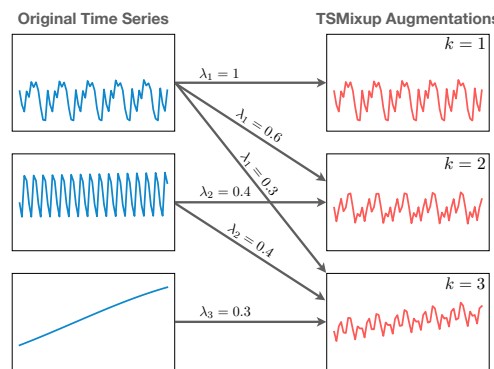

Figure 2: An illustration of TSMixup augmentation for $k = \{1, 2, 3\}$. TSMixup improves pattern diversity by taking weighted combinations of randomly-sampled time series from different datasets.

where $\tilde{\boldsymbol{x}}_{1:l}^{(i)}$ denotes the $i$-th scaled time series. The time series are scaled before mixing to ensure that time series with small and large values are given equal importance in the mixing process. The combination weights, $[\lambda_1, \ldots, \lambda_k]$, are sampled from a symmetric Dirichlet distribution, $\text{Dir}(\alpha)$, parameterized by the scalar concentration parameter $\alpha$. The complete pseudocode of TSMixup can be found in Algorithm 1 in Appendix A. Intuitively, TSMixup enhances the diversity of data by combining patterns from different time series. Figure 2 shows example augmentations generated by TSMixup and illustrates how different patterns are mixed.

### 4.2 KernelSynth: Synthetic Data Generation using Gaussian Processes

While TSMixup improves pattern diversity, it may still prove insufficient for training a generalist time series model, especially when real data is limited. To further supplement the training dataset, we propose KernelSynth, a method to generate synthetic time series using Gaussian processes (GPs). KernelSynth is inspired by the Automatic Statistician (Duvenaud et al., 2013), where a compositional search over a space of GP kernels is performed to explain the structure of a time series. We use the inverse of this process — randomly compose GP kernels to generate new time series.

GPs are distributions over functions defined by the mean function, $m(t)$, and the positive definite kernel, $\kappa(t, t')$, where $t \in \mathbb{R}$ is the domain. The kernel specifies a covariance function which defines the joint variability of the function values at an arbitrary pair of points, $(t, t')$, in the input domain. Diverse patterns can be generated by appropriately selecting the kernel. We constructed a kernel bank, $\mathcal{K}$, of basis kernels defining fundamental time series patterns. These include linear kernels for trend, RBF kernels for smooth local variation, and periodic kernels for seasonalities found in typical time series frequencies. The final kernel,

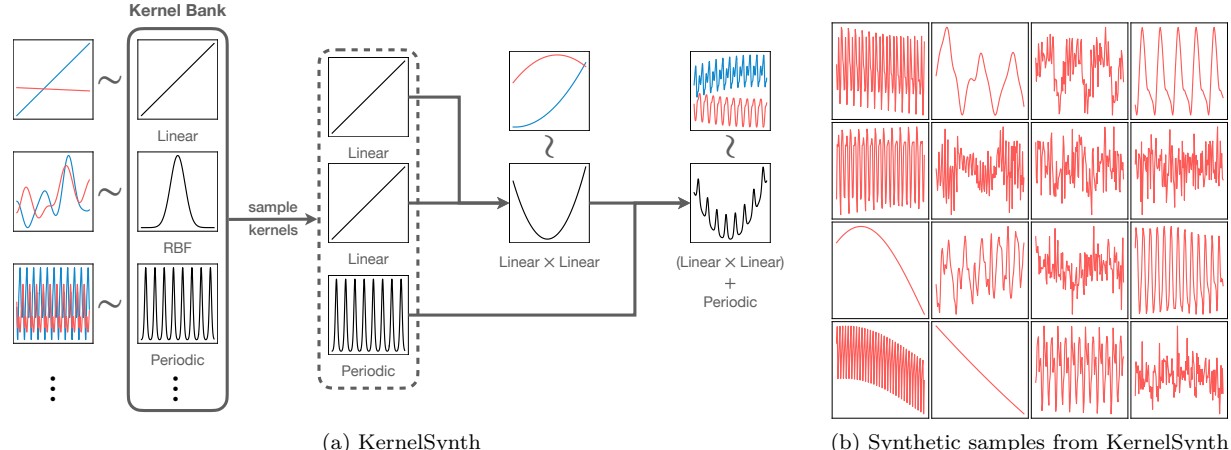

(a) KernelSynth

(b) Synthetic samples from KernelSynth

Figure 3: (a) An illustration of KernelSynth, a Gaussian process (GP)-based synthetic time series generation method. Kernels are sampled from a kernel bank and then randomly combined using a binary operator ($\times$ or $+$). The resultant kernel is used in a GP prior to generate synthetic time series. Random samples from kernels at each step are shown in red and blue colors. (b) Example synthetic time series generated by KernelSynth.

$\tilde{\kappa}(t, t')$, is constructed by sampling $j \sim \mathcal{U}\{1, J\}$ kernels from $\mathcal{K}$ with replacement and combining these kernels via random binary operations, $+$ or $\times$. A synthetic time series is generated by drawing a sample of length $l_{\text{syn}}$ from the GP prior, $\mathcal{GP}(m(t) = 0, \tilde{\kappa}(t, t'))$; see Algorithm 2 in Appendix A for details. Figure 3 depicts this generative process used in KernelSynth, illustrating how time series with intricate patterns can arise from the composition of simple basis kernels.

## 5 Experiments

In this section, we present empirical results on commonly used benchmark datasets. First, we give an overview of the datasets, training strategy, baselines, and evaluation metrics (Section 5.1-5.4). Table 1 provides a high-level summary of the datasets and baselines used in our experiments. We then (a) evaluate the performance of CHRONOS models in the in-domain and zero-shot settings against local models and task-specific deep learning models (Section 5.5); (b) analyze the effect of various design choices such as model size, initialization, synthetic data proportion, context length, and vocabulary size on the performance of CHRONOS models (Section 5.6); and (c) analyze the qualitative performance of CHRONOS models and highlight their limitations (Section 5.7). We discuss our key findings in this section and relegate specific experiment details to the appendices.

Table 1: A high-level summary of the datasets and baselines used in our experiments.

| Data Subset | # Datasets | # Series | Usage | Baselines |
|---|---|---|---|---|
| Pretraining-only | 13 | 795,936 | pretraining | – |
| Benchmark I | 15 | 97,272 | pretraining and in-domain evaluation | Naive, SeasonalNaive, AutoETS, Auto-Theta, SCUM, AutoARIMA, DeepAR, TFT, PatchTST, DLinear, WaveNet, N-BEATS, N-HiTS, GPT4TS, Lag-Llama, Moirai-1.0-R |
| Benchmark II | 27 | 190,674 | zero-shot evaluation | All the above, LLMTime and ForecastPFN |

### 5.1 Datasets

To train and evaluate CHRONOS models, we collected a wide variety of publicly available datasets spanning various application domains including energy, transport, healthcare, retail, web, weather, finance, and with sampling frequencies ranging from 5 minutes up to yearly. The complete list of datasets, together with their

respective sources and additional details, is given in Appendix B. In total, our dataset collection comprises 55 datasets from multiple sources, including the Monash Time Series Forecasting Repository (Godahewa et al., 2021), the M-competitions (Makridakis et al., 1979; Makridakis & Hibon, 2000; Makridakis et al., 2020; 2022), and public domain datasets from Kaggle.[1]

We categorize this collection into three subsets, based on how we use them for training and evaluating CHRONOS models: (a) datasets exclusively used for training (13 datasets); (b) Benchmark I datasets, employed for both training and evaluation, representing an *in-domain* evaluation (15 datasets); and (c) Benchmark II datasets, used solely for evaluation, constituting a *zero-shot* evaluation (27 datasets). In categorizing the datasets in this way, we tried to find a good balance between keeping as many datasets as possible for the zero-shot evaluation of CHRONOS models, among the ones most commonly used in the literature, while still having enough variety of domains and sampling frequencies in the training data. Overall, we used 28 datasets for training CHRONOS models, consisting of about 890K univariate time series with approximately 84B observations (tokens) in total. For both in-domain (I) and zero-shot (II) benchmark datasets, we used the last $H \in \mathbb{N}^+$ observations of each time series as a held-out test set: all models are judged by the accuracy of their forecast on such held-out set, which no model had access to for training purposes. The prediction length $H$ is task-specific (see Table 3 in Appendix B), where we define a task as a dataset and prediction length pair. Tasks in both benchmarks exhibit diverse properties, in terms of the dataset size, frequency, history length, and prediction length, making them rich benchmarks reflective of real world scenarios.

## 5.2 Training Corpus and Protocols

We selected T5 (Raffel et al., 2020) as the main architecture for CHRONOS in our experiments, since it is available in a variety of sizes, ranging from 16M (Tiny) to 11B (XXL) parameters (Tay et al., 2021). We also conducted experiments with the decoder-only GPT-2 model to demonstrate the applicability of the CHRONOS framework to decoder-only models. In the following, we discuss the training configurations used for our main results (Section 5.5) and explore alternatives for some of the hyperparameters in Section 5.6.

We trained T5 models of 4 sizes,[2] namely, Mini (20M), Small (46M), Base (200M) and Large (710M), and the GPT-2 base model (90M), on 10M TSMixup augmentations (see Section 4.1) generated from the 28 training datasets, with $K = 3$ in Algorithm 1, and 1M synthetic time series generated using Gaussian processes (see Section 4.2). Note that with this setup, original time series are adequately represented since they are included in the TSMixup augmentations with probability 1/3. We sampled time series from the augmentations and synthetic data in the ratio 9:1 during training. Each model is trained with an effective batch size of 256 sequences, using distributed data parallelism and gradient accumulation, whenever necessary. These sequences were constructed by slicing random windows from the time series, and then scaling and quantizing them into equal-sized bins within the interval $[c_1 = -15, c_B = +15]$, as described in Section 3.1. We set the vocabulary size, $\mathcal{V}_{\text{ts}}$, to 4096, including the special tokens (PAD and EOS). The context length of the sequences was set to 512, the default for T5 models, and the prediction length was set to 64, a value greater than the prediction lengths of all tasks we consider in our evaluation.

The models were optimized for 200K steps using the AdamW optimizer with a weight decay of 0.01. The learning rate was annealed linearly from its initial value of 0.001 to 0 over the training steps. The other model and training hyperparameters were set to their defaults used in the transformers library (Wolf et al., 2020). We used an AWS EC2 instance with 8 A100 (40GB) GPUs to train all CHRONOS models, and we employed faster floating point formats (TF32) and model compilation to speed up training. Table 6 in Appendix E reports the training time and the approximate cost of training CHRONOS models of different sizes.

## 5.3 Baselines

We assessed the performance of CHRONOS models against a variety of time series forecasting baselines. From statistical forecasting literature (Hyndman & Athanasopoulos, 2018), we included Naive, Seasonal Naive, AutoETS, AutoARIMA (Hyndman et al., 2008), AutoTheta (Assimakopoulos & Nikolopoulos, 2000) and a

---

[1]The datasets used in our experiments are available at https://huggingface.co/datasets/autogluon/chronos_datasets.
[2]Our code and model checkpoints are available at https://github.com/amazon-science/chronos-forecasting.

strong ensemble (SCUM) of statistical models (Petropoulos & Svetunkov, 2020). Additionally, we compared against several neural forecasting baselines, including WaveNet (Oord et al., 2016), DeepAR (Salinas et al., 2020), N-BEATS (Oreshkin et al., 2020), TFT (Lim et al., 2021), DLinear (Zeng et al., 2023), PatchTST (Nie et al., 2023), N-HiTS (Challu et al., 2023), and GPT4TS (Zhou et al., 2023a). Furthermore, from the recently proposed pretrained time series models, we included the ones with publicly available weights: Lag-Llama (Rasul et al., 2023) and Moirai-1.0-R (Woo et al., 2024). On Benchmark II (i.e., zero-shot datasets for CHRONOS models), we also evaluated against two zero-shot methods: ForecastPFN (Dooley et al., 2023) which is a transformer model pretrained only on synthetic time series data and LLMTime (Gruver et al., 2023) which uses LLMs for zero-shot forecasting.

We categorize CHRONOS models and the baselines into three groups: *local models* that estimate parameters for each time series individually; *task-specific models* trained or fine-tuned for each task separately; and *pretrained models* which do not perform task-specific training, instead using a single model across all tasks. Further details on the implementation and training of these baselines can be found in Appendix C.

## 5.4 Evaluation Metrics

Whenever possible,[3] we evaluated models both in terms of their probabilistic and point forecast performance. We used the weighted quantile loss (WQL) to assess the quality of the probabilistic forecasts: the WQL is related to the continuous ranked probability score (CRPS, Gneiting & Raftery (2007))[4] and is commonly used to evaluate probabilistic forecasts (Gasthaus et al., 2019; Shchur et al., 2023). The WQL measures the compatibility between the predictive distribution and the ground-truth observation at a uniformly-spaced grid of quantile levels; we compute the WQL on 9 uniformly-spaced quantile levels $\{0.1, 0.2, \ldots, 0.9\}$. Quantile forecasters such as TFT were directly trained on these quantile levels. For methods requiring sampling, we estimated the quantiles using 20 sample forecast paths. We used the mean absolute scaled error (MASE, Hyndman & Koehler (2006)) to evaluate the point forecast performance. The MASE is defined as the absolute error of the forecast scaled by the historical seasonal error of the time series, and was selected due to its favorable properties over other point forecasting metrics (Hyndman & Koehler, 2006). We used the median forecast (0.5-quantile) for computing the MASE for the probabilistic forecasters. See Appendix D for a detailed discussion on the evaluation metrics.

Since the magnitude of the evaluation metrics can vary across datasets, we adopt a different approach to aggregate scores than naive averaging. For each dataset, we compute the relative score of each model as the model's score divided by the score of a baseline model (here, Seasonal Naive). The relative scores are aggregated across all datasets using the geometric mean. The choice of the geometric mean is deliberate — Fleming & Wallace (1986) show that the arithmetic mean can yield misleading conclusions in this context, and the geometric mean is provably the only meaningful way to aggregate such relative scores. Furthermore, the geometric mean is also not sensitive to the choice of the baseline, and the model ordering stays intact if another baseline is selected instead. We used Seasonal Naive due to its simplicity and popularity as a forecasting baseline. For models that failed or could not finish evaluation within the allotted time on certain datasets, we used a relative score of 1, i.e., the baseline relative score, when aggregating the results. We assign equal weights to all tasks during aggregation, reflecting real-world scenarios where datasets may have different numbers of time series, frequencies, history and prediction lengths.

## 5.5 Main Results

In this section, we present our main results on 42 datasets, which comprise Benchmark I (15 datasets) and Benchmark II (27 datasets). CHRONOS models surpass classical statistical baselines, task-specific deep learning models, and other pretrained models on the in-domain datasets (Benchmark I; see Section 5.5.1). On the zero-shot datasets (Benchmark II; Section 5.5.2), CHRONOS models comfortably outperform statistical baselines and other pretrained models, while performing on par with the best deep learning models trained on these tasks. With an inexpensive fine-tuning regimen, our CHRONOS-T5 (Small) model achieves the top spot on Benchmark II, significantly outperforming all baselines.

---

[3]Some models (GPT4TS and ForecastPFN) only generate point forecasts and we only evaluate those.
[4]Many existing works (Ansari et al., 2021; Rasul et al., 2023; Kollovieh et al., 2023) use CRPS and WQL synonymously.

### 5.5.1 Benchmark I: In-domain Results

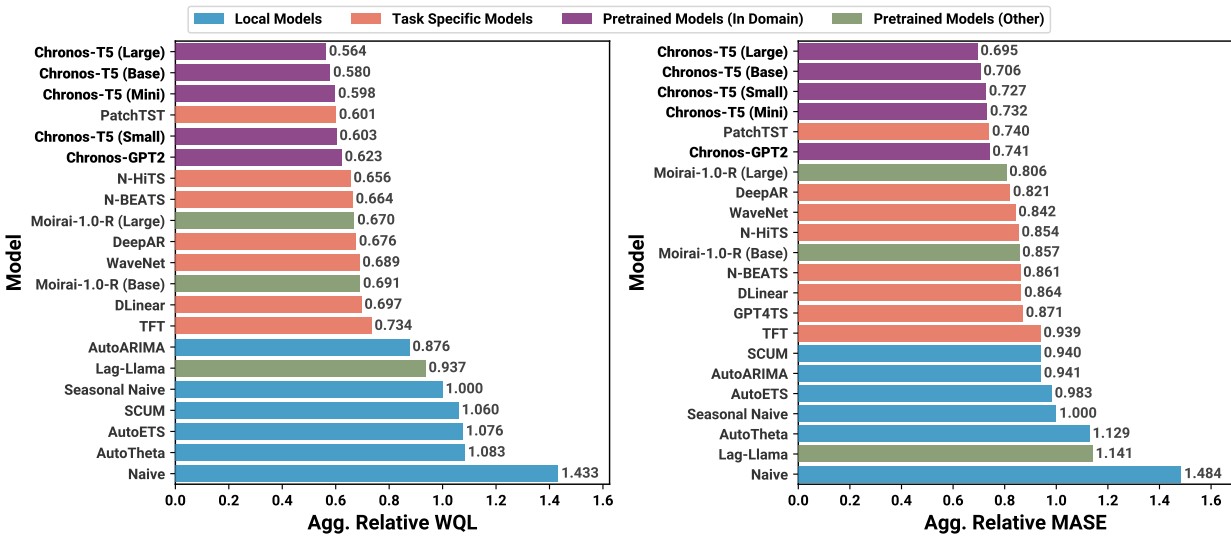

Figure 4: Performance of different models on Benchmark I, comprising 15 datasets also included in the training data of CHRONOS models. This benchmark showcases the in-domain performance of CHRONOS models against local statistical models, which fit parameters individually for each time series, task-specific models that train a separate model for each task, and pretrained models trained on a large corpus of time series data. Pretrained Models (Other) indicates that the in-domain setting does not apply to these models as they were trained on different corpora than CHRONOS. Specifically, this means that some datasets in Benchmark I were not part of their training corpus and (or) they were trained on the test sets of some datasets in Benchmark I. The probabilistic (WQL) and point (MASE) forecasting metrics (lower is better) are normalized using the scores of the Seasonal Naive baseline and aggregated through a geometric mean to obtain the aggregated relative WQL and MASE, respectively. Results for CHRONOS and task-specific models (except GPT4TS) have been averaged over 3 random seeds. Models producing point-forecasts (GPT4TS) are only compared based on MASE.

**Benchmark I** comprises 15 datasets that were also part of the training data of CHRONOS models, i.e., this benchmark evaluates the in-domain performance of CHRONOS models (see Table 3). Figure 4 summarizes the probabilistic and point forecasting performance for all models on the held-out test windows, in terms of their aggregated relative scores, computed as described in Section 5.4. The bigger CHRONOS-T5 models (Base and Large) significantly outperform baseline models, obtaining the best aggregated relative scores and average ranks (Figure 18 in Appendix E). These models not only perform better than local models (e.g., AutoETS and AutoARIMA), but they also perform better than task-specific deep learning models trained or fine-tuned for each dataset (e.g., PatchTST and DeepAR) and other pretrained models (e.g., Lag-Llama and Moirai-1.0-R).

The smaller CHRONOS-T5 models (Mini and Small) and CHRONOS-GPT2 also perform better than the majority of baselines. Between the two baseline pretrained models studied in this experiment, Moirai-1.0-R clearly outperforms Lag-Llama. Notably, the best Moirai-1.0-R model (Large, 311M) is still outperformed by the smallest Chronos-T5 model (Mini, 20M) even though Moirai-1.0-R models were trained on a significantly larger corpus of time series data. Task-specific deep learning models, trained across multiple time series for a specific task, perform better than local statistical models that fit parameters for each time series. Interestingly, the Seasonal Naive baseline performs competitively against other local models on this benchmark, suggesting that the datasets in this benchmark exhibit strong seasonal patterns. This is unsurprising since a majority of these datasets belong to domains such as energy and transport that tend to be highly seasonal in nature. The raw WQL and MASE values for individual datasets summarized in Figure 4 can be found in Tables 7 and 8 in Appendix E.

These results demonstrate the benefit of using models that are trained only once across multiple datasets, over task-specific models trained individually for each task. Such models could streamline production forecasting

systems, where forecasts from different time series tasks are required, by obviating the need for training separate models for each task.

### 5.5.2 Benchmark II: Zero-shot Results

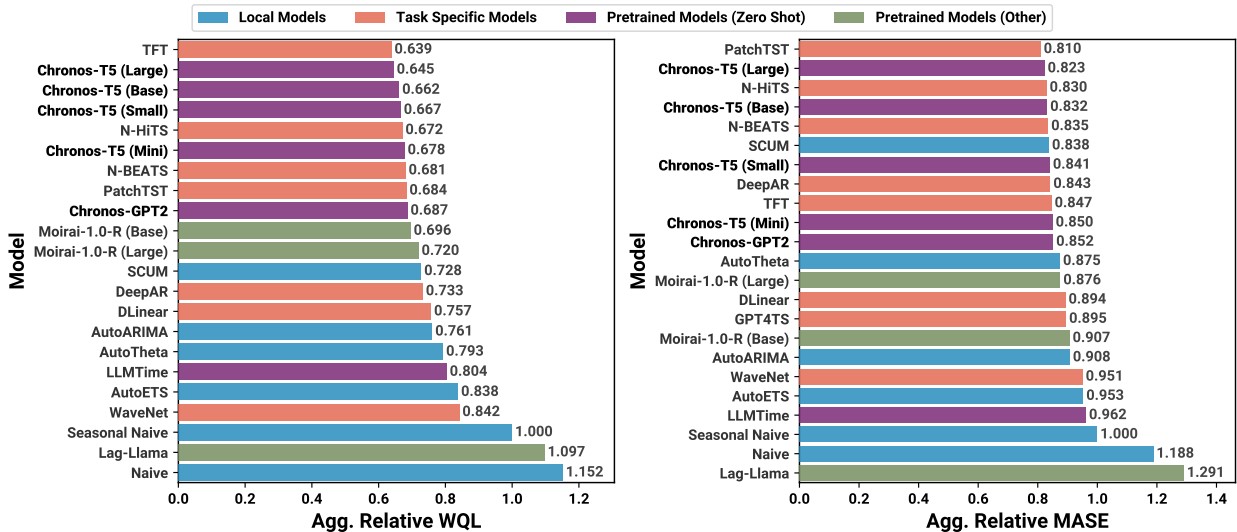

Figure 5: Performance of different models on Benchmark II, comprising 27 datasets not seen by CHRONOS models during training. This benchmark provides insights into the zero-shot performance of CHRONOS models against local statistical models, which fit parameters individually for each time series, task-specific models *trained on each task*, and pretrained models trained on a large corpus of time series data. Pretrained Models (Other) indicates that the zero-shot setting does not apply to these models as they were pretrained on some datasets in Benchmark II. The probabilistic (WQL) and point (MASE) forecasting metrics (lower is better) were normalized using the scores of the Seasonal Naive baseline and aggregated through a geometric mean to obtain the aggregated relative WQL and MASE, respectively. Results for CHRONOS and task-specific models (except GPT4TS) have been averaged over 3 random seeds. Models producing point-forecasts (GPT4TS and ForecastPFN) are only compared based on MASE.

**Benchmark II** consists of 27 datasets that were not used during CHRONOS models' training (see Table 3 in appendix B), i.e., this benchmark evaluates the zero-shot performance of these models. These datasets belong to diverse domains and frequencies, some of which are not even part of the training data, making this a challenging benchmark for CHRONOS.[5] Figure 5 summarizes the results on Benchmark II in terms of the aggregated relative scores. This benchmark is clearly more challenging than Benchmark I (Figure 4), as the best models tend to offer lower improvements relative to the baseline.

Nevertheless, despite never having seen these datasets during training, CHRONOS models significantly outperform standalone local statistical models. On probabilistic forecasting (aggregate relative WQL), CHRONOS models achieve the $2^{nd}$ to $4^{th}$ spots, performing better than most task-specific models that have been *trained on these tasks*. In terms of the point forecasting performance, CHRONOS-T5 (Large) places $2^{nd}$, surpassing most baselines, including the strong SCUM ensemble. CHRONOS models also significantly outperform other pretrained models such as Moirai-1.0-R, Lag-Llama, LLMTime, and ForecastPFN, and even GPT4TS, which fine-tunes a pretrained GPT-2 model on each dataset. Moirai-1.0-R obtains the best performance after Chronos, although the evaluation setup may have been advantageous for Moirai-1.0-R as many datasets in Benchmark II were part of its pretraining corpus. The raw WQL and MASE values for individual datasets summarized in Figure 5 can be found in Tables 9 and 10 in Appendix E.

---

[5]From a rigorous standpoint, to prevent information leakage, the start time of any dataset within this category must be after the timestamp of the last observation from the pretraining dataset and Benchmark I. Nevertheless, we consider the risk to be minimal given that the datsets bear no overlap beyond high-level conceptual categorization.

The results on this benchmark highlight the promise of CHRONOS as a generalist time series forecaster — it performs significantly better than local models that are commonly used in a zero-shot setting, and it performs on par with the best task-specific deep learning models.

**Fine tuning.** Motivated by the remarkable zero-shot performance of CHRONOS models, we conducted a preliminary investigation into fine-tuning CHRONOS models individually on datasets from Benchmark II.

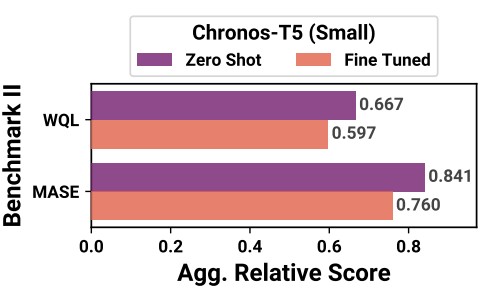

We selected the CHRONOS-T5 (Small) model for this experiment due to its good zero-shot performance with a relatively low training cost. We fine-tuned the model in a dataset-agnostic fashion with an initial learning rate of 0.001, annealed linearly to 0 over 1000 steps. Figure 6 shows that fine-tuning significantly improves the aggregate performance of the model on Benchmark II. The fine-tuned CHRONOS-T5 (Small) model now takes the top spot on Benchmark II overall, overtaking both larger (zero shot) CHRONOS models and the best task-specific models. Notably, CHRONOS-T5 (Small) is not even the most accurate variant of CHRONOS on Benchmark II in the

Figure 6: When fine-tuned on individual datasets from Benchmark II, CHRONOS-T5 (Small) significantly improves over the zero-shot performance and becomes the best performing model on average (see Figure 5).

zero shot setting, suggesting that further improvements may be obtained by fine-tuning larger CHRONOS-T5 variants.

## 5.6 Analysis of Hyperparameters

Here, we explore the effect of different design choices on the downstream model performance, beginning with a comparison of different model sizes and initializations. We then analyze the effect of training steps, synthetic data proportion, context length, and vocabulary size, on the performance of CHRONOS-T5 (Small). We only vary the parameter of interest, keeping everything else fixed to the value used in the main results.

**Model size.** We experimented with four model sizes ranging from 20M to 710M parameters.[6] Unsurprisingly, the training loss improves with the model capacity, as shown in Figure 7a. We also observe this trend in the downstream model performance — it improves with the model size for both in-domain and zero-shot benchmarks, as shown in Figure 7b. These trends suggest that even larger models may improve performance further. However, we did not explore larger models due to slow inference times which would render them impractical for real-world applications.

**Initialization.** We investigated whether initializing CHRONOS models to the corresponding T5 language models pretrained by Tay et al. (2021) on the C4 dataset (Raffel et al., 2020) has any impact on the training dynamics or the downstream performance. Figure 8 shows the training loss curve for models initialized randomly and those initialized with language model weights. Notably, models initialized randomly tend to converge to a lower training loss compared to their counterparts initialized with language model weights. For the larger models (Base and Large), models initialized with language model weights initially exhibit a faster decrease in training loss, but they ultimately converge to a higher final loss.

Overall, these observations suggest that language model weights are not particularly remarkable in the context of time series forecasting and offer no improvement over random initialization. These conclusions are further reinforced by Figure 9 which shows the downstream performance of models initialized with language model weights against three randomly-initialized models of each size. Across all model sizes, the performance of models initialized with language model weights either overlaps with or slightly underperforms compared to randomly initialized models. These results suggest that LLM initialization offers relatively little advantage in the context of time series forecasting, and instead random initialization may be the preferable choice.

---

[6] These numbers differ from the original sizes of the T5 models in Tay et al. (2021) due to the change in the vocabulary size.

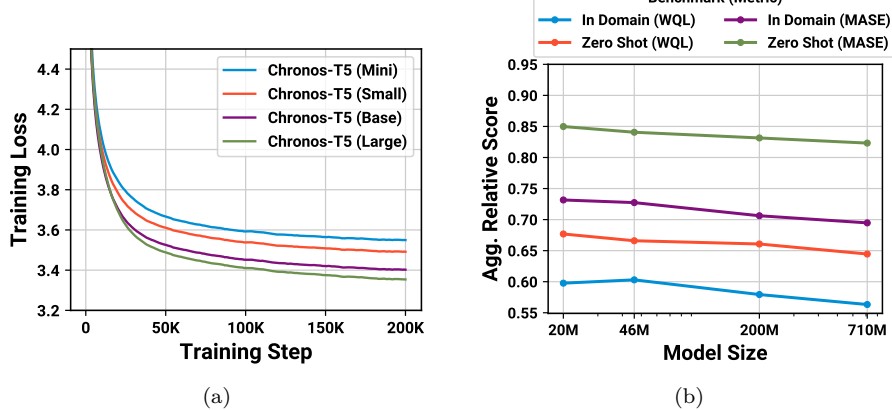

Figure 7: Model size. (a) Training loss curves of CHRONOS models of different sizes. (b) In-domain and zero-shot performance of CHRONOS models varying over model size (lower is better).

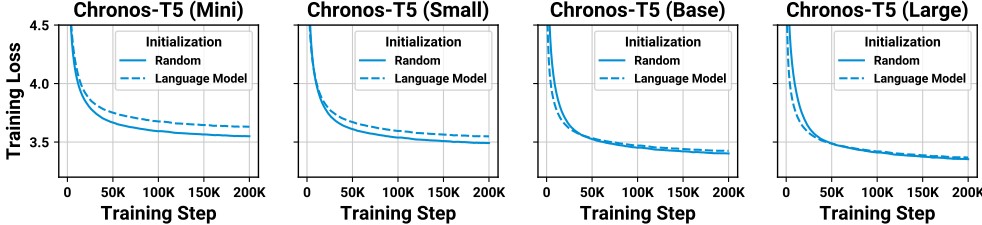

Figure 8: Initialization. Comparison of training loss of randomly-initialized CHRONOS models of different sizes against those initialized with language model weights.

**TSMixup augmentations.** As described in Section 5.2, we trained CHRONOS models on TSMixup augmentations rather than directly on the original time series. In this experiment, we investigate whether using TSMixup augmentations is advantageous for downstream performance. Figure 10a compares the performance of CHRONOS-T5 (Small, 46M) models trained with and without TSMixup augmentations. The model trained on TSMixup augmentations obtains similar in-domain performance to the model trained without augmentations. However, the zero-shot performance improves when using TSMixup augmentations. This suggests that TSMixup enchances the diversity of training data which leads to improved performance on unseen datasets. Figure 10a also shows that the zero-shot performance obtains an additional boost with the inclusion of synthetic data. We investigate this further in the next experiment.

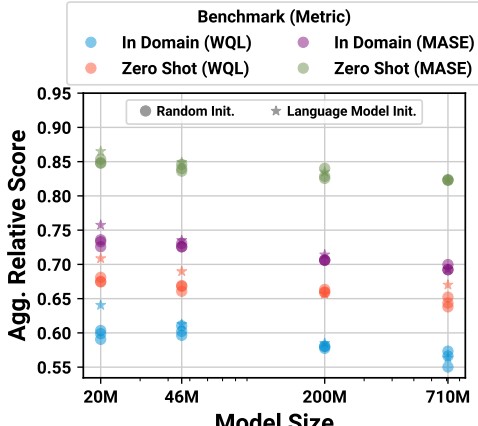

Figure 9: Comparison of the in-domain and zero-shot performance (lower is better) of models initialized with language model weights (marked as star) and three randomly initialized models (marked as circles) across different model sizes.

**Synthetic data proportion.** We systematically explored the impact of KernelSynth on downstream model performance. We trained CHRONOS-T5 (Small, 46M) models with time series sampled from TSMixup augmentations and KernelSynth data in different ratios, ranging from 0% (i.e., trained solely on TSMixup augmentations) to 100% synthetic data.

Figure 10b shows the performance of models trained with different proportions of synthetic data. Both in-domain and zero-shot metrics improve with the incorporation of synthetic data in training. The most

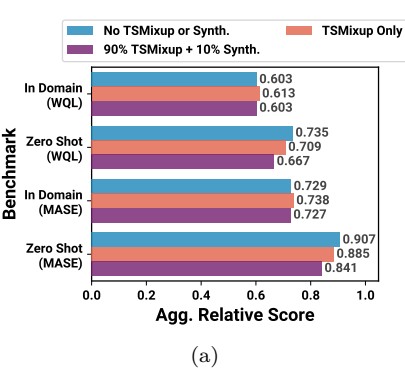
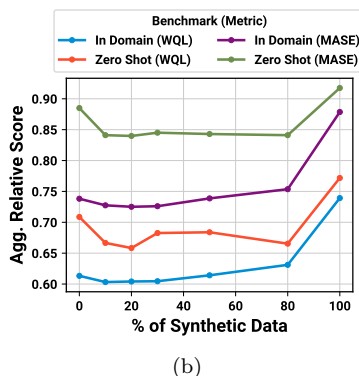

Figure 10: (a) Comparison of in-domain and zero-shot performance of CHRONOS-T5 (Small) models trained with and without TSMixup augmentations. (b) In-domain and zero-shot performance of CHRONOS-T5 (Small) models with varying proportion of KernelSynth data in the training corpus.

consistent improvement is observed around the 10% synthetic data proportion. Further increasing the proportion of synthetic data tends to worsen performance. This is unsurprising since the synthetic data generated using Gaussian processes is not representative of all real-world time series.

While the model trained only on synthetic data performs worse relative to models with real data in their training corpus, it performs reasonably well in terms of its absolute performance. Figure 20 (Appendix E) shows that it performs significantly better than ForecastPFN (Dooley et al., 2023), another model that is trained solely on synthetic data (generated differently from KernelSynth). Surprisingly, it also outperforms several other baselines in our benchmarks,[7] despite never having seen real data during training. These results attest the quality of our synthetic data, and they open up directions for future work to close the performance gap further.

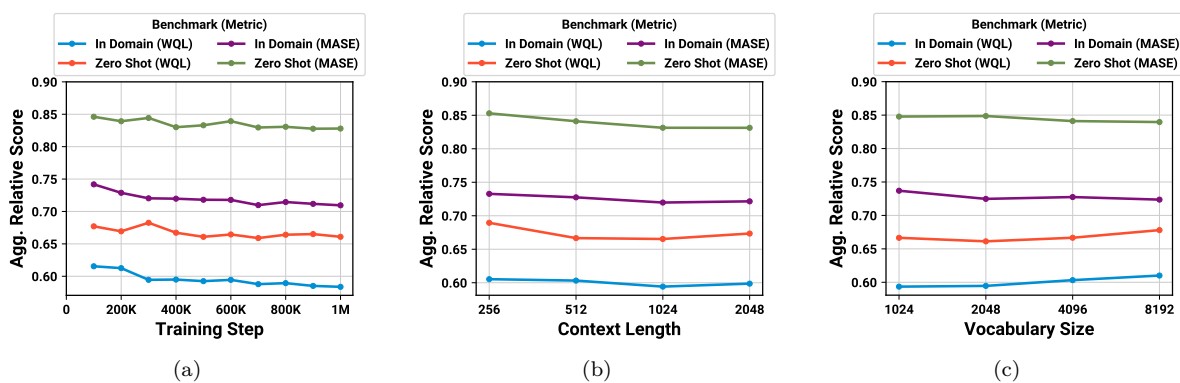

Figure 11: In-domain and zero-shot performance of a CHRONOS-T5 (Small) models varying over (a) the number of training steps, (b) the training context length, and (c) the vocabulary size.

**Training steps.** We trained a CHRONOS-T5 (Small, 46M) for 1M training steps to study the effect of longer training on model performance. Figure 11a shows that the downstream model performance improves over the course of training, both on in-domain and zero-shot benchmarks. This suggests that performance of the larger models (Base and Large) can potentially be improved by training them for longer.

**Context length.** We studied the effect of the context length on downstream performance by training CHRONOS-T5 (Small, 46M) models with four distinct context lengths. Figure 11b shows how the performance varies with increasing context length. We observe improvements on both in-domain and zero-shot metrics as

---

[7]All benchmarks are zero-shot for this model, since it was only trained on synthetic data.

context length increases up to 1024, showing that a longer context helps the models to forecast better to a certain degree. However, increasing the context length further tends to saturate or worsen the performance, which may partly be due to a limitation of our evaluation setup: it does not include enough high-frequency datasets ($>=$ 15 min). Hence, further evaluation is required to conclusively study the impact of longer context lengths. We posit that high-frequency datasets may benefit from a longer context, which may be necessary to correctly capture the long-term seasonal patterns.

**Vocabulary size.** The vocabulary size governs the precision with which the model can process the scaled time series. To explore its impact on performance, we trained CHRONOS-T5 (Small, 46M) models with varying vocabulary sizes. Figure 11c shows modest improvements in the point forecasting metric (MASE) as the vocabulary size increases. In contrast, the WQL initially improves but deteriorates for larger vocabulary sizes. We hypothesize that this behavior is an artifact of the chosen metrics. The MASE, which is invariant to the scale of individual series, is closely aligned to our training loss, which is also invariant to scale. Hence, MASE exhibits an improvement with increased precision, just as one expects for the training loss. Conversely, WQL, a scale-dependent metric, does not correlate closely with the training loss and behaves less predictably as precision increases. See Appendix D for a discussion on the properties of these metrics. Beyond this experiment, we posit that selecting the vocabulary size in the context of a model like CHRONOS would pose a trade-off. A vocabulary that is too small would lead to poor forecasting accuracy due to large discretization errors; however, a large vocabulary would lead to the bins being too fine, potentially leading to generalization errors due to fewer datapoints falling into each bin.

## 5.7 Qualitative Analysis and Limitations

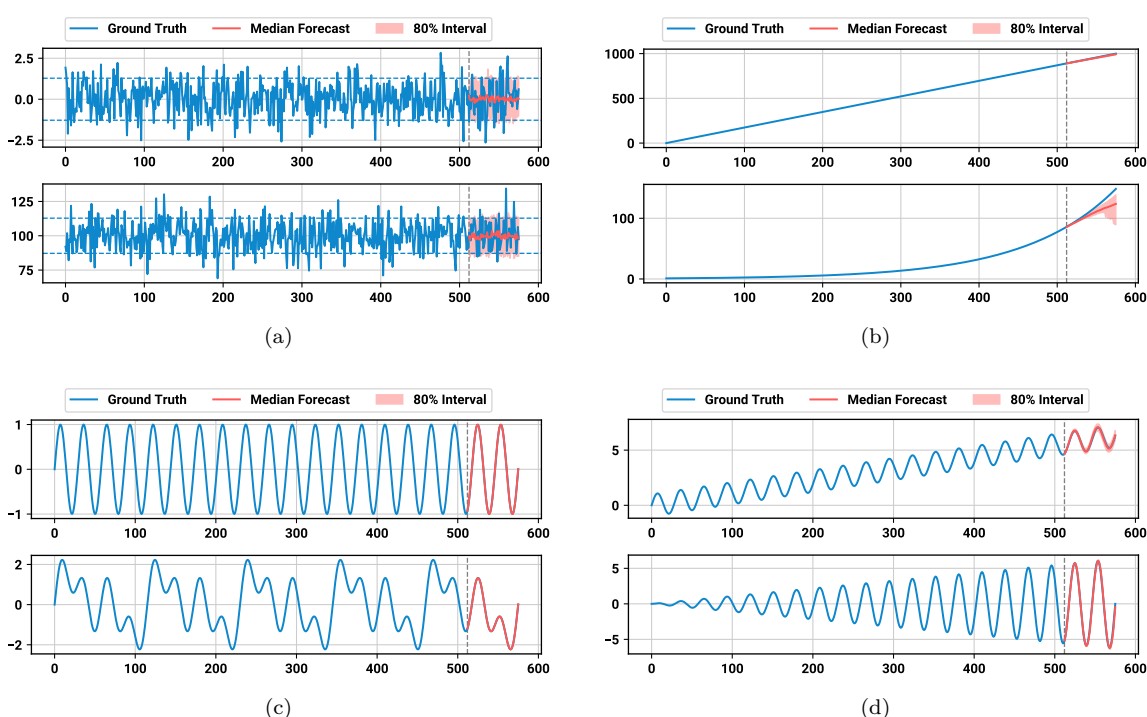

(a)

(b)

(c)

(d)

Figure 12: Forecasts generated by CHRONOS-T5 (Base) on synthetically generated patterns. (a) **Noise**: CHRONOS generates reasonable forecasts for Gaussian noise with the 80% prediction interval matching the interval of the underlying distribution (shown by the horizontal dashed blue line). (b) **Trend**: CHRONOS forecasts a linear trend (top) correctly but struggles with an exponential trend (bottom). (c) **Seasonality**: CHRONOS accurately models seasonal patterns of varying degrees of complexity (single seasonality at the top and three seasonalities at the bottom). (d) **Combined Patterns**: CHRONOS forecasts time series generated by the additive (top) or multiplicative (bottom) combination of trend and seasonal patterns accurately.

In this section, we analyze forecasts generated by CHRONOS models qualitatively, and we also highlight some limitations of our tokenization technique. We primarily focus on synthetically generated time series for a controlled analysis of different types of time series patterns. For example forecasts from real datasets, see Figures 22 to 24 in Appendix E.

**I.I.D. Noise.** We generated time series comprised purely of Gaussian observations, $\mathcal{N}(0, 1)$ and $\mathcal{N}(100, 10)$, and used CHRONOS-T5 (Base) to forecast these. Figure 12a shows that CHRONOS generates plausible forecasts for such time series and the predicted 80% interval coincides with the ground truth 80% interval shown by the dashed blue lines.

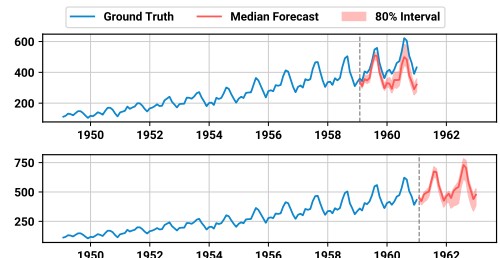

Figure 13: When the context is not sufficiently long, CHRONOS-T5 (Base) tends to underestimate trend, as shown in this example with the classic Air Passengers data (monthly) and a forecast horizon of 24. Top: with only 120 observations as context, the median prediction plateaus compared to the previous trend. Bottom: with the full context of 144 observations, the prediction picks up the trend more closely.

**Trend and seasonality.** We generated time series following linear and exponential trends: CHRONOS-T5 (Base) predicts the linear trend accurately but struggles with the exponential trend, as shown in Figure 12b. This may be due to a limited representation of exponential trends in the training data. A potential resolution for generating better forecasts for time series with exponential trends is to perform logarithmic scaling before feeding the time series into CHRONOS models. We also observed that CHRONOS models tend to underestimate the trend when the context is not sufficiently long. This phenomenon is depicted in Figure 13 where the model forecasts the pattern correctly but underpredicts the trend when a short context is provided. However, with a longer context, the model picks up the correct pattern and trend. In our analysis, we observed that CHRONOS models recognize seasonal patterns in time series particularly well. We generated purely seasonal time series using sinusoids with different frequencies. As shown in Figure 12c, CHRONOS-T5 (Base) precisely forecasts both time series. When fundamental patterns such as trend and seasonality are combined, either additively or multiplicatively, CHRONOS forecasts them accurately. This is demonstrated in Figure 12d on time series generated via addition and multiplication of a linear function with a sinusoid.

**Autoregressive processes.** An autoregressive (AR) process of order $p$ is defined as

$$X_t = \sum_{i=1}^{p} \varphi_i X_{t-i} + \varepsilon_t,$$

where $\varepsilon_t \sim \mathcal{N}(0, 1)$ and $\varphi_1, \ldots, \varphi_p$ are the parameters of the model. We generated time series from stationary AR processes of different orders ranging from 1 to 4, and we compared the forecasts generated by CHRONOS-T5 (Base) against those generated by three models: (a) the ground truth AR model that was used to generate the time series; (b) an AR model with the correct order ($p$) fitted to the time series; and (c) an AutoARIMA model fitted to the time series. Figure 14 shows the results for the AR(1) and AR(4) processes, and Figure 21 (Appendix E) shows the results for AR(2) and AR(3). We observe that CHRONOS-T5 (Base) generates plausible forecasts across all four AR processes. The simpler AR(1) and AR(2) processes are easier for the correctly-specified AR model and AutoARIMA model to fit, resulting in a better MSE than CHRONOS-T5 (Base). However, with increasing complexity in AR(3) and AR(4) processes, CHRONOS-T5 (Base) not only outperforms the AutoARIMA model (which belongs the same family as the ground truth model) but also performs on par with the fitted AR model with correct order. These results highlight that CHRONOS models can recognize fundamental patterns present in time series data.

**Flexible predictive distributions.** Using a categorical distribution to encode predictions gives CHRONOS flexibility in producing predictive distributions of different shapes. This is shown in Figure 15, illustrating kernel density estimate (KDE) plots of token IDs sampled from a CHRONOS model, for the first five time steps in the forecast horizon, across three datasets. Despite the fact that cross-entropy is not distance-aware, CHRONOS outputs predictive distributions over a contiguous set of tokens, and with different shapes,

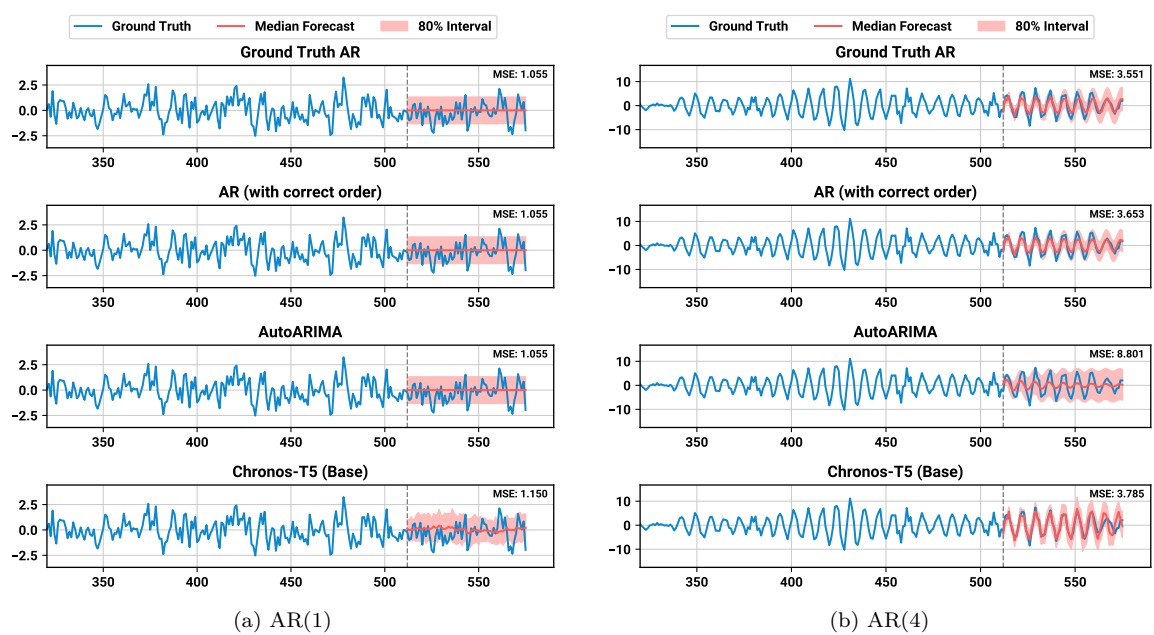

Figure 14: Forecasts generated by CHRONOS-T5 (Base) for time series generated from AR(1) and AR(4) processes compared against forecasts generated by the ground truth AR model, a fitted AR model of the correct order, and an AutoARIMA model. CHRONOS-T5 (Base) generates plausible forecasts and prediction intervals in both cases. All AR models fit the simpler AR(1) process correctly and obtain better MSE than CHRONOS-T5 (Base); however, with the increased complexity in the AR(4) process, CHRONOS-T5 (Base) performs second best after the ground truth AR model.

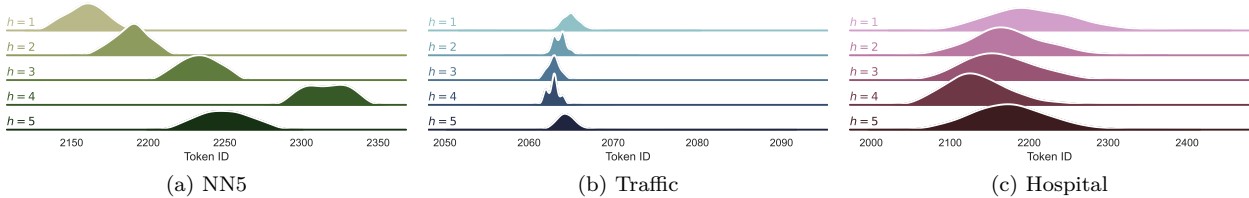

Figure 15: Forecast distributions from a CHRONOS model on series from the NN5 (Daily), Traffic, and Hospital datasets respectively. Each plot shows the predictive distribution for five prediction steps ($h = 1, \ldots, 5$): the densities were obtained via kernel density estimation from sample forecasts. Even though the cross entropy is not distance-aware, the model learns to estimate distributions over neighboring tokens, and of diverse shapes, including multimodal ones.

including multi-modal ones. Although CHRONOS learns the topology of the space directly from the data, we hypothesize that providing explicit topological information to the model during training may expedite the process and make the model robust for tokens where fewer datapoints are available. A potential method to inject topological information into the cross-entropy loss is through a type of label smoothing — assigning non-zero probability mass to tokens (i.e., bins) in the neighborhood of the the correct token. Farebrother et al. (2024) have obtained promising results with such a distance-aware regression-via-classification objective in the context of reinforcement learning. An in-depth theoretical and empirical analysis of the regression-via-classification paradigm in the context of time series forecasting would constitute interesting future research.

**Overflow and loss of precision.** One limitation of CHRONOS comes from the proposed tokenization approach (see Section 3.1). Specifically, the tokens we select represent bin centers in the range $[-15, +15]$, which ultimately represent original time series values in the range $[-15s, 15s]$, where $s$ is the scale of the time series (mean absolute value). If $s$ is very *small* compared to the range of values in the series, then

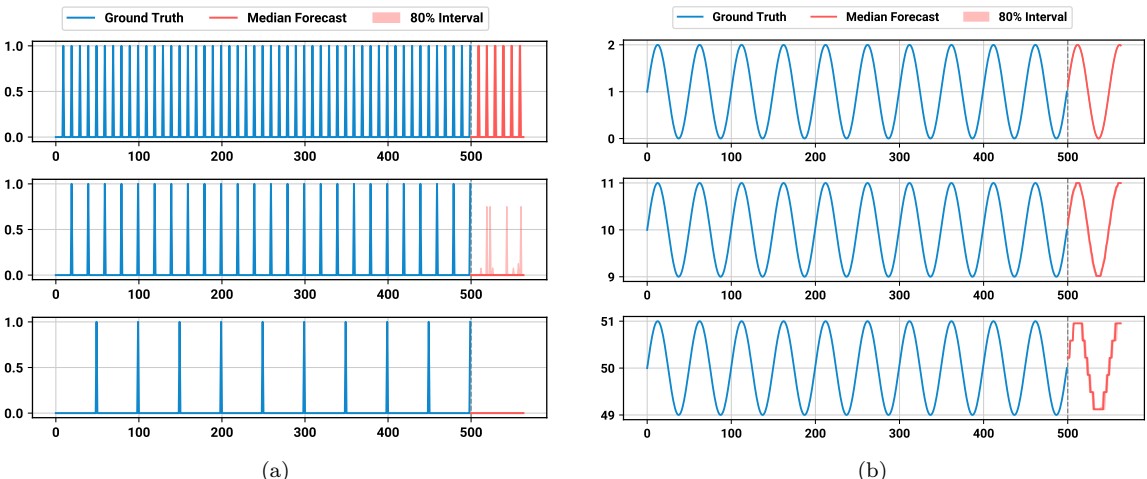

Figure 16: Loss of precision due to scaling and quantization. In (a), data consists of unit spikes every $n = 10, 20, 50$ observations (top to bottom): the scale here is $1/n$, hence the maximum representable value is $15/n$. When $1 > 15/n$ then the model cannot possibly capture the spikes appropriately (all but the top case), since their value is not represented accurately by tokens. In (b), data is a sine wave shifted up by $\mu = 1, 10, 50$: the scale here is $\mu$, and as the variance of the signal becomes smaller and smaller relative to $\mu$, the tokens precision decreases.

some observations will fall out of the representable range. An example of this behaviour is with sparse series, and as shown in Figure 16a. On the other hand, very *large* values of $s$ compared to the variance result in loss of precision: in the original space, tokens are spaced $30s/(B-1)$ from each other, where $B$ is the number of bins (we used $B = 4094$ in our experiments); values closer than that to each other may be mapped to the same token, with an apparent loss of precision. An example of this behaviour is given in Figure 16b. An inference-time heuristic solution to this problem is to preprocess the time series using an alternative normalization scheme, such as standardization, for time series with large scale and small variance. Improving the tokenization to overcome these edge cases without heuristics is subject for future work, but the results from Section 5.5 suggest that the CHRONOS models performs well on real-world data despite the limitations.

## 6 Discussion

CHRONOS represents one of the first endeavours in practical pretrained time series forecasting models, with remarkable zero-shot performance on a comprehensive collection of test datasets. This work opens up various research avenues, some of which we discuss below.

### 6.1 Beyond Zero-shot Univariate Forecasting

In our experiments, we evaluated CHRONOS in a zero-shot manner for most datasets. Such a setup highlights the competitiveness of zero-shot CHRONOS models against task-specific baselines. We expect that both in-domain and zero-shot results could be enhanced further through fine-tuning, an avenue we briefly explored in Section 5.5.2. This can be done using any parameter-efficient fine-tuning methods such as those based on low-rank adapters (LoRA) (Hu et al., 2022; Zhang et al., 2023). Alternatively, CHRONOS can be calibrated for a specific task with conformal methods (Romano et al., 2019; Stankeviciute et al., 2021; Xu & Xie, 2021). CHRONOS is especially attractive in the context of conformal prediction since it requires no training set, so all available data can be used for calibration.

In this work, we have focused on univariate forecasting of uniformly-spaced time series since it constitutes the most common of real-world time series use-cases. Nevertheless, practical forecasting tasks often involve exogenous information that must be taken into account or may require modeling of irregularly-sampled time

series (Rubanova et al., 2019; Ansari et al., 2023). One example of exogenous information is covariates, that can be either time-independent (e.g., color of the product) or time-varying (e.g., on which days the product is on sale). Another closely related problem is multivariate forecasting, where historic values of one time series (e.g., interest rates) can influence the forecast for another time series (e.g., housing prices). The number of covariates or multivariate dimensions can vary greatly across tasks, which makes it challenging to train a single model that can handle all possible combinations. A possible solution may involve training task-specific adaptors that inject the covariates into the pretrained forecasting model (Rahman et al., 2020). As another option, we can build stacking ensembles (Ting & Witten, 1997) of Chronos and other light-weight models that excel at handling covariates such as LightGBM (Ke et al., 2017).

Thus far, our exploration has centered on the problem of time series forecasting. However, several other time series analysis tasks, such as classification, clustering, and anomaly detection (Dau et al., 2018; Wu & Keogh, 2021; Ismail Fawaz et al., 2019; Goswami et al., 2024), could potentially benefit from a pretrained model like Chronos. We hypothesize that the representations learned by the encoders of Chronos-T5 models are universal and can be used for these tasks. An exploration of Chronos-T5 representations for various downstream tasks would constitute interesting future work.

## 6.2 Inference

A potential limitation of the larger Chronos models is their inference speed compared to task-specific deep learning models. Figure 17 illustrates the inference time of generating forecasts for a single time series, averaged across datasets. The inference speed of the larger Chronos models is comparable to some statistical local models. Moreover, while Chronos models are slower than task-specific models, they are not too large to be prohibitively slow. Furthermore, task-specific models need to be trained for each task individually, which requires additional time and compute. In contrast, Chronos models can be deployed for datasets with diverse history lengths, frequencies, prediction horizons, and context lengths. This makes model deployment significantly easier and drastically simplifies forecasting pipelines, obviating the need for task-specific training.

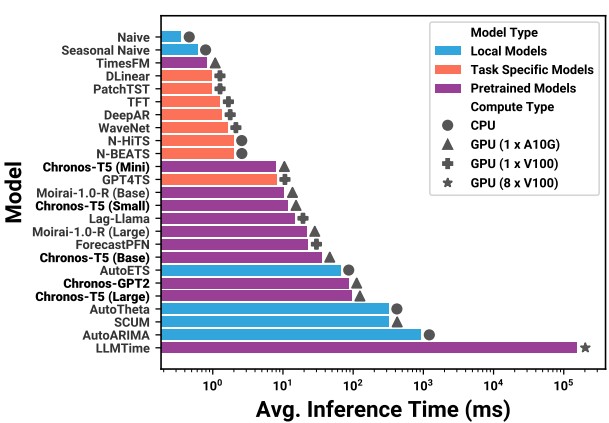

Figure 17: Inference time of different models for forecasting a single time series, averaged across datasets. The compute requirements of individual models have been highlighted.

By leveraging a language modeling framework for time series, we make developments in the NLP community immediately transferable to Chronos models. For instance, inference speed can be improved by using CUDA kernels optimized for modern Ampere GPUs, quantization (Dettmers et al., 2022), and faster decoding techniques, including speculative (Leviathan et al., 2023) and lookahead (Fu et al., 2023) decoding. Developments in long-context language models (Sun et al., 2022; Dao, 2023) may help improve Chronos models' applicability to high-frequency datasets that require longer contexts to capture seasonal patterns. Other techniques popularly used for text language models, such as temperature tuning, beam search (Freitag & Al-Onaizan, 2017), Top-K sampling (Fan et al., 2018), nucleus sampling (Holtzman et al., 2019), could enhance the quality of forecasts. These may particularly be helpful in improving the speed and quality of point forecasts, which currently require aggregation over multiple samples.

## 6.3 Data

Our findings underscore that training larger models on a large corpus of time series data yields excellent in-domain and zero-shot performance. Nevertheless, in contrast to NLP, high-quality public time series data remains limited. This poses a dilemma when training models on a large corpus of diverse datasets — selecting more datasets for training leaves fewer for zero-shot evaluation. The time series community would benefit greatly from the availability of larger time series datasets that could be used to develop and improve

pretrained model such as CHRONOS. There have been some recent efforts on building large-scale time series datasets for specific domains (Emami et al., 2023; Liu et al., 2023) and cross-domain (Borchert et al., 2022), albeit further investment is needed.

Another direction to address the problem of limited data involves developing better methods for generating synthetic time series. Our work has made significant strides in this direction by clearly demonstrating the utility of synthetic data generated using Gaussian processes, improving model performance when incorporated into the training data. Even models trained solely on synthetic data exhibit reasonable forecasting performance. Future research could delve into the failure modes of these models, proposing enhancements to bridge the gap between real and synthetic data.

## 7 Conclusion

In this work, we approach the problem of developing generalist pretrained forecasting models from the lens of a minimalist. We adapt existing language model architectures and training procedures for time series forecasting, challenging the notion that time-series-specific features or architectures are necessary for forecasting. This results in CHRONOS, a language modeling framework for time series that is, paradoxically, agnostic to time. The defining characteristic of CHRONOS is its compatibility with any language model architecture, only requiring minimal modifications — tokenization though scaling and quantization. Our pretrained models significantly outperform existing local models and task-specific deep learning baselines in terms of their in-domain performance. More remarkably, CHRONOS models obtain excellent results on unseen datasets (zero-shot performance), performing competitively with the best deep-learning baselines trained on these datasets, while showing promising evidence of further improvements through fine-tuning.

Our contributions are significant in two key aspects. First, we show that existing language model architectures are capable of performing forecasting without time-series-specific customizations. This paves the way for accelerated progress by leveraging developments in the area of LLMs and through better data strategies. Second, on a practical level, the strong performance of CHRONOS models suggests that large (by forecasting standards) pretrained language models can greatly simplify forecasting pipelines without sacrificing accuracy, offering an inference-only alternative to the conventional approach involving training and tuning a model on individual tasks.

## Acknowledgements

We are indebted to Stefano Soatto for challenging us to think about the fundamental question regarding language models and time series modeling, ultimately leading to the creation of the present work. We are grateful to our fellow researchers who have contributed to this work with insightful discussions and valuable feedback, including but not limited to George Karypis, Huzefa Rangwala, Devamanyu Hazarika, Imry Kissos, Laurent Callot, Baris Kurt, Valentin Flunkert, David Salinas, Boran Han, Xiaoyong Jin, Luke Huan, Youngsuk Park, Gaurav Gupta, Karthick Gopalswamy, Tim Januschowski, Jan Gasthaus, Bing Xiang, Kashif Rasul, Juba Nait Saada, Matthias Karlbauer, Hugo Senetaire, Mononito Goswami and Gerald Woo.

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

## A    Algorithms

Algorithm 1 and algorithm 2 present the pseudocode for TSMixup and KernelSynth, respectively.

---

**Algorithm 1** TSMixup: Time Series Mixup

---

**Input:** Time series datasets $\{\mathcal{X}_1, \ldots, \mathcal{X}_{N_d}\}$, maximum time series to be mixed $K = 3$, symmetric Dirichlet concentration parameter $\alpha = 1.5$, and (minimum, maximum) length of the augmented time series ($l_{\min} = 128, l_{\max} = 2048$).
**Output:** An augmented time series.
1:  $k \sim \mathcal{U}\{1, K\}$                                         ▷ number of time series to mix
2:  $l \sim \mathcal{U}\{l_{\min}, l_{\max}\}$                           ▷ length of the augmented time series
3:  **for** $i \leftarrow 1, k$ **do**
4:      $n \sim \mathcal{U}\{1, N_d\}$                                   ▷ sample a dataset index
5:      $\boldsymbol{x}_{1:l}^{(i)} \sim \mathcal{X}_n$                  ▷ sample a time series of length $l$ from dataset $n$
6:      $\tilde{\boldsymbol{x}}_{1:l}^{(i)} \leftarrow \dfrac{\boldsymbol{x}_{1:l}^{(i)}}{\frac{1}{l}\sum_{j=1}^{l}|x_j^{(i)}|}$   ▷ apply mean scaling to the time series
7:  **end for**
8:  $[\lambda_1, \ldots, \lambda_k] \sim \mathrm{Dir}([\alpha_1 = \alpha, \ldots, \alpha_k = \alpha])$   ▷ sample mixing weights
9:  **return** $\sum_{i=1}^{k} \lambda_i \tilde{\boldsymbol{x}}_{1:l}^{(i)}$   ▷ take weighted combination of time series

---

**Algorithm 2** KernelSynth: Synthetic Data Generation using Gaussian Processes

---

**Input:** Kernel bank $\mathcal{K}$ (see table 2), maximum kernels per time series $J = 5$, and length of the time series $l_{\mathrm{syn}} = 1024$.
**Output:** A synthetic time series $\boldsymbol{x}_{1:l_{\mathrm{syn}}}$.
1:  $j \sim \mathcal{U}\{1, J\}$                                        ▷ sample the number of kernels
2:  $\{\kappa_1(t, t'), \ldots, \kappa_j(t, t')\} \overset{\mathrm{i.i.d}}{\sim} \mathcal{K}$   ▷ sample $j$ kernels from $\mathcal{K}$
3:  $\kappa^*(t, t') \leftarrow \kappa_1(t, t')$
4:  **for** $i \leftarrow 2, j$ **do**
5:      $\star \sim \{+, \times\}$                                     ▷ sample a random binary operator
6:      $\kappa^*(t, t') \leftarrow \kappa^*(t, t') \star \kappa_i(t, t')$   ▷ compose kernels
7:  **end for**
8:  $\boldsymbol{x}_{1:l_{\mathrm{syn}}} \sim \mathcal{GP}(0, \kappa^*(t, t'))$   ▷ sample from the GP prior
9:  **return** $\boldsymbol{x}_{1:l_{\mathrm{syn}}}$

---

| Kernel | Formula | Hyperparameters |
|---|---|---|
| Constant | $\kappa_{\mathrm{Const}}(x, x') = C$ | $C = 1$ |
| White Noise | $\kappa_{\mathrm{White}}(x, x') = \sigma_n \cdot \mathbf{1}_{(x=x')}$ | $\sigma_n \in \{0.1, 1\}$ |
| Linear | $\kappa_{\mathrm{Lin}}(x, x') = \sigma^2 + x \cdot x'$ | $\sigma \in \{0, 1, 10\}$ |
| RBF | $\kappa_{\mathrm{RBF}}(x, x') = \exp\left(-\frac{\|x-x'\|^2}{2l^2}\right)$ | $l \in \{0.1, 1, 10\}$ |
| Rational Quadratic | $\kappa_{\mathrm{RQ}}(x, x') = \left(1 + \frac{\|x-x'\|^2}{2\alpha}\right)^{-\alpha}$ | $\alpha \in \{0.1, 1, 10\}$ |
| Periodic | $\kappa_{\mathrm{Per}}(x, x') = \exp\left(-2\sin^2\left(\pi \frac{\|x-x'\|}{p}\right)\right)$ | $p \in \{24, 48, 96, 168, 336, 672,$ $7, 14, 30, 60, 365, 730,$ $4, 26, 52, 6, 12, 40, 10\}$ |

Table 2: The kernel bank, $\mathcal{K}$, used in KernelSynth (algorithm 2).

# B  Datasets

The complete list of datasets used for our empirical evaluation is provided in Table 3. The table is divided into three sections, representing how the datasets were used for CHRONOS models: in total, 55 datasets where used for experiments, 13 of which for pretraining only, 15 for in-domain evaluation, and 27 for zero-shot evaluation (see also Section 5). In the following, we provide a brief description of each dataset, organized by its domain.

## B.1  Energy

**Australian Electricity** (Godahewa et al., 2021) contains electricity demand data from 5 states in Australia.

**Electricity (15 Min., Hourly, Weekly)** contains electricity consumption (in kW) for 370 households. Original data has 15 minutes frequency and was obtained from https://archive.ics.uci.edu/dataset/321/electricityloaddiagrams20112014; hourly and weekly aggreations are from Godahewa et al. (2021).

**ERCOT Load** contains hourly energy load in 8 US regions between 2004 and 2021.

**ETT (15 Min., Hourly)** (Zhou et al., 2021) contains oil temperatures and other covariates of electrical transformers from two stations in China, measured at 15 minutes granularity.

**London Smart Meters** contains half-hourly energy consumption of 5561 households in the UK between 2011 and 2014. Data was obtained from https://data.london.gov.uk/dataset/smartmeter-energy-use-data-in-london-households.

**Solar (5 Min., Hourly)** contains data about solar power generation in the US in 2006. The original data has 5 minute frequency and was obtained from https://www.nrel.gov/grid/solar-power-data.html; the hourly version was obtained via mean aggregation.

**Spanish Energy and Weather** contains 4 years of electricity consumption, generation, pricing, and weather data for Spain. Electricity data is for all of Spain, weather data is provided for each of 5 major Spanish cities. The data was obtained from https://www.kaggle.com/datasets/nicholasjhana/energy-consumption-generation-prices-and-weather.

**Wind Farms (Hourly, Daily)** (Godahewa et al., 2021) contains energy production data from wind farms in Australia. Original data was collected at 1 minute frequencey, which we aggregated to hourly and daily using the mean.

## B.2  Finance and economics

**CIF 2016** (Godahewa et al., 2021) contains banking data that was used in the CIF 2016 forecasting competition. Of all time series included, 24 are real data while the other 48 are artificially generated.

**Exchange Rate** contains daily exchange rates for currencies of eight countries (Australia, British, Canada, Switzerland, China, Japan, New Zealand and Singapore) between 1990 and 2016.

**FRED-MD** (Godahewa et al., 2021) contains monthly macro-economic indicators from the Federal Reserve Bank. Data was extracted from the FRED-MD database, and the were differenced and log-transformed.

**NN5 (Daily, Weekly)** (Godahewa et al., 2021) contains cash withdrawal data from ATMs.

## B.3  Healthcare

**Covid Deaths** (Godahewa et al., 2021) contains daily count data of COVID-19 deaths in a set of countries and states, between January and August, 2020.

**Hospital** (Godahewa et al., 2021) contains monthly time series that represent the patient counts related to medical products from January 2000 to December 2006.

Table 3: All datasets that are used for experiments. The datasets are partitioned according to how they are used for training and evaluation of CHRONOS models: *pretraining-only* data is only used for CHRONOS training; *in-domain evaluation* data is used for training CHRONOS models and other task-specific baselines, except for the *H* observations that are held out for in-domain testing only; *zero-shot evaluation* data is *not* used in training CHRONOS models, but only for evaluation (final *H* observations), as well as for training task-specific baselines (excluding the final *H* observations).

| Dataset | Domain | Freq. | Num. Series | Series Length | | | Prediction |
| | | | | min | avg | max | Length ($H$) |
|---|---|---|---|---|---|---|---|
| **Pretraining-only** | | | | | | | |
| Brazilian Cities Temperature | nature | M | 12 | 492 | 757 | 1320 | - |
| Mexico City Bikes | transport | 1H | 494 | 780 | 78313 | 104449 | - |
| Solar (5 Min.) | energy | 5min | 5166 | 105120 | 105120 | 105120 | - |
| Solar (Hourly) | energy | 1H | 5166 | 8760 | 8760 | 8760 | - |
| Spanish Energy and Weather | energy | 1H | 66 | 35064 | 35064 | 35064 | - |
| Taxi (Hourly) | transport | 1H | 2428 | 734 | 739 | 744 | - |
| USHCN | nature | 1D | 6090 | 5906 | 38653 | 59283 | - |
| Weatherbench (Daily) | nature | 1D | 225280 | 14609 | 14609 | 14610 | - |
| Weatherbench (Hourly) | nature | 1H | 225280 | 350633 | 350639 | 350640 | - |
| Weatherbench (Weekly) | nature | 1W | 225280 | 2087 | 2087 | 2087 | - |
| Wiki Daily (100k) | web | 1D | 100000 | 2741 | 2741 | 2741 | - |
| Wind Farms (Daily) | energy | 1D | 337 | 71 | 354 | 366 | - |
| Wind Farms (Hourly) | energy | 1H | 337 | 1715 | 8514 | 8784 | - |
| **In-domain evaluation** | | | | | | | |
| Electricity (15 Min.) | energy | 15min | 370 | 16032 | 113341 | 140256 | 24 |
| Electricity (Hourly) | energy | 1H | 321 | 26304 | 26304 | 26304 | 24 |
| Electricity (Weekly) | energy | 1W | 321 | 156 | 156 | 156 | 8 |
| KDD Cup 2018 | nature | 1H | 270 | 9504 | 10897 | 10920 | 48 |
| London Smart Meters | energy | 30min | 5560 | 288 | 29951 | 39648 | 48 |
| M4 (Daily) | various | 1D | 4227 | 107 | 2371 | 9933 | 14 |
| M4 (Hourly) | various | 1H | 414 | 748 | 901 | 1008 | 48 |
| M4 (Monthly) | various | 1M | 48000 | 60 | 234 | 2812 | 18 |
| M4 (Weekly) | various | 1W | 359 | 93 | 1035 | 2610 | 13 |
| Pedestrian Counts | transport | 1H | 66 | 576 | 47459 | 96424 | 48 |
| Rideshare | transport | 1H | 2340 | 541 | 541 | 541 | 24 |
| Taxi (30 Min.) | transport | 30min | 2428 | 1469 | 1478 | 1488 | 48 |
| Temperature-Rain | nature | 1D | 32072 | 725 | 725 | 725 | 30 |
| Uber TLC (Daily) | transport | 1D | 262 | 181 | 181 | 181 | 7 |
| Uber TLC (Hourly) | transport | 1H | 262 | 4344 | 4344 | 4344 | 24 |
| **Zero-shot evaluation** | | | | | | | |
| Australian Electricity | energy | 30min | 5 | 230736 | 231052 | 232272 | 48 |
| CIF 2016 | banking | 1M | 72 | 28 | 98 | 120 | 12 |
| Car Parts | retail | 1M | 2674 | 51 | 51 | 51 | 12 |
| Covid Deaths | healthcare | 1D | 266 | 212 | 212 | 212 | 30 |
| Dominick | retail | 1D | 100014 | 201 | 296 | 399 | 8 |
| ERCOT Load | energy | 1H | 8 | 154854 | 154854 | 154854 | 24 |
| ETT (15 Min.) | energy | 15min | 14 | 69680 | 69680 | 69680 | 24 |
| ETT (Hourly) | energy | 1H | 14 | 17420 | 17420 | 17420 | 24 |
| Exchange Rate | finance | 1B | 8 | 7588 | 7588 | 7588 | 30 |
| FRED-MD | economics | 1M | 107 | 728 | 728 | 728 | 12 |
| Hospital | healthcare | 1M | 767 | 84 | 84 | 84 | 12 |
| M1 (Monthly) | various | 1M | 617 | 48 | 90 | 150 | 18 |
| M1 (Quarterly) | various | 3M | 203 | 18 | 48 | 114 | 8 |
| M1 (Yearly) | various | 1Y | 181 | 15 | 24 | 58 | 6 |
| M3 (Monthly) | various | 1M | 1428 | 66 | 117 | 144 | 18 |
| M3 (Quarterly) | various | 3M | 756 | 24 | 48 | 72 | 8 |
| M3 (Yearly) | various | 1Y | 645 | 20 | 28 | 47 | 6 |
| M4 (Quarterly) | various | 3M | 24000 | 24 | 100 | 874 | 8 |
| M4 (Yearly) | various | 1Y | 23000 | 19 | 37 | 841 | 6 |
| M5 | retail | 1D | 30490 | 124 | 1562 | 1969 | 28 |
| NN5 (Daily) | finance | 1D | 111 | 791 | 791 | 791 | 56 |
| NN5 (Weekly) | finance | 1W | 111 | 113 | 113 | 113 | 8 |
| Tourism (Monthly) | various | 1M | 366 | 91 | 298 | 333 | 24 |
| Tourism (Quarterly) | various | 1Q | 427 | 30 | 99 | 130 | 8 |
| Tourism (Yearly) | various | 1Y | 518 | 11 | 24 | 47 | 4 |
| Traffic | transport | 1H | 862 | 17544 | 17544 | 17544 | 24 |
| Weather | nature | 1D | 3010 | 1332 | 14296 | 65981 | 30 |

## B.4 Nature

**Brazilian Cities Temperature** contains monthly time series representing the weather at 12 different cities in Brazil. Data is originally from NOAA, and we used the post-processed version from https://www.kaggle.com/datasets/volpatto/temperature-timeseries-for-some-brazilian-cities.

**KDD Cup 2018** (Godahewa et al., 2021) contains various air quality indicators (including PM2.5, PM10, NO2, CO, O3 and SO2), measured in 59 stations in Beijing and London, between January 1, 2017 and March 31, 2018.

**Temperature-Rain** (Godahewa et al., 2021) contains daily temperature observations and rain forecasts from 422 stations in Australia, between 2015 and 2017.

**USHCN** contains daily measurements of five climate indicators (precipitation, snow, snow depth, minimum temperature, maximum temperature) from climate stations located in 48 states in the USA. Data was obtained from https://cdiac.ess-dive.lbl.gov/ftp/ushcn_daily/.

**Weather** (Godahewa et al., 2021) contains daily time series of four weather variables (rain, mintemp, maxtemp and solar radiation) measured at weather stations in Australia.

**Weatherbench (Hourly, Daily, Weekly)** contains WeatherBench data at the spatial resolution of 5.625° (32×64 grid points). WeatherBench is a comprehensive benchmark dataset for weather prediction research and contains hourly values of the many weather-related variables over 40 years from 1979 to 2018 (including temperature, humidity, wind, precipitations). The original data has hourly frequency and was obtained from https://github.com/pangeo-data/WeatherBench; we aggregated it to daily and weekly using mean, except for "total precipitation" which was aggregated by sum.

## B.5 Retail

**Car Parts** (Godahewa et al., 2021) contains monthly sales data for various car parts, measured between January 1998 and March 2002.

**Dominick** (Godahewa et al., 2021) contains weekly time series representing the profit of individual stock keeping units from a retailer. Original data is from https://www.chicagobooth.edu/research/kilts/datasets/dominicks.

## B.6 Mobility and transport

**Mexico City Bikes** contains hourly usage statistics for 494 bike stations in Mexico City from 2010 to 2022. Each value in the time series corresponds to the number of bikes returned at the given station at the given hour of the day. Data was obtained from https://ecobici.cdmx.gob.mx/en/open-data. Time series that contain less than 50 non-zero observations were removed.

**Pedestrian Counts** (Godahewa et al., 2021) contains data from 66 sensors in Melbourne, counting pedestrians between 2009 and 2020.

**Rideshare** contains various hourly statistics of Uber and Lyft services in New York, between November 26, 2018 and December 18, 2018.

**Taxi (30 Min., Hourly)** contains spatio-temporal traffic time series of New York taxi rides taken at 1214 locations every 30 minutes in the months of January 2015 and January 2016. Original data has 30 minutes frequency, the hourly version was obtain by aggregation with sum.

**Tourism (Monthly to Yearly)** (Athanasopoulos et al., 2011; Godahewa et al., 2021) Tourism dataset from, used for the Kaggle Tourism Forecasting competition.

**Traffic** (Godahewa et al., 2021) contains hourly road occupancy readings from sensors in the San Francisco Bay area.

**Uber TLC (Hourly, Daily)** contains the number of Uber pick-ups from various locations in New York, between January and June 2015. Data was obtained from https://github.com/fivethirtyeight/uber-tlc-foil-response and aggregated hourly and daily.

### B.7 Various

**M1 (Monthly to Yearly)** (Makridakis et al., 1979; Godahewa et al., 2021) contains the time time series used in the M1 forecasting competition. Data spans micro-/macroeconomics, industry, and demographics.

**M3 (Monthly to Yearly)** (Makridakis & Hibon, 2000; Godahewa et al., 2021) contains the time time series used in the M1 forecasting competition. Data spans micro-/macroeconomics, industry, finance and demographics.

**M4 (Hourly to Yearly)** (Makridakis et al., 2020; Godahewa et al., 2021) contains data from various domains, at different sampling periods, used for the M4 forecasting competition. Domains include micro-/macroeconomics, demographic, industry, and finance.

**M5** (Makridakis et al., 2022) contains products sales data, used for the M5 forecasting competition. The data includes sales up to the end of the validation set (end of public leaderboard), but not values for the test set (private leaderboard).

### B.8 Web

**Wiki Daily (100k)** contains daily page views on the top-100k English Wikipedia articles between 2007 and 2022, ranked by number of observations (non-missing). Data was obtained from https://dumps.wikimedia.org/other/pageviews/.

## C    Baselines

We considered a total of 17 baseline methods for benchmarking CHRONOS. Local statistical baselines were AutoETS, AutoARIMA, Naive, Seasonal Naive, and AutoTheta (Assimakopoulos & Nikolopoulos, 2000); for these, we relied on implementations in the StatsForecast library (Garza et al., 2022). For task-specific deep learning architectures, DeepAR (Salinas et al., 2020), PatchTST (Nie et al., 2023), TFT (Lim et al., 2021), DLinear (Zeng et al., 2023), and WaveNet (Oord et al., 2016), we based evaluations on the implementations in GluonTS (Alexandrov et al., 2020). However, N-BEATS (Oreshkin et al., 2020) and N-HiTS (Challu et al., 2023), experiments were based on implementations in the NeuralForecast (Olivares et al., 2022) library. Finally, we used reference implementations of ForecastPFN[8] (Dooley et al., 2023), GPT4TS[9] (One-Fits-All) (Zhou et al., 2023a), LLMTime[10] (Gruver et al., 2023), Lag-Llama[11] (Rasul et al., 2023), and Moirai-1.0-R[12] (Woo et al., 2024).

WaveNet and GPT4TS models were trained on AWS EC2 `p3.2xlarge` instances which have 1 NVIDIA V100 GPUs with 16GB VRAM. All other baselines were trained on the CPU on Intel-based EC2 instances. Task-specific deep learning baselines not based on large language models (DeepAR, PatchTST, TFT, DLinear, WaveNet, N-BEATS, and N-HiTS) were trained and evaluated three times and their performance averaged in order to account for high variance inherent in their optimization.

For inference, we used EC2 CPU instances for local models, N-HiTS, and N-BEATS. The `p3.2xlarge` instance ($1 \times$ V100 16GB) was used for inference for other task-specific deep learning models and pretrained models such as Lag-Llama, Moirai-1.0-R, and ForecastPFN. Since LLMTime uses a Llama-2 70B model which has significantly larger compute requirements, LLMTime inference was performed on the `p3dn.24xlarge` AWS EC2 instance with 8 NVIDIA V100 32GB GPUs.

---

[8] https://github.com/abacusai/ForecastPFN
[9] https://github.com/DAMO-DI-ML/NeurIPS2023-One-Fits-All
[10] https://github.com/ngruver/llmtime
[11] https://github.com/time-series-foundation-models/lag-llama
[12] https://github.com/SalesforceAIResearch/uni2ts

Table 4: The multiplier used to set the context length in GPT4TS for each frequency. The context length is set equal to the multiplier times the prediction length, rounded to the nearest whole number.

| Frequency | Multiplier |
|---|---|
| 15min | 20 |
| 30min | 10 |
| 1H | 10 |
| 1D or 1B | 10 |
| 1W | 10 |
| 1M | 1.5 |
| 3M or 1Q | 1.5 |
| 1Y | 1.5 |

Statistical baselines (AutoETS, AutoARIMA, AutoTheta and SeasonalNaive) were used with their default hyperparameters in StatsForecast, but with season lengths implied by their frequencies. For example, daily frequency data had season length set to 7, hourly data 24, and so on. For this heuristic, we used the helper function `get_seasonality` from GluonTS.

Unless otherwise specified, the default hyperparameter configurations provided in baseline implementations were kept as is, and no dataset specific or global hyperparameter tuning was performed. GluonTS-based implementations were optimized with a batch size of 128, for a time limit of 4 hours and early stopping patience of 200 epochs. In PatchTST and DLinear, we experimented with two loss functions: original losses aimed at point forecasting (L1 or L2 loss) as well as default probabilistic forecasting heads used in their GluonTS implementations, where the loss is set to the negative Student's-$t$ log likelihood of the forecast horizon. Due to the consistently superior performance, our final results include the probabilistic versions of PatchTST and DLinear only. For GPT4TS, we set the context length equal to a multiple of the prediction length, with the multiplier depending on the frequency of the dataset (Table 4). We used the MASE loss function for fine-tuning in GPT4TS due to its superior performance.

For LLMTime, we experimented only with the Llama-2 70B due to the prohibitively high costs of running the benchmark through OpenAI APIs. We used the same hyperparameters as used in the Monash experiment in the original paper (Gruver et al., 2023) with a few notable differences. We set the context length to 512, same as for CHRONOS models, instead of 500. During our experiments, we observed that the default hyperparameters may lead to a significant drop in the scale of the last prediction on some datasets. To alleviate this issue, we set the `STEP_MULTIPLIER` to 1.4 (instead of 1.2) and increased the prediction length by 1 (this extra prediction is removed before computing the metrics). The inference time for LLMTime (Llama-2 70B) is ≈0.8 seconds per observation on `p3dn.24xlarge`. As an example, this will take 92 hours to generate all the predictions on the Traffic dataset (862 time series, 24 as prediction length, 20 samples). Due to the very high compute cost, we skip the evaluation of LLMTime on some large datasets.

A summary of the baseline models used along with details of hyperparameter values is provided in Table 5.

## D  Evaluation Metrics

In what follows, we consider a dataset of $N$ time series $\{\boldsymbol{x}_i = [x_{i,1}, \ldots, x_{i,C+H}]\}_{i=1}^{N}$, each spanning both the context length $C$ and prediction horizon $H$. We are interested in evaluating the accuracy of predictions for $\boldsymbol{x}_{i,C+1:C+H}$, for all $i \in \{1, \ldots, N\}$, which can be either point forecasts or probabilistic ones.

A point forecast for $\boldsymbol{x}_i$ is denoted as as $\hat{\boldsymbol{x}}_i = [\hat{x}_{i,C+1}, \ldots, \hat{x}_{i,C+H}]$. To evaluate point forecasts, we use the mean absolute scaled error (MASE, Hyndman & Koehler (2006)). For each series, this is simply the mean absolute error (MAE) divided by the empirical error of a seasonal naïve model:

$$\text{MASE}(\hat{\boldsymbol{x}}_i, \boldsymbol{x}_i) = \frac{C-S}{H} \frac{\sum_{t=C+1}^{C+H} |\hat{x}_{i,t} - x_{i,t}|}{\sum_{t=1}^{C-S} |x_{i,t} - x_{i,t+S}|},$$

where $S$ is a seasonality parameter. Since the denominator scales proportionally to $\boldsymbol{x}_i$, this error metric is independent of the scale of the data. To aggregate MASE over the entire dataset, we average over all $i$.

Table 5: Baseline models and hyperparameter choices. Hyperparameters not specified are set to defaults in their respective implementations. $C$ stands for context length, $d_h$ for hidden layer dimension, $n_L$ for number of layers, $n_H$ for number of heads, and $\eta$ for learning rate.

| Model | Model Type | Implementation | Probabilistic | Hyperparameters |
|---|---|---|---|---|
| Naive | Local | StatsForecast | Yes | N/A |
| SeasonalNaive | Local | StatsForecast | Yes | N/A |
| AutoETS | Local | StatsForecast | Yes | $C = 2500$ |
| AutoARIMA | Local | StatsForecast | Yes | $C = 1000$ |
| AutoTheta | Local | StatsForecast | Yes | $C = 2500$ |
| DeepAR | Task-specific | GluonTS | Yes | $d_h = 40, n_L = 2$ |
| TFT | Task-specific | GluonTS | Yes | $d_h = 32, n_H = 4$ |
| PatchTST | Task-specific | GluonTS | Yes | Patch length: 16, Stride: 8, $d_h = 32, n_L = 2, n_H = 4$ |
| DLinear | Task-specific | GluonTS | Yes | Kernel size: 25, $d_h = 20$ |
| WaveNet | Task-specific | GluonTS | Yes | Residual channels: 24, Skip channels: 3 |
| N-BEATS | Task-specific | NeuralForecast | No | Input size multiplier: 5 |
| N-HiTS | Task-specific | NeuralForecast | No | Input size multiplier: 5 |
| GPT4TS | Task-specific | Reference | No | Fine-tuning epochs: 100, cos: 1, tmax: 10, $n_L = 6, \eta = 10^{-3}$, with pretrained GPT-2 weights |
| ForecastPFN | Pretrained | Reference | No | $C = 100$ (as in the released pretrained model) |
| LLMTime | Pretrained | Reference | Yes | $C = 512$, STEP_MULTIPLIER $= 1.4$ (refer to the text for details) |
| Lag-Llama | Pretrained | Reference | Yes | $C = 32$ |
| Moirai-1.0-R | Pretrained | Reference | Yes | $C = 1024$, Patch length: selected by dataset-specific validation |

Probabilistic forecasts are given in terms of predicted quantiles $\boldsymbol{q}_i^{(\alpha)} = [q_{i,C+1}^{(\alpha)}, \dots, q_{i,C+H}^{(\alpha)}]$ at levels $\alpha \in (0, 1)$. To evaluate the quality of such predicted quantiles, we use the weighted quantile loss (WQL): this is an aggregation of the quantile loss (Koenker & Hallock, 2001), which is defined for the predicted $\alpha$-quantile $q$ of a real observation $x$, as

$$\mathrm{QL}_\alpha(q, x) = \begin{cases} \alpha(x - q), & \text{if } x > q, \\ (1 - \alpha)(q - x), & \text{otherwise.} \end{cases} \quad (4)$$

To aggregate Eq. (4) over multiple series and prediction instants, we consider the weighted average

$$\mathrm{WQL}_\alpha = \frac{2 \sum_{i,t} \mathrm{QL}_\alpha(q_{i,t}^{(\alpha)}, x_{i,t})}{\sum_{i,t} |x_{i,t}|}.$$

We average the above over a finite set of levels $\{\alpha_1, \dots, \alpha_K\}$ to obtain

$$\mathrm{WQL} = \frac{1}{K} \sum_{j=1}^{K} \mathrm{WQL}_{\alpha_j}.$$

In all experiments, we use quantiles at level $\alpha \in \{0.1, 0.2, \dots, 0.9\}$ to compute WQL, so that $K = 9$. Note that, being a weighted average of the quantile loss at different levels, WQL approximates (a weighted average of) the continuous ranked probability score (CRPS), a commonly used metric for evaluating probabilistic predictions (Gneiting & Raftery, 2007; Gasthaus et al., 2019). Unlike for MASE, where errors are scaled by a term proportional to the scale of each series, WQL aggregates absolute errors: as such, its value is affected by the relative scale of all series in the dataset.

## E    Additional Results

This section complements Section 5.5 by providing additional details to the experimental results. Table 6 reports the training time and cost of CHRONOS-T5 models on a p4d.24xlarge EC2 instance. Tables 7 and 8 report the raw WQL and MASE scores together with the aggregate relative score and average rank obtained by all models on the datasets in Benchmark I. Similarly, Tables 9 and 10 report these scores on Benchmark II. Figures 18 and 19 show the average ranks obtained by different models on Benchmark I and II, respectively. Figure 20 illustrates the zero-shot performance of CHRONOS-T5-Synth (Small), a model trained solely on synthetic data generated using KernelSynth, against various baselines.

Table 6: Training time and the cost of training CHRONOS models on a single `p4d.24xlarge` instance. On-demand EC2 pricing of $32.773/hr was used to compute the cost (rounded to the nearest dollar).

| Model | Training Time (hrs) | Cost (USD) |
|---|---|---|
| Chronos-T5 (Mini) | 7.68 | 252 |
| Chronos-T5 (Small) | 7.73 | 253 |
| Chronos-T5 (Base) | 17.96 | 588 |
| Chronos-T5 (Large) | 63.05 | 2066 |

Table 7: WQL scores of different models for datasets in Benchmark I, comprising 15 datasets also included in the training data of CHRONOS models. Models achieving the **_first_**, **second**, and _third_ best scores have been highlighted. Scores for CHRONOS and task-specific models have been averaged over 3 random seeds. The aggregated relative score was computed as described in Section 5.4.

| | Pretrained Models (In Domain) | | | | | Pretrained Models (Other) | | | Task Specific Models | | | | | | | Local Models | | | | | |
|---|---|---|---|---|---|---|---|---|---|---|---|---|---|---|---|---|---|---|---|---|---|
| | Chronos-T5 (Large) | Chronos-T5 (Base) | Chronos-T5 (Small) | Chronos-T5 (Mini) | ChronosGPT2 | Lag-Llama | Moirai-1.0-R (Base) | Moirai-1.0-R (Large) | PatchTST | DeepAR | WaveNet | TFT | DLinear | N-HiTS | N-BEATS | SCUM | AutoETS | AutoTheta | AutoARIMA | Seasonal Naive | Naive |
| Electricity (15 Min.) | **0.077** | 0.078 | 0.080 | 0.082 | **0.077** | 0.319 | 0.104 | 0.105 | 0.082 | 0.090 | 0.091 | 0.189 | 0.079 | 0.081 | 0.084 | - | - | 0.229 | - | 0.117 | 0.279 |
| Electricity (Hourly) | 0.101 | 0.114 | 0.105 | **0.089** | 0.117 | 0.104 | 0.121 | 0.117 | **0.089** | 0.106 | 0.109 | 0.125 | 0.095 | 0.128 | 0.127 | 0.132 | 0.129 | 0.198 | 0.126 | 0.147 | 0.363 |
| Electricity (Weekly) | **0.059** | 0.062 | 0.073 | 0.067 | 0.062 | 0.147 | 0.117 | 0.166 | 0.069 | 0.116 | 0.105 | 0.106 | 0.146 | 0.098 | 0.097 | 0.168 | 0.151 | 0.146 | 0.138 | 0.198 | 0.198 |
| KDD Cup 2018 | 0.272 | **0.268** | 0.289 | 0.271 | 0.377 | 0.369 | 0.288 | 0.278 | **0.252** | 0.330 | 0.280 | 0.571 | 0.312 | 0.302 | 0.315 | 7.631 | 2.266 | 0.521 | 0.528 | 0.556 | - |
| London Smart Meters | 0.423 | 0.428 | 0.431 | 0.436 | 0.431 | 0.384 | 0.358 | **0.350** | **0.346** | 0.405 | 0.374 | 0.365 | 0.369 | 0.358 | 0.357 | - | - | 0.660 | - | 0.541 | 0.731 |
| M4 (Daily) | 0.022 | 0.022 | **0.022** | 0.022 | **0.021** | 0.043 | 0.024 | 0.023 | 0.023 | 0.023 | 0.023 | 0.023 | 0.024 | 0.022 | 0.022 | 0.024 | 0.027 | 0.024 | 0.023 | 0.028 | 0.028 |
| M4 (Hourly) | **0.022** | 0.024 | 0.024 | 0.025 | 0.033 | 0.111 | 0.025 | **0.022** | 0.027 | 0.038 | 0.046 | 0.033 | 0.038 | 0.040 | 0.045 | 0.044 | 0.066 | 0.041 | - | 0.048 | 0.166 |
| M4 (Monthly) | 0.101 | 0.103 | 0.103 | 0.103 | 0.110 | 0.153 | 0.102 | 0.100 | 0.095 | 0.101 | 0.107 | 0.097 | 0.111 | **0.094** | **0.093** | - | 0.100 | 0.098 | - | 0.146 | 0.140 |
| M4 (Weekly) | **0.037** | **0.037** | 0.040 | 0.041 | 0.040 | 0.078 | 0.050 | 0.047 | 0.039 | 0.046 | 0.045 | 0.051 | 0.044 | 0.039 | 0.040 | 0.049 | 0.052 | 0.053 | 0.050 | 0.063 | 0.063 |
| Pedestrian Counts | **0.187** | 0.204 | 0.237 | 0.236 | 0.173 | 0.262 | 0.272 | 0.259 | 0.257 | 0.229 | 0.248 | 0.261 | 0.247 | 0.254 | 0.241 | 0.354 | 0.619 | 1.818 | 0.340 | 0.319 | 0.814 |
| Rideshare | 0.140 | 0.137 | 0.140 | **0.133** | 0.168 | 0.158 | 0.164 | 0.158 | 0.135 | **0.130** | 0.184 | 0.134 | 0.159 | 0.152 | 0.172 | 0.157 | 0.154 | 0.138 | 0.157 | 0.186 | - |
| Taxi (30 Min.) | **0.268** | **0.274** | 0.312 | 0.313 | 0.337 | 0.357 | 0.512 | 0.368 | 0.363 | 0.395 | 0.347 | 0.382 | 0.335 | 0.306 | 0.305 | - | - | 0.456 | - | 0.471 | 0.741 |
| Temperature-Rain | **0.663** | 0.669 | 0.685 | 0.704 | 0.687 | 0.717 | **0.655** | 0.685 | 0.804 | 0.718 | 0.708 | 0.670 | 0.848 | 0.780 | 0.798 | 0.886 | 1.182 | 1.060 | 0.869 | 1.424 | - |
| Uber TLC (Daily) | **0.096** | 0.097 | 0.100 | 0.105 | 0.097 | 0.176 | 0.114 | 0.107 | 0.100 | 0.110 | 0.126 | 0.111 | 0.106 | 0.116 | 0.108 | 0.162 | 0.167 | 0.190 | 0.151 | 0.231 | 0.231 |
| Uber TLC (Hourly) | **0.153** | 0.153 | 0.155 | 0.161 | 0.162 | 0.176 | 0.177 | 0.165 | 0.167 | 0.176 | 0.168 | 0.179 | 0.234 | 0.166 | 0.161 | 0.273 | 0.462 | 0.433 | 0.311 | 0.299 | 0.625 |
| **Agg. Relative Score** | **0.564** | **0.580** | 0.603 | 0.598 | 0.623 | 0.937 | 0.691 | 0.670 | 0.601 | 0.676 | 0.689 | 0.734 | 0.697 | 0.656 | 0.664 | 1.060 | 1.076 | 1.083 | 0.876 | 1.000 | 1.433 |
| **Avg. Rank** | **3.400** | 4.667 | 6.200 | 6.067 | 7.533 | 14.533 | 11.133 | 9.133 | 6.333 | 9.533 | 10.733 | 10.400 | 10.467 | 8.200 | 8.533 | 17.367 | 17.200 | 15.333 | 16.567 | 18.000 | 19.667 |

Table 8: MASE scores of different models for datasets in Benchmark I, comprising 15 datasets also included in the training data of CHRONOS models. Models achieving the **_first_**, **second**, and _third_ best scores have been highlighted. Scores for CHRONOS and task-specific models have been averaged over 3 random seeds. The aggregated relative score was computed as described in Section 5.4.

| | Pretrained Models (In Domain) | | | | | Pretrained Models (Other) | | | Task Specific Models | | | | | | | | Local Models | | | | | |
|---|---|---|---|---|---|---|---|---|---|---|---|---|---|---|---|---|---|---|---|---|---|---|
| | Chronos-T5 (Large) | Chronos-T5 (Base) | Chronos-T5 (Small) | Chronos-T5 (Mini) | ChronosGPT2 | Lag-Llama | Moirai-1.0-R (Base) | Moirai-1.0-R (Large) | PatchTST | DeepAR | WaveNet | TFT | DLinear | N-HiTS | N-BEATS | GPT4TS | SCUM | AutoETS | AutoTheta | AutoARIMA | Seasonal Naive | Naive |
| Electricity (15 Min.) | **0.391** | 0.394 | 0.418 | 0.445 | **0.388** | 1.169 | 0.707 | 0.623 | 0.450 | 0.515 | 0.637 | 1.108 | 0.452 | 0.579 | 0.567 | 0.508 | 1.706 | 1.774 | 0.583 | 1.715 | 0.498 | 1.270 |
| Electricity (Hourly) | 1.439 | 1.590 | 1.477 | **1.348** | 1.636 | 1.573 | 1.710 | 1.673 | **1.349** | 1.528 | 1.537 | 1.799 | 1.369 | 1.880 | 1.848 | 1.487 | 3.063 | 3.086 | 3.078 | 3.009 | 3.037 | 3.037 |
| Electricity (Weekly) | 1.739 | 1.801 | 1.942 | 1.954 | 1.770 | 2.979 | 2.868 | 2.758 | **1.631** | 2.517 | 1.929 | 2.800 | 2.613 | 1.975 | 2.035 | 1.880 | 3.040 | - | - | - | 3.037 | 3.037 |
| KDD Cup 2018 | 0.683 | **0.646** | 0.687 | 0.667 | 0.881 | 0.844 | 0.662 | **0.656** | 0.616 | 0.779 | 0.671 | 1.022 | 0.695 | 0.674 | 0.731 | 0.737 | 0.971 | 1.014 | 1.138 | 1.023 | 0.994 | - |
| London Smart Meters | 0.828 | 0.838 | 0.846 | 0.857 | 0.842 | 0.792 | 0.770 | **0.754** | **0.733** | 0.832 | 0.824 | 0.788 | 0.799 | 0.777 | 0.781 | 0.794 | - | - | 0.966 | - | 0.966 | 1.297 |
| M4 (Daily) | 3.114 | 3.160 | 3.148 | 3.154 | **3.079** | 8.038 | 3.448 | 3.377 | 3.450 | 3.305 | 3.306 | 3.292 | 3.461 | **3.143** | 3.155 | 5.109 | 3.224 | 3.270 | 3.335 | 3.278 | 3.278 | 3.278 |
| M4 (Hourly) | 0.882 | **0.694** | 0.721 | 0.758 | **0.710** | 3.807 | 1.210 | 0.950 | 0.967 | 1.215 | 1.613 | 1.833 | 1.867 | 3.231 | 3.457 | 1.511 | 1.300 | 1.604 | 2.458 | - | 1.193 | 1.608 |
| M4 (Monthly) | 0.960 | 0.970 | 0.982 | 0.991 | 1.044 | 2.090 | 1.032 | 1.005 | **0.962** | 1.040 | 1.101 | 1.009 | 1.022 | 0.994 | **0.942** | 0.979 | - | 0.970 | 0.966 | - | 1.260 | 1.205 |
| M4 (Weekly) | 1.998 | 2.021 | 2.113 | 2.155 | 2.225 | 5.658 | 2.484 | 2.448 | **1.996** | 2.346 | 2.523 | 2.745 | 2.429 | 2.094 | **1.976** | 3.040 | 2.394 | 2.548 | 2.657 | 2.373 | 2.777 | 2.777 |
| Pedestrian Counts | **0.272** | 0.286 | 0.304 | 0.303 | **0.271** | 0.342 | 0.354 | 0.330 | 0.339 | 0.311 | 0.334 | 0.364 | 0.327 | 0.324 | 0.315 | 0.393 | 0.382 | 0.487 | 1.275 | 0.383 | 0.369 | 0.842 |
| Rideshare | 0.865 | 0.862 | **0.554** | 0.830 | 0.921 | 0.891 | 0.910 | 0.900 | **0.827** | 0.996 | 0.983 | 1.067 | 1.148 | 0.933 | 0.919 | 1.088 | 0.944 | 0.910 | 0.970 | 1.028 | 1.250 | - |
| Taxi (30 Min.) | **0.830** | **0.849** | 0.941 | 0.944 | 1.037 | 1.069 | 1.374 | 1.088 | 1.077 | 1.158 | 1.070 | 1.113 | 1.018 | 0.950 | 0.954 | 1.113 | - | - | 1.193 | - | 1.160 | 1.768 |
| Temperature-Rain | 0.980 | 0.986 | 1.012 | 1.029 | **0.974** | 1.031 | **0.963** | 0.988 | 1.250 | 1.015 | 1.076 | 0.994 | 1.370 | 1.232 | 1.343 | 1.226 | 1.625 | 1.968 | 1.945 | 1.524 | 2.243 | - |
| Uber TLC (Daily) | 0.821 | 0.839 | 0.870 | 0.906 | **0.835** | 1.289 | 0.940 | 0.871 | **0.813** | 0.905 | 0.938 | 0.916 | 0.855 | 0.877 | 0.879 | 0.838 | 1.174 | 1.228 | 1.312 | 1.114 | 1.378 | 1.378 |
| Uber TLC (Hourly) | **0.670** | 0.673 | **0.677** | 0.689 | 0.706 | 0.711 | 0.730 | 0.716 | 0.696 | 0.703 | 0.776 | 0.746 | 0.778 | 0.716 | 0.751 | 0.754 | 0.877 | 1.009 | 1.036 | 0.982 | 0.931 | 1.390 |
| **Agg. Relative Score** | **0.695** | 0.706 | 0.727 | 0.732 | 0.741 | 1.141 | 0.857 | 0.806 | 0.740 | 0.821 | 0.842 | 0.939 | 0.864 | 0.854 | 0.861 | 0.871 | 0.940 | 0.983 | 1.129 | 0.941 | 1.000 | 1.484 |
| **Avg. Rank** | **3.333** | 4.733 | 6.067 | 6.467 | 6.933 | 14.200 | 11.533 | 9.467 | **5.733** | 10.867 | 12.133 | 13.933 | 11.800 | 9.667 | 9.400 | 12.133 | 16.500 | 16.667 | 17.333 | 17.567 | 16.667 | 19.867 |

Table 9: WQL scores of different models for datasets in Benchmark II, comprising 27 datasets not seen by CHRONOS models during training. Models achieving the **_first_**, **second**, and _third_ best scores have been highlighted. Scores for CHRONOS and task-specific models have been averaged over 3 random seeds. The aggregated relative score was computed as described in Section 5.4.

| | Pretrained Models (Zero Shot) | | | | | | Pretrained Models (Other) | | | Task Specific Models | | | | | | | Local Models | | | | | |
|---|---|---|---|---|---|---|---|---|---|---|---|---|---|---|---|---|---|---|---|---|---|---|
| | Chronos-T5 (Large) | Chronos-T5 (Base) | Chronos-T5 (Small) | Chronos-T5 (Mini) | ChronosGPT2 | LLMTime | Lag-Llama | Moirai-1.0-R (Base) | Moirai-1.0-R (Large) | PatchTST | DeepAR | WaveNet | TFT | DLinear | N-HiTS | N-BEATS | SCUM | AutoETS | AutoTheta | AutoARIMA | Seasonal Naive | Naive |
| Australian Electricity | 0.067 | 0.075 | 0.074 | 0.063 | 0.078 | 0.069 | 0.097 | 0.055 | 0.046 | 0.037 | 0.087 | 0.052 | **0.036** | 0.066 | **0.034** | 0.038 | 0.070 | 0.125 | 0.055 | 0.073 | 0.084 | 0.159 |
| Car Parts | 1.060 | 1.057 | 1.029 | 1.024 | 1.028 | - | 1.011 | 1.655 | 1.617 | 0.998 | 0.967 | 0.941 | **0.871** | 1.119 | **0.880** | 0.877 | 1.283 | 1.309 | 1.337 | - | 1.600 | - |
| CIF 2016 | 0.014 | 0.013 | 0.015 | 0.013 | 0.015 | 0.014 | 0.041 | **0.010** | 0.048 | 0.140 | 0.136 | 0.086 | **0.011** | 0.033 | 0.032 | 0.039 | 0.024 | 0.039 | 0.027 | 0.017 | 0.015 | **0.009** |
| Covid Deaths | 0.045 | 0.048 | 0.059 | 0.084 | 0.079 | **0.032** | 0.276 | 0.038 | 0.035 | 0.065 | 0.108 | 0.918 | 0.053 | 0.077 | 0.038 | 0.056 | 0.037 | 0.064 | 0.094 | **0.029** | 0.133 | 0.133 |
| Dominick | 0.332 | 0.333 | 0.338 | 0.346 | 0.336 | - | 0.443 | 0.361 | 0.346 | 0.345 | 0.364 | 0.327 | **0.320** | 0.435 | **0.313** | **0.312** | 0.439 | 0.483 | 0.485 | - | 0.453 | 0.453 |
| ERCOT Load | 0.019 | **0.016** | 0.018 | 0.018 | **0.017** | 0.053 | 0.033 | 0.019 | 0.022 | **0.017** | 0.032 | 0.024 | 0.023 | 0.020 | 0.020 | 0.050 | 0.032 | 0.122 | 0.041 | 0.052 | 0.037 | 0.181 |
| ETT (15 Min.) | 0.068 | 0.069 | 0.064 | 0.072 | 0.073 | 0.088 | 0.080 | 0.075 | 0.069 | **0.054** | 0.069 | 0.113 | 0.075 | 0.071 | **0.051** | **0.053** | 0.061 | 0.095 | 0.079 | 0.073 | 0.141 | 0.121 |
| ETT (Hourly) | 0.073 | 0.081 | 0.080 | 0.085 | 0.080 | 0.122 | 0.106 | 0.096 | 0.085 | **0.071** | 0.081 | 0.142 | 0.082 | 0.076 | 0.081 | **0.071** | 0.087 | 0.132 | 0.133 | 0.105 | 0.122 | 0.202 |
| Exchange Rate | 0.013 | 0.014 | 0.013 | 0.012 | 0.013 | 0.015 | 0.011 | 0.010 | 0.012 | 0.010 | **0.009** | 0.016 | 0.011 | **0.008** | 0.010 | 0.011 | 0.011 | 0.010 | **0.010** | 0.011 | 0.013 | 0.015 |
| FRED-MD | **0.020** | 0.022 | **0.017** | **0.017** | 0.022 | 0.041 | 0.389 | 0.045 | 0.049 | 0.042 | 0.043 | 0.058 | 0.112 | 0.069 | 0.057 | 0.061 | 0.059 | 0.055 | 0.057 | 0.056 | 0.122 | 0.064 |
| Hospital | 0.056 | 0.056 | 0.057 | 0.058 | 0.057 | 0.066 | 0.093 | 0.060 | 0.057 | 0.070 | 0.056 | 0.064 | 0.053 | 0.089 | **0.052** | **0.050** | 0.058 | 0.053 | 0.055 | 0.058 | 0.073 | 0.087 |
| M1 (Monthly) | 0.130 | **0.128** | 0.139 | 0.138 | **0.131** | 0.181 | 0.196 | 0.155 | 0.151 | 0.165 | 0.150 | 0.150 | 0.175 | 0.189 | 0.187 | 0.162 | 0.162 | 0.159 | 0.146 | 0.191 | 0.130 | 0.130 |
| M1 (Quarterly) | 0.107 | 0.105 | 0.103 | 0.103 | 0.116 | 0.115 | 0.141 | 0.111 | 0.107 | **0.078** | 0.089 | 0.094 | 0.122 | **0.079** | 0.111 | 0.085 | 0.083 | 0.083 | 0.082 | 0.091 | 0.150 | 0.130 |
| M1 (Yearly) | 0.183 | 0.181 | 0.172 | 0.179 | 0.204 | 0.144 | 0.293 | 0.194 | 0.190 | 0.165 | 0.139 | 0.168 | 0.245 | 0.198 | 0.182 | **0.135** | 0.142 | **0.137** | 0.160 | 0.209 | 0.209 | - |
| M3 (Monthly) | 0.096 | 0.097 | 0.100 | 0.099 | 0.106 | 0.108 | 0.155 | 0.102 | 0.101 | 0.113 | 0.099 | 0.100 | 0.096 | 0.101 | 0.092 | **0.094** | **0.093** | **0.095** | 0.102 | 0.149 | 0.158 | - |
| M3 (Quarterly) | 0.074 | 0.076 | 0.079 | 0.081 | 0.078 | 0.084 | 0.134 | 0.080 | 0.085 | 0.074 | 0.073 | 0.072 | **0.071** | 0.086 | 0.076 | 0.080 | 0.072 | 0.074 | 0.080 | 0.077 | **0.063** | 0.103 |
| M3 (Yearly) | 0.151 | 0.153 | 0.155 | 0.159 | 0.148 | 0.148 | 0.192 | 0.167 | 0.170 | 0.133 | **0.122** | 0.130 | 0.130 | 0.143 | 0.152 | 0.181 | 0.144 | **0.127** | **0.128** | 0.162 | 0.167 | 0.167 |
| M4 (Quarterly) | 0.082 | 0.083 | 0.084 | 0.086 | 0.087 | - | 0.132 | 0.081 | 0.080 | **0.074** | 0.080 | 0.079 | 0.080 | 0.085 | **0.073** | **0.073** | 0.079 | 0.080 | 0.079 | 0.082 | 0.119 | 0.110 |
| M4 (Yearly) | 0.134 | 0.137 | 0.136 | 0.140 | 0.148 | - | 0.178 | 0.121 | 0.138 | **0.106** | 0.111 | **0.109** | 0.110 | 0.115 | - | 0.560 | 0.114 | 0.118 | 0.115 | 0.130 | 0.161 | 0.161 |
| M5 | 0.587 | 0.586 | 0.590 | 0.595 | 0.598 | - | 0.635 | 0.692 | 0.584 | 0.597 | 0.667 | 0.594 | **0.560** | 0.563 | **0.560** | 0.653 | 0.628 | 0.636 | 0.624 | 1.024 | 1.024 | - |
| NN5 (Daily) | 0.156 | 0.161 | 0.169 | 0.173 | 0.162 | 0.242 | 0.261 | 0.181 | 0.162 | **0.149** | 0.155 | 0.154 | **0.145** | 0.159 | 0.149 | **0.147** | 0.293 | 0.264 | 0.294 | 0.312 | 0.425 | 0.425 |
| NN5 (Weekly) | 0.091 | 0.091 | 0.090 | 0.091 | 0.094 | 0.092 | 0.111 | 0.092 | 0.093 | **0.081** | **0.087** | 0.098 | **0.086** | 0.090 | 0.098 | 0.114 | 0.088 | 0.090 | 0.090 | 0.090 | 0.123 | 0.123 |
| Tourism (Monthly) | 0.100 | 0.103 | 0.113 | 0.109 | 0.095 | 0.125 | 0.213 | 0.121 | 0.111 | 0.092 | 0.092 | 0.104 | 0.096 | 0.101 | 0.092 | **0.084** | **0.083** | 0.090 | 0.091 | 0.093 | 0.104 | 0.297 |
| Tourism (Quarterly) | **0.061** | 0.069 | 0.069 | 0.074 | 0.071 | 0.100 | 0.085 | 0.074 | 0.072 | 0.074 | 0.072 | 0.074 | 0.080 | 0.077 | **0.063** | 0.075 | 0.070 | **0.061** | 0.102 | 0.104 | 0.119 | 0.166 |
| Tourism (Yearly) | 0.183 | 0.207 | 0.200 | 0.218 | 0.194 | 0.163 | 0.238 | 0.168 | 0.161 | 0.136 | **0.127** | 0.179 | **0.102** | 0.165 | 0.139 | 0.154 | 0.162 | 0.159 | 0.176 | 0.156 | 0.209 | 0.209 |
| Traffic | 0.256 | 0.264 | 0.263 | 0.264 | 0.254 | 0.287 | 0.256 | **0.225** | **0.231** | 0.246 | **0.233** | 0.234 | 0.264 | 0.250 | 0.263 | 0.270 | - | 0.557 | 0.905 | - | 0.362 | 0.643 |
| Weather | **0.139** | 0.140 | 0.142 | 0.150 | 0.144 | - | 0.164 | **0.135** | **0.132** | 0.143 | 0.147 | 0.152 | 0.151 | 0.174 | 0.143 | 0.144 | 0.174 | 0.214 | 0.217 | 0.185 | 0.217 | 0.227 |
| **Agg. Relative Score** | **0.645** | 0.662 | 0.667 | 0.678 | 0.687 | 0.804 | 1.097 | 0.696 | 0.720 | 0.684 | 0.733 | 0.842 | **0.639** | 0.757 | 0.672 | 0.681 | 0.728 | 0.838 | 0.793 | 0.761 | 1.000 | 1.152 |
| **Avg. Rank** | **8.333** | 9.407 | 9.889 | 11.296 | 11.185 | 15.352 | 18.148 | 11.778 | 11.259 | **7.037** | **8.333** | 11.407 | **7.111** | 12.333 | 9.037 | 8.741 | 10.093 | 10.852 | 10.444 | 12.778 | 18.667 | 19.519 |

Table 10: MASE scores of different models for datasets in Benchmark II, comprising 27 datasets not seen by CHRONOS models during training. Models achieving the **first**, **second**, and third best scores have been highlighted. Scores for CHRONOS and task-specific models have been averaged over 3 random seeds. The aggregated relative score was computed as described in Section 5.4.

| | Pretrained Models (Zero Shot) | | | | | | | Pretrained Models (Other) | | | Task Specific Models | | | | | | | | Local Models | | | | | |
|---|---|---|---|---|---|---|---|---|---|---|---|---|---|---|---|---|---|---|---|---|---|---|---|---|
| | Chronos-T5 (Large) | Chronos-T5 (Base) | Chronos-T5 (Small) | Chronos-T5 (Mini) | Chronos-GPT2 | LLMTime | ForecastPFN | Lag-Llama | Moirai-1.0-R (Base) | Moirai-1.0-R (Large) | PatchTST | DeepAR | WaveNet | TFT | DLinear | N-HiTS | N-BEATS | GPT4TS | SCUM | AutoETS | AutoTheta | AutoARIMA | Seasonal Naive | Naive |
| Australian Electricity | 1.333 | 1.319 | 1.399 | 1.114 | 1.310 | 1.186 | 2.158 | 1.635 | 1.258 | 1.009 | 0.871 | 1.473 | 0.997 | **0.810** | 1.278 | 0.794 | 0.828 | 1.161 | 1.427 | 2.391 | 0.897 | 1.393 | 1.253 | 2.362 |
| Car Parts | 0.906 | 0.899 | 0.887 | 0.891 | 0.881 | - | 2.657 | 0.816 | 1.735 | 1.542 | 0.803 | **0.798** | 0.817 | 0.799 | 0.879 | 0.803 | 0.803 | 0.891 | 1.157 | 1.185 | 1.229 | - | 1.201 | - |
| CIF 2016 | 0.986 | 0.981 | 0.989 | 1.051 | 1.046 | 1.384 | 3.588 | 2.235 | 1.197 | 1.160 | 1.537 | 1.363 | 1.309 | 1.553 | 1.145 | 1.389 | 1.440 | 0.960 | **0.907** | 0.957 | 1.002 | 1.006 | 1.299 | 1.263 |
| Covid Deaths | 42.550 | 42.687 | 42.670 | 43.621 | 48.215 | 32.143 | 91.515 | 78.456 | 33.062 | 33.108 | 36.465 | 38.203 | 102.457 | **30.635** | 40.418 | 31.771 | 31.730 | 75.909 | 33.595 | 38.114 | 45.407 | 31.705 | 46.912 | 46.912 |
| Dominick | 0.818 | 0.816 | 0.819 | 0.833 | 0.820 | - | 3.274 | 1.250 | 0.879 | 0.845 | 0.867 | 0.851 | 0.812 | 0.800 | 0.880 | **0.782** | **0.782** | 1.813 | 0.891 | 0.885 | 1.016 | - | 0.871 | 0.871 |
| ERCOT Load | 0.617 | **0.558** | 0.573 | 0.588 | 0.561 | 1.319 | 3.975 | 0.834 | 0.583 | 0.667 | **0.553** | 1.197 | 0.780 | 0.690 | 0.651 | 0.615 | 0.648 | 0.558 | 1.308 | 2.826 | 1.306 | 1.284 | 0.761 | 4.234 |
| ETT (15 Min.) | 0.741 | 0.739 | 0.710 | 0.792 | 0.796 | 1.042 | 1.138 | 0.967 | 0.981 | 0.753 | 0.652 | 0.874 | 1.339 | 0.962 | 0.724 | 0.643 | 0.659 | 0.574 | 0.673 | 1.183 | **0.583** | 0.879 | 1.169 | 1.164 |
| ETT (Hourly) | 0.735 | 0.789 | 0.789 | 0.797 | 0.768 | 1.232 | 1.833 | 1.002 | 0.902 | 0.845 | **0.729** | 0.814 | 1.509 | 0.875 | **0.695** | 0.811 | 0.782 | 0.768 | 0.850 | 1.139 | 0.900 | 0.977 | 0.932 | 1.651 |
| Exchange Rate | 2.375 | 2.433 | 2.252 | 2.030 | 2.335 | 1.743 | 7.583 | 3.087 | **1.507** | 1.909 | 1.540 | 1.615 | 3.105 | 2.361 | 1.459 | 2.041 | 2.149 | 2.709 | 1.749 | 1.643 | 1.648 | 1.882 | 1.740 | 1.874 |
| FRED-MD | 0.500 | 0.486 | 0.496 | 0.483 | **0.468** | 0.513 | 2.621 | 2.283 | 0.607 | 0.593 | 0.745 | 0.621 | 0.849 | 0.929 | 0.713 | 0.696 | 0.635 | 0.693 | 0.492 | 0.544 | 0.566 | **0.473** | 1.101 | 0.622 |
| Hospital | 0.810 | 0.810 | 0.815 | 0.817 | 0.831 | 0.861 | 1.775 | 0.939 | 0.821 | 0.826 | 0.859 | 0.804 | 0.857 | 0.799 | 0.940 | 0.781 | **0.760** | 0.793 | 0.748 | 0.760 | 0.761 | 0.820 | 0.921 | 0.968 |
| M1 (Monthly) | 1.090 | 1.117 | 1.169 | 1.174 | 1.182 | 1.415 | 2.172 | 1.875 | 1.272 | 1.238 | 1.208 | 1.122 | 1.266 | 1.326 | 1.369 | 1.333 | 1.236 | 1.198 | **1.023** | 1.072 | 1.099 | 1.153 | 1.314 | 1.468 |
| M1 (Quarterly) | 1.713 | 1.739 | 1.764 | 1.785 | 1.785 | 1.802 | 9.931 | 3.036 | 1.896 | 1.840 | 1.920 | 1.741 | 1.904 | 2.144 | 1.943 | 2.061 | 2.043 | 1.958 | 1.692 | 1.710 | **1.683** | 1.770 | 2.078 | 1.952 |
| M1 (Yearly) | 4.301 | 4.624 | 4.659 | 4.958 | 4.751 | 4.077 | 23.089 | 7.149 | 4.623 | 4.708 | 4.042 | 3.685 | 4.727 | 4.316 | 11.565 | 5.568 | 6.212 | 3.675 | **3.571** | 4.110 | 3.697 | 3.870 | 4.894 | 4.894 |
| M3 (Monthly) | **0.857** | 0.868 | 0.885 | 0.900 | 0.930 | 0.996 | 2.240 | 1.846 | 0.946 | 0.924 | 1.225 | 0.943 | 0.950 | 0.916 | 1.161 | 0.899 | 0.883 | 0.950 | 0.827 | 0.869 | 0.861 | 0.933 | 1.146 | 1.175 |
| M3 (Quarterly) | 1.181 | 1.199 | 1.256 | 1.289 | 1.241 | 1.450 | 10.176 | 2.886 | 1.428 | 1.429 | 1.264 | 1.209 | 1.257 | 1.160 | 1.572 | 1.202 | 1.147 | 1.448 | 1.135 | **1.125** | 1.130 | 1.419 | 1.425 | 1.464 |
| M3 (Yearly) | 3.106 | 3.209 | 3.276 | 3.385 | 3.158 | 3.140 | 18.728 | 5.114 | 3.661 | 3.822 | 2.949 | 2.827 | 3.026 | 2.960 | 3.435 | 3.432 | 3.547 | 3.418 | 2.703 | 2.696 | **2.613** | 3.165 | 3.172 | 3.172 |
| M4 (Quarterly) | 1.216 | 1.231 | 1.246 | 1.271 | 1.312 | - | 6.927 | 2.663 | 1.286 | 1.259 | 1.150 | 1.254 | 1.241 | 1.248 | 1.229 | 1.157 | **1.129** | 1.215 | 1.145 | 1.188 | 1.193 | 1.276 | 1.602 | 1.477 |
| M4 (Yearly) | 3.606 | 3.678 | 3.651 | 3.743 | 3.933 | - | - | 5.866 | 3.599 | 4.175 | 3.072 | 3.178 | 3.221 | 3.119 | 3.295 | - | - | 3.374 | **3.013** | 3.374 | 3.124 | 3.730 | 3.974 | 3.974 |
| M5 | 0.944 | 0.939 | 0.940 | 0.944 | 0.969 | - | 1.530 | 0.965 | 1.442 | 0.929 | 0.919 | 0.956 | 0.959 | **0.909** | 1.027 | 0.917 | 0.917 | 0.935 | 1.096 | 1.101 | 1.100 | 1.057 | 1.399 | 1.399 |
| NN5 (Daily) | 0.573 | 0.585 | 0.615 | 0.642 | 0.601 | 0.953 | 1.375 | 0.992 | 0.698 | 0.625 | 0.575 | 0.585 | 0.585 | **0.556** | 0.604 | 0.571 | 0.571 | 0.720 | 1.052 | 1.039 | 1.073 | 1.214 | 1.292 | 1.292 |
| NN5 (Weekly) | 0.940 | 0.938 | 0.944 | 0.947 | 0.963 | 0.968 | 1.349 | 1.141 | 0.980 | 1.009 | 0.877 | 0.920 | 1.034 | **0.896** | 0.966 | 0.919 | 1.014 | 1.268 | 0.974 | 0.978 | 0.984 | 0.995 | 1.063 | 1.063 |
| Tourism (Monthly) | 1.761 | 1.828 | 1.900 | 1.950 | 1.783 | 2.139 | 4.348 | 3.030 | 2.039 | 1.910 | 1.572 | 1.529 | 1.629 | 1.686 | 1.514 | 1.486 | 1.573 | 1.573 | **1.441** | 1.497 | 1.680 | 1.573 | 1.631 | 3.591 |
| Tourism (Quarterly) | 1.677 | 1.717 | 1.730 | 1.829 | 1.828 | 1.916 | 5.595 | 3.695 | 2.722 | 2.281 | 1.723 | 1.586 | 1.769 | 1.729 | 1.690 | **1.585** | 1.618 | 1.750 | 1.501 | 1.590 | 1.658 | 1.661 | 1.699 | 3.633 |
| Tourism (Yearly) | 3.755 | 3.900 | 3.901 | 4.048 | 3.862 | 3.309 | 12.093 | 3.755 | **3.047** | 3.301 | 3.138 | 3.702 | 4.130 | **3.047** | 3.406 | 3.448 | 3.564 | - | 3.276 | 3.138 | 3.078 | 4.043 | 3.552 | 3.552 |
| Traffic | 0.804 | 0.828 | 0.837 | 0.850 | 0.818 | 0.973 | 1.909 | 0.829 | 0.726 | 0.759 | 0.790 | **0.737** | 0.797 | 0.880 | 0.821 | 0.927 | 0.968 | 0.787 | - | 1.685 | 1.794 | - | 1.077 | 2.052 |
| Weather | **0.822** | 0.824 | 0.836 | 0.853 | 0.858 | 0.832 | 2.003 | 1.001 | 0.831 | **0.807** | 0.860 | 0.911 | 0.945 | 0.913 | 0.997 | 0.910 | 0.888 | 0.972 | 0.933 | 1.079 | 0.991 | 0.907 | 1.004 | 1.004 |
| Agg. Relative Score | **0.823** | 0.832 | 0.841 | 0.850 | 0.852 | 0.962 | 2.450 | 1.291 | 0.907 | 0.876 | 0.810 | 0.843 | 0.951 | 0.847 | 0.894 | 0.830 | 0.835 | 0.895 | 0.838 | 0.953 | 0.875 | 0.908 | 1.000 | 1.188 |
| Avg. Rank | 8.481 | 9.296 | 10.593 | 12.037 | 11.630 | 16.593 | 23.204 | 19.667 | 13.037 | 12.444 | **8.222** | 9.111 | 14.074 | 9.778 | 12.704 | 9.463 | 9.648 | 12.111 | 8.204 | 10.704 | 9.593 | 13.444 | 16.778 | 19.185 |

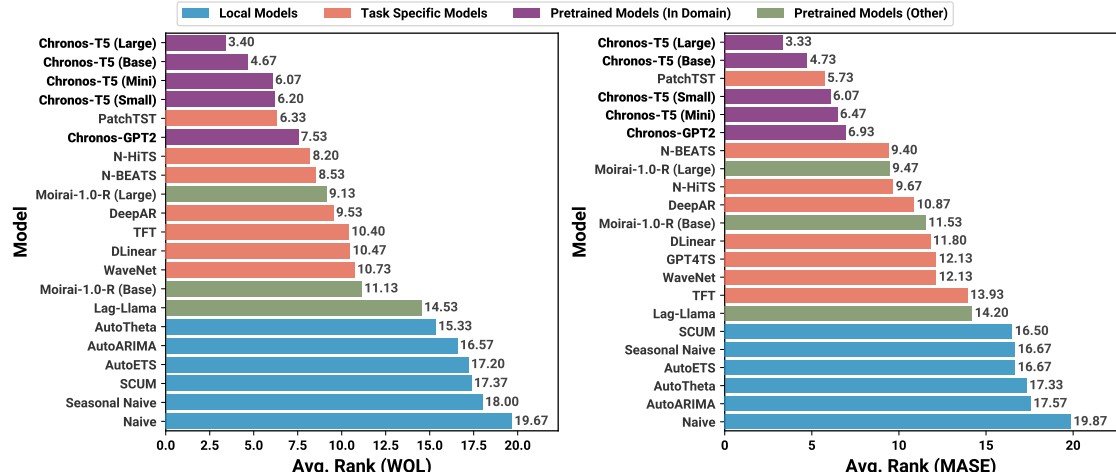

Figure 18: Average rank of different models on Benchmark I, comprising 15 datasets also included in the training data of CHRONOS models.

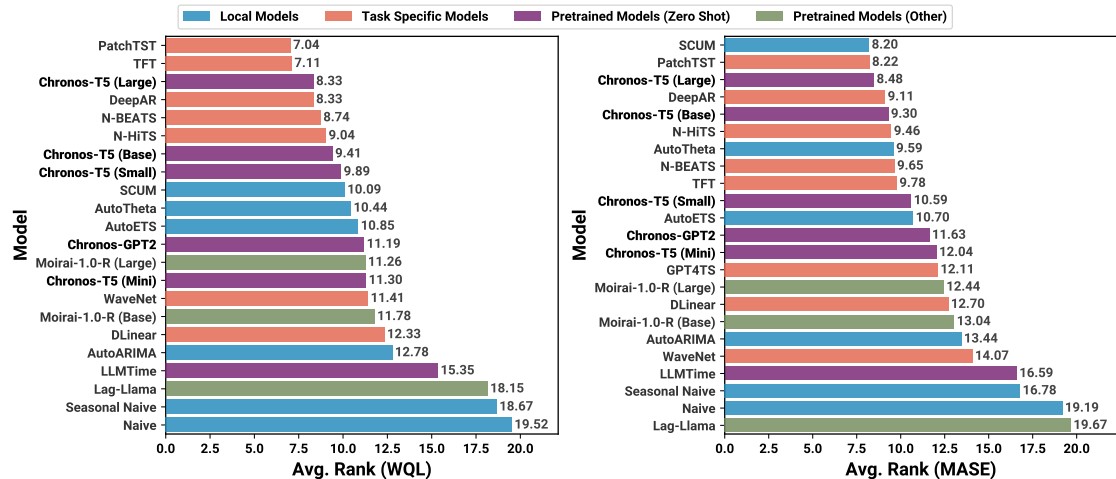

Figure 19: Average rank of different models on Benchmark II, comprising 27 datasets not seen by CHRONOS models during training.

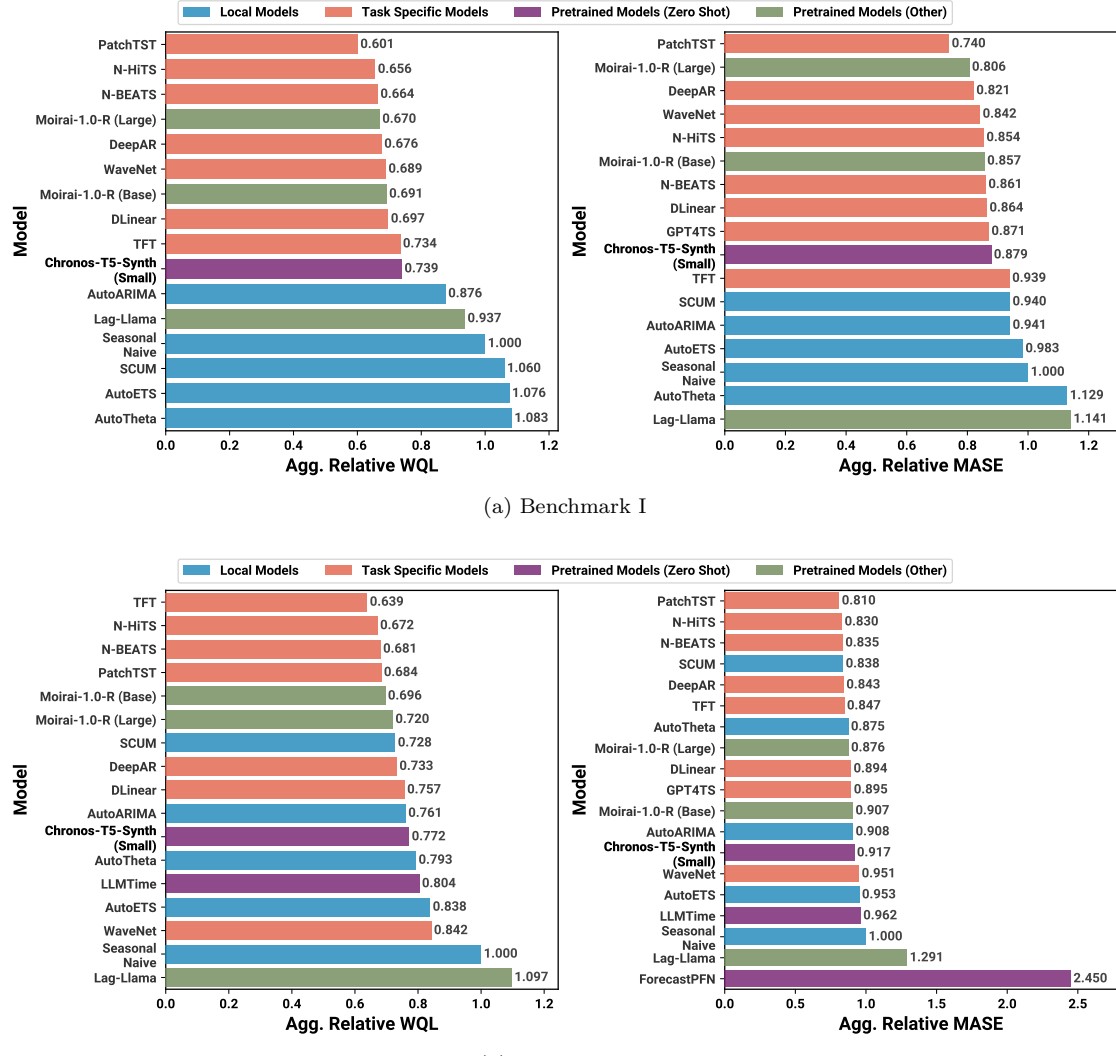

(a) Benchmark I

(b) Benchmark II

Figure 20: Performance of CHRONOS-T5-Synth (Small), a CHRONOS model that was only trained on synthetic data, on Benchmark I and II, against local and task-specific models. Note that unlike other CHRONOS models also trained on real data, both these benchmarks are zero-shot for CHRONOS-T5-Synth (Small).

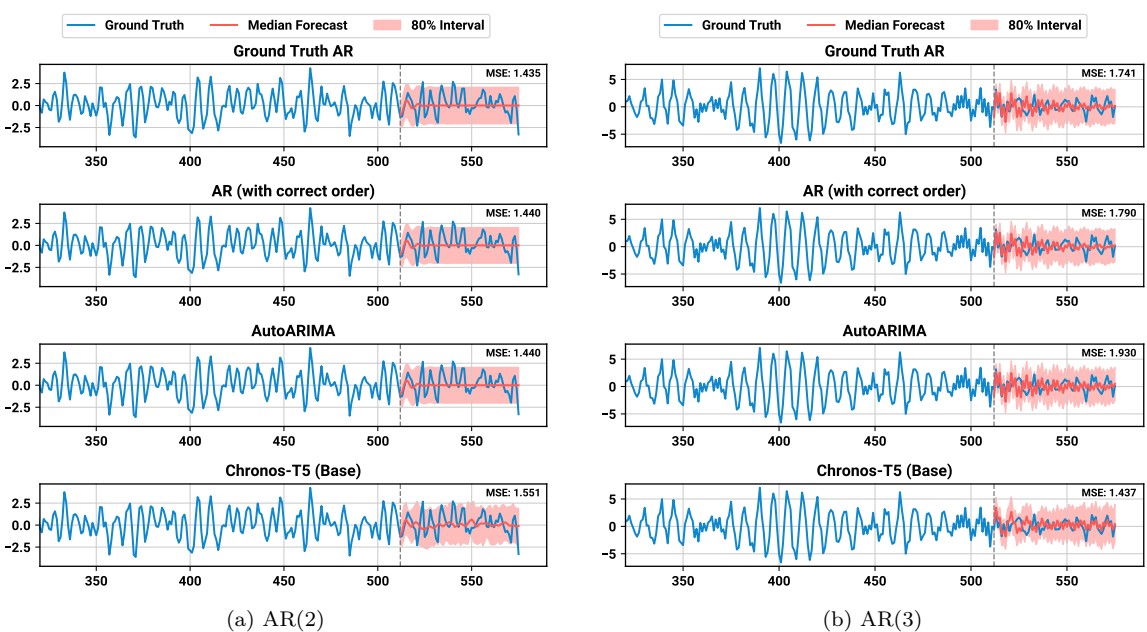

(a) AR(2)                                                    (b) AR(3)

Figure 21: Forecasts generated by Chronos-T5 (Base) for time series generated from AR(2) and AR(3) processes compared against forecasts generated by the ground truth AR model, a fitted AR model of the correct order, and an AutoARIMA model. Chronos-T5 (Base) generates plausible forecasts and prediction intervals in both cases. All AR models fit the simpler AR(2) process well and obtain better MSE than Chronos-T5 (Base); however, with the increased complexity in the AR(3) process, Chronos-T5 (Base) performs better than other models.

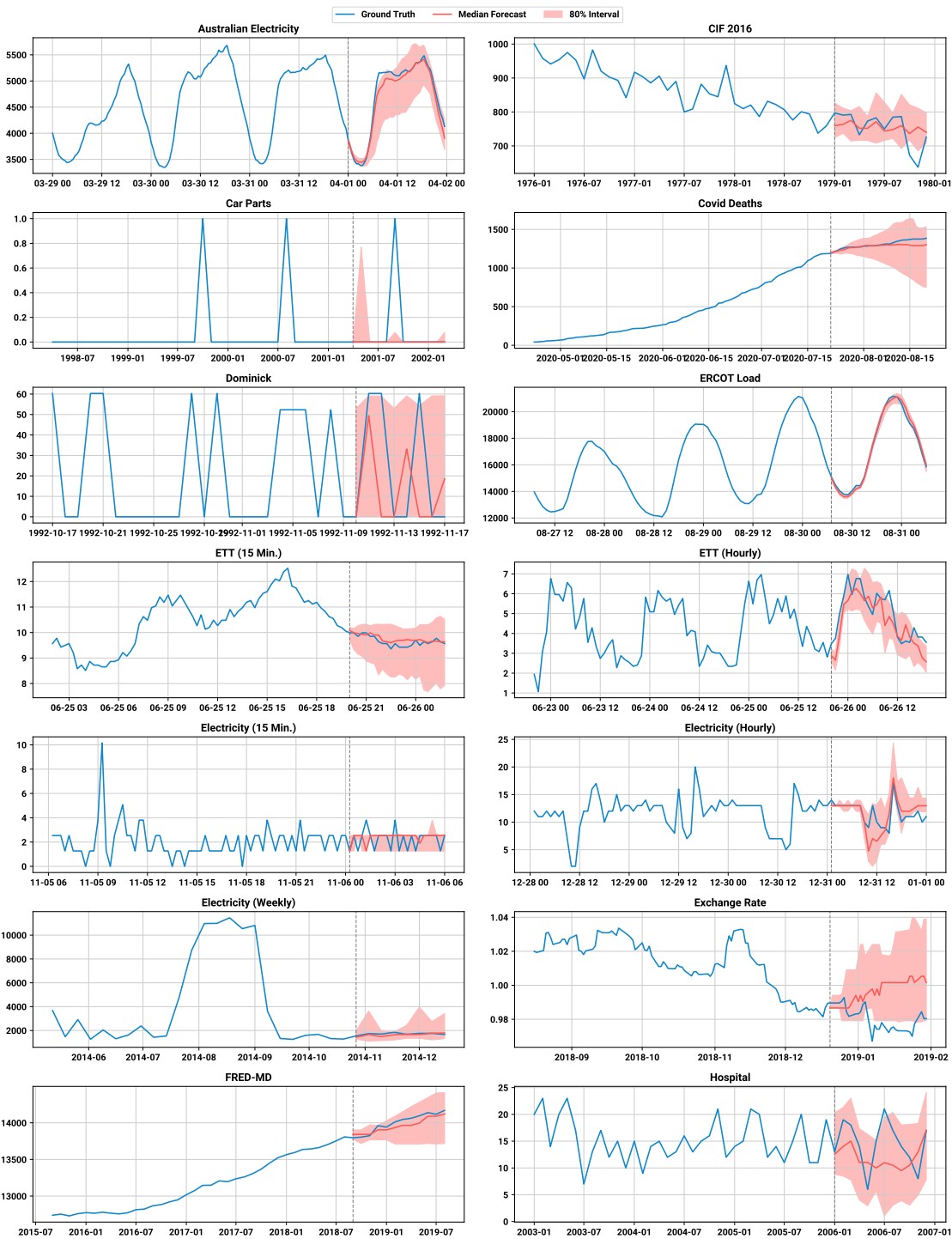

Figure 22: Example of forecasts from CHRONOS-T5 (Base) on the test datasets used in experiments.

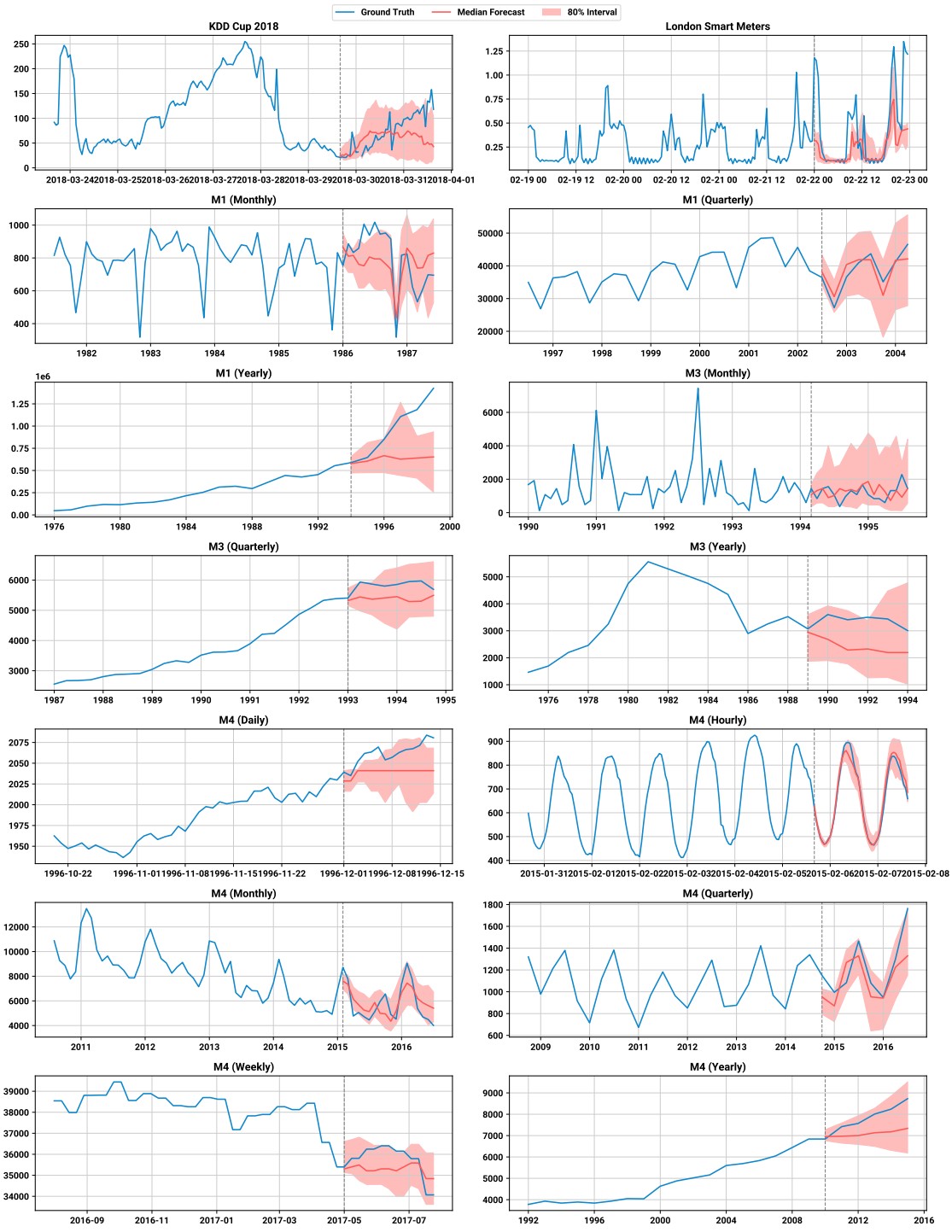

Figure 23: Example of forecasts from CHRONOS-T5 (Base) on the test datasets used in experiments.

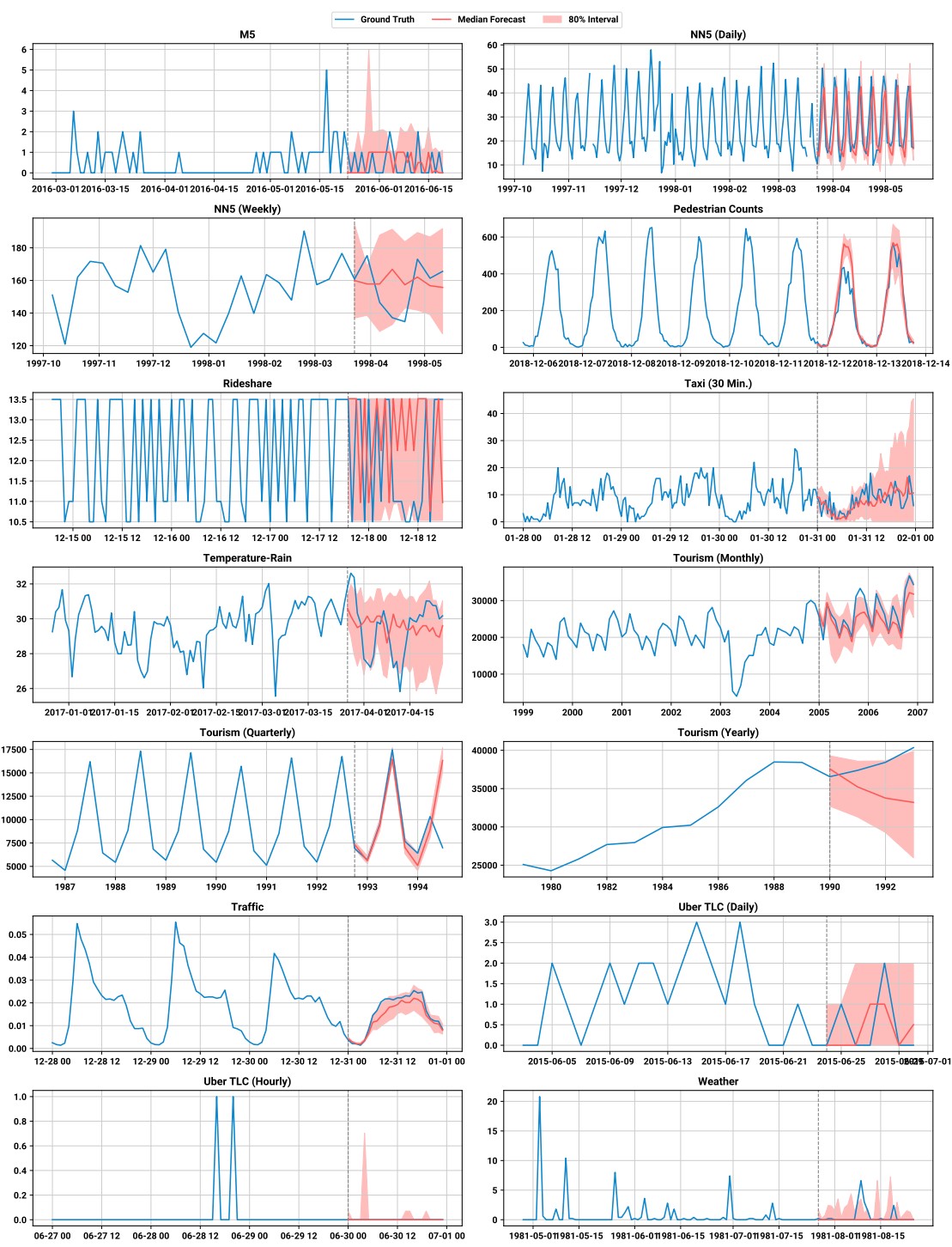

Figure 24: Example of forecasts from CHRONOS-T5 (Base) on the test datasets used in experiments.

