# OpenReview forum: "Chronos: Learning the Language of Time Series"
_TMLR — Accepted by TMLR_

### Review · Reviewer_Tq2a · 2024-06-13

**Summary Of Contributions:**

The paper introduces a foundation model for time series forecasting. They introduce Chronos, an architecture which leverages the existing T5 architecture from the NLP literature. To bridge the gap between time series data and language, they discretize time series data into discrete tokens. They train the model on a collection of diverse time series data, and show that it has strong capabilities for both in-domain and zero-shot forecasting. They also introduce KernelSynth, a synthetic data generation process for time series data, and show that incorporating synthetic data improves overall performance of the model. Finally they perform extensive empirical analysis and ablations to provide greater understanding of the proposed method.

**Audience:**

Yes

**Broader Impact Concerns:**

No concerns.

**Claims And Evidence:**

Yes

**Requested Changes:**

### Changes critical to securing recommendation for acceptance
1. TSMixup has some issues that should be clarified:
    a. Why is mean scaling performed before mixing? Is there any data leakage given that mean scaling is performed on the whole series?
    b. What does it mean for $\alpha=1.5$? Is this a vector of length $k$ with all elements as 1.5?
2. More analysis or at least elaboration about the different scaling methods should be given. Why was mean scaling chosen over min-max, standardization, etc...? It seems that standardization would solve many of the problems related to loss of precision presented in the analysis section, why was mean scaling chosen over standardization?

### Changes that would strengthen the work
1. Please add in related work section to the extensive literature of ordinal regression and regression via classification.
2. Please improve the writing regarding quantization.

**Strengths And Weaknesses:**

### Strengths
1. The paper presents a strong foundation model, and is one of the first papers to concurrently show the possibility and benefits of training of a large corpus of time series data.
2. The paper shows that discretization is an effective approach for modeling time series data. The benefits of such an approach is that it removes the complexities of dealing with real values, and probabilistic forecasting is made simper as the categorical distribution can easily approximate any complex distribution.
3. The paper performs presents extensive details of model training and datasets. It also contains extensive empirical analysis to understand the strengths and weaknesses of the proposed method.

### Weaknesses
1. The method can only handle univariate time series.
2. Prediction length is limited to a maximum of 64. Autoregressive models suffer from error accumulation - it is unclear whether such a method extend to longer horizons.
3. The paper writing leans heavily into the idea of being motivated by LLMs. It ignores extensive related work of ordinal regression and regression via classification (RvC). Per my understanding, the main difference between this work and the aforementioned is that in this paper, inputs are also discretized, whereas in ordinal regression and RvC, it is mainly the outputs that are discretized.
4. The writing related to quantization can be significantly improved. Currently it is quite confusing to understand the exact approach. [c_1,c_B], [l=-15, r=15] and [-15s, 15s] have been used to describe the prediction range. This is quite confusing. All the details of the exact approach used in training should be described in one place, and use a unified notation. The vocabulary size of the pre-trained models should also appear in Section 5.2.

---

> ### Author Response · Authors · 2024-06-25
> **Author Response (1/2)**
>
> Thank you for your review and suggestions on improving the clarity of our work. Please find our responses to your comments below.
>
> **Comment**: “The method can only handle univariate time series.”
> **Response**: The scope of this work is limited to univariate forecasting as we believe *zero-shot* univariate forecasting needs to be convincingly addressed before moving on to more general settings. Univariate forecasting forms a critical chunk of practical time series problems and is a stepping stone to forecasting with exogenous features, multivariate forecasting, and multimodal forecasting.
>
> **Comment**: “Prediction length is limited to a maximum of 64. Autoregressive models suffer from error accumulation - it is unclear whether such a method extend to longer horizons.”
> **Response**: Chronos models’ prediction length was set to a value (64) greater than the largest prediction length in the evaluation benchmark. This is not a limitation of the proposed approach and the models can be trained with longer and/or variable prediction lengths. Error accumulation is a general problem with autoregressive models, not specific to Chronos. Nevertheless, it is worth noting that all popular language models today are autoregressive and are capable of generating thousands of tokens without degradation in the quality of generated text.
>
> In the context of the current pretrained Chronos models, a simple technique enables forecasting beyond the maximum prediction length. Specifically, the median of the prediction is concatenated with the context and passed to the model as ground truth to forecast beyond 64 steps. The following example from the electricity dataset shows that this simple technique generates reasonable predictions for horizons longer than 64 (120 in this example).
>
> Link to image: https://imgur.com/a/chronos-predictions-on-prediction-length-64-AFnUJhR
>
> **Comment**: Related work on ordinal regression and regression via classification.
> **Response**: Thank you for this suggestion. For ordinal regression, we have identified [3-6] as potentially relevant: we will add a discussion on the topic by including relevant references, together with a discussion on regression via classification [8-10] for which we already have reference [7] in the manuscript. If you have suggestions on other specific works in these areas that would improve the discussion, we would be happy to include them.
>
> **Comment**: Discussion on quantization is confusing.
> **Response**: We have removed the redundant $[l,r]$ notation and modified the discussion on quantization. The scaled time series is quantized into bins with uniformly-spaced centers $\\{c_1​,c_2​,\dots,c_B​\\}$. This means that the distance between successive bin centers, $∣c_{i+1}​−c_i​∣$, is fixed and the bin edges fall halfway between the bins centers, $b_i=\frac{(c_{i} + c_{i+1})}{2}$​. For our experiments, we set $c_1​=−15$ and $c_B​=+15$. Note that these are the limits after scaling the time series. The tokens can represent (unscaled) values in the range $[−15s,+15s]$, where s is the scale of the time series. We have clarified this in the **overflow and loss of precision** paragraph.
>
> **Comment**: (a) Why is mean scaling performed before mixing in TSMixup? (b) Is there any data leakage? (c) What does it mean for $\alpha=1.5$?
> **Response**: (a) Normalization is performed to ensure that time series with different (small or large) scales are giving equal importance in the mixing process. In the absence of normalization, time series with large values will completely dominate the process. (b) There is no data leakage as only the training portion of the time series is used for generating the augmentations. (c) We used the [symmetric Dirichlet distribution](https://en.wikipedia.org/wiki/Dirichlet_distribution#Special_cases) which is parameterized by a scalar concentration parameter $\alpha$. This just means that $\alpha_i=\alpha \\: \forall \\: i \in \\{1, \dots,k\\}$. We have clarified this in the main text and the algorithm in the appendix.

---

> ### Author Response · Authors · 2024-06-25
> **Author Response (2/2)**
>
> **Comment**: Why was mean scaling selected?
> **Response**: We did not investigate different normalization techniques in the context of this work. The decision to use mean scaling (as opposed to median, std scaling, etc.) was primarily based on its successful application in prior works [1, 2]. An attractive feature of mean scaling is that it preserves zero values in the time series, which are often semantically meaningful, such as zero sales for a product or zero solar energy generation at night. Nevertheless, we agree with the reviewer that some limitations related to the loss of precision due to small variation (Fig. 16b) can potentially be solved by alternative normalization schemes such as standardization. Note that one doesn’t necessarily need to re-train the models to address these precision issues. Standardization can be used as a preprocessing step to shift and scale the time series before feeding them into Chronos. Here is the result when standardization is applied as a preprocessing step on the example in Fig. 16b:
>
> Link to image: https://imgur.com/a/standardization-as-preprocessing-chronos-x3qoBtt
>
> Intrigued by the reviewer’s suggestion, we conducted a small-scale experiment comparing mean scaling with standardization. We trained a Chronos-T5 (Small) model with mean scaling and another with standardization, while controlling for everything else. The following table shows the aggregated relative scores (lower is better) for both models.
>
>
> | Benchmark   | Metric   |   Chronos-T5 (Small, Mean) |   Chronos-T5 (Small, Std) |
> |:------------|:---------|---------------------------:|--------------------------:|
> | In Domain   | MASE     |                   **0.776143** |                  0.789948 |
> | In Domain   | WQL      |                   **0.636094** |                  0.660940 |
> | Zero Shot   | MASE     |                   0.895351 |                  **0.876285** |
> | Zero Shot   | WQL      |                   **0.706839** |                  0.712488 |
>
>
> Mean scaling outperforms standardization on 3/4 metrics. These results suggest that mean scaling may be a superior choice for general time series forecasting problems on average. A comprehensive analysis of tokenization design choices (normalization, quantization, higher-order tokens, etc.) would constitute interesting future research.
>
> We hope that we have satisfactorily addressed your questions and comments. If you have further questions, we will happy to answer them.
>
> ### References
>
> [1] Salinas, David, et al. "DeepAR: Probabilistic forecasting with autoregressive recurrent networks." *International journal of forecasting* 36.3 (2020): 1181-1191.
> [2] Rabanser, Stephan, et al. "The effectiveness of discretization in forecasting: An empirical study on neural time series models." *arXiv preprint arXiv:2005.10111* (2020).
> [3] Whinship and Mare, “Regression models with ordinal variables.” *American Sociological Review* Vol. 49 (1984): 512-525 https://scholar.harvard.edu/files/cwinship/files/asr_1984.pdf
> [4] Cheng et al. “A neural network approach to ordinal regression.” *2008 IEEE International Joint Conference on Neural Networks* (2008) https://ieeexplore.ieee.org/abstract/document/4633963
> [5] Cao et al. “Rank consistent ordinal regression for neural networks with application to age estimation.” *Pattern Recognition Letters *Vol. 140 (2020): 325-331 https://www.sciencedirect.com/science/article/pii/S016786552030413X
> [6] Hu et al. “Transformer-Based Deep Survival Analysis.” *Proceedings of AAAI Spring Symposium on Survival Prediction - Algorithms, Challenges, and Applications* (2020) https://proceedings.mlr.press/v146/hu21a.html
> [7] Luis Torgo and Joao Gama. “Regression using Classification Algorithms.” *Intelligent Data Analysis* 1(4): 275–292, 1997.
> [8] Van Den Oord, Aäron, Nal Kalchbrenner, and Koray Kavukcuoglu. "Pixel recurrent neural networks." *International conference on machine learning*. PMLR, 2016.
> [9] Stewart, Lawrence, et al. "Regression as classification: Influence of task formulation on neural network features." *International Conference on Artificial Intelligence and Statistics*. PMLR, 2023.
> [10] Farebrother, Jesse, et al. "Stop regressing: Training value functions via classification for scalable deep RL." *arXiv preprint arXiv:2403.03950* (2024).

---

> ### Comment · Reviewer_Tq2a · 2024-09-06
>
> Thank you authors for the detailed response, my queries and concerns have been addressed.

---

> > ### Author Response · Authors · 2024-09-07
> >
> > We are glad to know that. Thank you for the update and for reviewing our paper.

---

### Review · Reviewer_WdST · 2024-06-27

**Summary Of Contributions:**

This paper presents a framework for pretrained probabilistic time series models by directly adapting language model architectures for time series forecasting. The proposed Chronos is pre-trained on top of the T5 family with public and synthetic datasets with quantization techniques to get tokens. Chronos models achieve better performance under in-domain scenarios and on-par zero-shot performance compared with baselines.

**Audience:**

Yes

**Claims And Evidence:**

Yes

**Requested Changes:**

Address comments 1-4 above.

**Strengths And Weaknesses:**

Strengths:

(i) The adaptation of language models for time series modeling via scaling and quantization and using off-the-shelf language models with minimal modifications provides a practical solution to integrate with future advancements in LLMs

(ii) The proposed data augmentation techniques and synthetic data generation are simple yet effective, providing solutions for addressing the limited availability of time series data

(iii) The authors provide comprehensive experimental results across 42 datasets and analysis to support the design choices of Chronos

Weaknesses:

See more details in comments below

(i) More comprehensive comparison with baselines

(ii) More details on finetuning results

(iii) Model scalability

(iv) Data contamination problem

Comment:
1. Although the authors compared Chronos with more than ten baseline methods, they did not include comparisons with strong baselines such as ensembles. These models often require much less inference time, and it remains unclear how their accuracy compares with that of Chronos. A more thorough comparison with ensembling baselines would provide a clearer understanding of Chronos' relative performance.

2. The paper lacks detailed information on fine-tuning techniques and results. The comparison is only limited to Chronos (small), and it would be beneficial to see the performance limits of the model under fine-tuning scenarios.

3. While the authors conducted experiments to assess the impact of the portion of synthetic data, the paper does not address how the model’s accuracy changes with respect to the total amount of training data. It would be valuable to include an analysis of model scalability, illustrating how Chronos performs as the training dataset size varies.

4. The authors used 55 public datasets, splitting them into different components for pretraining and evaluation. However, there is a concern regarding potential data contamination. It is crucial to highlight whether data from the same domain (similar time series) is used in both pretraining and zero-shot scenarios, as this could affect the validity of the zero-shot performance claims.

---

> ### Author Response · Authors · 2024-08-29
> **Author Response (1/2)**
>
> We would like to thank the reviewer for their review and suggestions to improve our work. Please find our responses to your comments below.
>
> **Comment**: More comprehensive comparison with baselines. Although the authors compared Chronos with more than ten baseline methods, they did not include comparisons with strong baselines such as ensembles. These models often require much less inference time, and it remains unclear how their accuracy compares with that of Chronos. A more thorough comparison with ensembling baselines would provide a clearer understanding of Chronos' relative performance.
> **Response**: Model ensembling often yields accuracy improvements in time series forecasting tasks. However, this applies to all models (including Chronos or other pretrained models) and it’s not clear what a “fair” comparison in this sense would be, or where to draw the line (in terms of what ensembles to consider) in the context of the paper. For this reason, we decided to focus our attention on “standalone forecasters”, i.e., existing, popular local (e.g., AutoARIMA) and global (e.g., PatchTST) forecasting models. Note that such an experimental design is not unique to our work and is followed by other recent works on pretrained models as well [3, 4]. In this sense, our benchmark against **17 baselines** is comprehensive and supports our claim that zero-shot performance of Chronos models is competitive with the performance of fully-trained (local and global) models.
>
> Furthermore, any ensemble will take at least as much computing resources as the most compute-intensive model in the ensemble. A popular ensemble from the literature [1] features AutoARIMA among its component models, which can be seen to be slower than Chronos-T5 (Large), on average, on the test datasets we considered (see Figure 17 in the paper). We conducted an experiment with this ensemble and the results are tabulated below. Chronos models perform significantly better overall in terms of the forecast accuracy while being much faster on average.
>
> | Model              |   In Domain (WQL) |   In Domain (MASE) |   Zero Shot (WQL) |    Zero Shot (MASE) |   Avg. Inference Time (ms) |
> |:-------------------|------------------:|-------------------:|------------------:|--------------------:|---------------------------:|
> | Chronos-T5 (Small) |             0.603 |              0.727 |             0.667 |               0.841 |                     11.810 |
> | Chronos-T5 (Large) |             **0.564** |              **0.695** |             **0.645** |               **0.823** |                     96.010 |
> | SCUM Ensemble      |             1.060 |              0.940 |             0.728 |               0.838 |                    326.180 |
>
>
> **Comment**: More details on finetuning results. The paper lacks detailed information on fine-tuning techniques and results. The comparison is only limited to Chronos (small), and it would be beneficial to see the performance limits of the model under fine-tuning scenarios.
> **Response**: We deliberately chose to only run a small experiment with dataset-agnostic hyperparameters for fine-tuning. This experiment serves as a proof of concept to demonstrate that there is room for improving Chronos’ accuracy by fine-tuning the models. The optimal fine-tuning procedure and settings may be very much domain (or even dataset)-dependent. We agree that fine-tuning is an important component of one’s forecasting pipeline when using pretrained models. However, we argue that developing an optimal fine-tuning procedure and exploring its limitations is a standalone research problem what would warrant a dedicated study.
>
> **Comment**: Model scalability. While the authors conducted experiments to assess the impact of the portion of synthetic data, the paper does not address how the model’s accuracy changes with respect to the total amount of training data. It would be valuable to include an analysis of model scalability, illustrating how Chronos performs as the training dataset size varies.
> **Response**: Indeed, this would be a very interesting result to show, akin to the scaling properties (“scaling laws”) that were demonstrated for natural language tasks. However, this would arguably require a significant number of additional experiments and discussion, and to gain access to a significantly larger amount of time series data: the most prominent such analysis for natural language tasks [2] is based on 400 model trainings on up to 500B tokens. We believe that with time series, we are not currently in the data regime to conduct a meaningful analysis, although very recent works have made preliminary strides in this direction [5, 6].

---

> ### Author Response · Authors · 2024-08-29
> **Author Response (2/2)**
>
> **Comment**: Data contamination problem. The authors used 55 public datasets, splitting them into different components for pretraining and evaluation. However, there is a concern regarding potential data contamination. It is crucial to highlight whether data from the same domain (similar time series) is used in both pretraining and zero-shot scenarios, as this could affect the validity of the zero-shot performance claims.
> **Response**: The zero-shot datasets in the context of Chronos are datasets that were not part of the training corpus of Chronos. This definition is same as the one used by other contemporary pretrained models [3, 4].  Our zero-shot datasets are from well-documented public sources and consist of distinct time series, originating from distinct processes compared to the training ones. This ensures that the zero-shot time series are not the same as the time series in our training corpus. It is definitely possible (and hoped) that the patterns in the downstream unseen time series resemble parts of the training time series to a certain degree. In fact, this is a key motivation of any model trained on large scale data. The two data augmentation schemes presented in our paper also improve the pattern diversity of the training corpus to facilitate downstream generalizability such that the model is not “perplexed” when seeing an unseen time series during inference. This does not affect the validity of our results, since the baselines (local models, and task-specific global models) are trained on the training portion of the time series from the datasets in Benchmark II (i.e. data prior to the test portion of the time series used for evaluation), while Chronos was not.
>
> We hope that we have satisfactorily addressed the reviewer’s concerns and questions. If you have further questions, we will be happy to answer them.
>
> ### References
>
> [1] Petropoulos, Svetunkov, “A simple combination of univariate models”, International Journal of Forecasting, Volume 36, Issue 1, January–March 2020, Pages 110-115
> [2] Hoffmann et al., “Training Compute-Optimal Large Language Models”, https://arxiv.org/pdf/2203.15556
> [3] Woo, Gerald, et al. "Unified training of universal time series forecasting transformers." arXiv preprint arXiv:2402.02592 (2024).
> [4] Das, Abhimanyu, et al. "A decoder-only foundation model for time-series forecasting." arXiv preprint arXiv:2310.10688 (2023).
> [5] Edwards, Thomas DP, et al. "Scaling-laws for Large Time-series Models." arXiv preprint arXiv:2405.13867 (2024).
> [6] Shi, Jingzhe, et al. "Scaling Law for Time Series Forecasting." arXiv preprint arXiv:2405.15124 (2024).

---

> ### Author Response · Authors · 2024-09-10
>
> Dear reviewer,
>
> Thank you again for your review and valuable suggestions. We hope that our response has satisfactorily addressed your questions and comments. If you have follow-up questions, we would be happy to answer them.

---

> ### Author Response · Authors · 2024-09-19
>
> Dear Reviewer,
>
> Thank you again for reviewing our paper. With the final recommendation deadline approaching soon (25 Sept), we hope that we have addressed the questions and concerns raised by you. If you have follow-up questions, we would be happy to answer them.

---

> > ### Comment · Reviewer_WdST · 2024-11-04
> >
> > Thanks for addressing my comments.

---

### Review · Reviewer_B6a4 · 2024-08-28

**Summary Of Contributions:**

This paper introduces Chronos, a simple yet effective way of modelling time series by casting it as a language modelling problem. To that end, the paper proposes a way of discretely tokenising real-valued time-series data through scaling and quantisation. Through this step, one can then simply apply any state-of-the-art language modelling architectures to predict the next discrete tokens in the sequence, in the exact same fashion as to how one traines a standard text-based language model that predicts the next word, conditional on the previous ones. The forecasts of the time series predictor can then be converted back into real-valued outputs by simply reversing the tokenisation process (i.e. de-scaling & de-quantisation) in a deterministic fashion.

Nevertheless, one remaining issue with time series forecasting is the relative lack of pre-training data, especially when compared to LLM training where very large amounts of unlabelled natural language text can be easily obtained by crawling the web. To that end, this paper uses two data augmentation techniques: (i) TSMixUp (combining different, randomly-selected time series data through convex combinations) and (ii) KernelSynth (generating synthetic time series data by composing different Kernel functions).

Experiments on a wide range of time series forecasting problems demonstrate that the proposed T5-based Chronos model outperforms various classical and deep learning time series forecasting approaches. Furthermore, the approach yields a decent performance in a **zero-shot** setup, where the pre-trained model is applied to an unseen time series task out of the box without any fine-tuning. This demonstrates the plausibility of the pre-trained model as a **generalist** foundation model, although fine-tuning yields even better performance (as expected). The paper then conducted extensive analyses and ablation studies to disentangle the impact of various hyper-parameters, such as the model size and context length.

**Audience:**

Yes

**Broader Impact Concerns:**

No broader impact concerns from my side.

**Claims And Evidence:**

Yes

**Requested Changes:**

1. **Critical**: More discussion and clarification around the tokenisation strategy, and how it handles time series with very different standard deviations (otherwise the time series with low standard deviations would have their values squeezed into a small number of buckets).

2. **Recommended**: Addressing the questions below, which would further improve the clarity of the paper and resolve any doubt.
    - For Eq. (2), would it help to train the model on all tokens, including the context tokens (in addition to the H tokens that need to be predicted)? This can at least be used for the autoregressive models, where standard LMs also use every token for training. For the T5-style encoder-decoder models, it makes sense to only train on the H tokens, and use the C context tokens with the bidirectional attention / encoder. (as is done currently)
    - How easy would it be to incorporate a simple cost function that weighs different kinds of mistakes? This would penalise mistakes that are "further off", e.g. predicting the value "1" when the correct answer is "1,000".
    - Regarding the KernelSynth (Section 4.2), how big is the Kernel bank?
    - To confirm, are all deep learning approaches listed on the paper trained on the exact same pre-training dataset?
    - Currently, the proposed time series foundation model only operates on time series data. Would adding natural language information help? E.g. one could imagine giving information in natural language about what kind of time series it is, whether there is seasonality aspect, where the data came from, what time period it encompasses, etc. Naturally this would require the model to represent and model both time series data and natural language text, but given the success of LLMs, this could probably work better.
    - How does the method compare with state space models like S4 (Albert Gu et al., 2021)? These kinds of models have shown great promise for time series modelling and modelling long-range dependencies in linear time.

3. **Recommended**: Incorporate the suggestions below.
    - For the results (e.g. Figures 4 - 6), it would be nice to include information on the caption that **lower is better**. This would help readers who are not familiar with the evaluation metric to better interpret the results. This holds for Figures 10 - 11 as well.
    - It seems like not all the models in Figure 4, etc. are trained on the exact same dataset (due to these models being pre-trained off-the-shelf models that are trained on a different pre-training dataset). It would be nice to mark these models as such, for example by putting an asterisk next to the model name when listed in Figure 4, etc. This would let the readers know that these models are not directly comparable.
    - On page 12, "Fine tuning" section, it seems like the fine-tuning is done with an initial learning rate and a linear decay. In my experience, using a "triangular" learning rate, where the learning rate is warmed up from 0 to the peak value (here 0.001), and then decayed linearly after that, would work better.

**Strengths And Weaknesses:**

# Strengths

1. Given the impressive recent success of LLMs and foundations, which primarily operate on discrete text tokens / words, how we can extend this success to the real-valued time series forecasting task remains an important research question, with a high potential for real-world impact. This paper proposes a simple way to do so, by simply converting the time series forecasting problem into discrete tokens. This means that any future state-of-the-art language modelling improvements can also transfer easily to the time series forecasting problem.

2. The paper demonstrates strong empirical results on various time series datasets, compared to both classical and deep learning methods. The zero-shot generalisation property is particularly appealing, demonstrating the pre-trained model's ability to generalise and transfer to new tasks.

3. The experiments are rigorous and thorough. I especially appreciate the extensive and thorough ablation studies, which disentangle the impact of different factors such as model size, context length, etc.

4. The paper is generally very clear and well-written.

# Weaknesses

1. I find some aspects of the proposed approach to be rather counter-intuitive. For example:
    - My primary concern is regarding the tokenisation approach. In my understanding, the approach works by discretising the real values into fixed buckets, after doing mean scaling. However, some time series might have a much higher standard deviation than others. This means that time series with low standard deviations would have their range of values "squeezed" into a small number of bins, which makes predicting accurate values much more tricky (standard scaling, which takes into account both the mean and the standard deviation, should address this concern to some extent). Another issue, which is already pointed out in the paper, is that the model cannot predict values that are outside of the range of the bins, which may happen with a new, unseen time series. The root cause of both of these issues is that the model is using the same tokenisation scheme for all time series, which encourages generality, although it may come at the expense of the ability to adapt to unseen times series with very different scale & standard deviation properties than those seen at training time.
    - Based on Eq. (2), it seems that the model is not trained on the context tokens (i.e. given C + H tokens, where C is the number of context tokens and H is the number of tokens to predict, the model is only trained on tokens 1 to H). Training the model to also predict the context tokens, as we do in language modelling (i.e. all tokens are used for training in language modelling, from the first to the last token), may alleviate the data scarcity issue further.
    - It seems like there is a very clear ordering of values here, so not all mistakes are created equal (e.g. predicting a slightly wrong value should have a much less severe cost than predicting a value that is way off). The paper shows that the model can already learn this ordering to some extent, but intuitively, taking into account this "cost weighting" should improve the performance further. Even simple approaches that penalise the different mistakes differently should be able to take this into account.

2. It seems that the paper does not compare against recent state-space models, which obtained strong results on time series modelling, such as the S4 model (Efficiently Modeling Long Sequences with Structured State Spaces, Albert Gu et al., 2021).

3. There are some suggestions (particularly relating to the presentation, etc.) and questions that can be clarified in order to make the paper stronger. See the "requested changes" section below.

---

> ### Author Response · Authors · 2024-08-29
> **Author Response (1/2)**
>
> Thank you for your comprehensive review and insightful comments. Please find our responses below.
>
> **Comment**: The proposed tokenization scheme may be problematic for time series with small standard deviations.
> **Response**: Indeed, tokenization based on mean scaling and quantization may run into precision issues (i.e. large quantization errors) when dealing with time series with a large scale and small standard deviation. We have already highlighted this limitation of our tokenization scheme in Fig. 16 (b), where we observe quantization artifacts in a shifted sinusoid. Nevertheless, based on our strong empirical results on 42 real-world datasets, we can conclude that such time series are not pervasive in general forecasting tasks.
>
> Our design decision of using mean scaling was motivated by its successful application in existing works [1, 2] and the fact that mean scaling preserves zeros, which are often semantically meaningful (e.g., zero demand in retail and zero solar energy production during the night). We agree with your comment that standardization may address some of these issues. However, as mentioned in our response to a similar comment from Rev. Tq2a, one doesn’t necessarily need to re-train the models to address these precision issues. Based on some heuristic threshold on the standard deviation, standardization can be used as a preprocessing step to shift and scale the time series before feeding them into Chronos. In the case of the example in Fig. 16b, this preprocessing results in an accurate forecast.
>
> Link to image: https://imgur.com/a/standardization-as-preprocessing-chronos-x3qoBtt
>
> Based on Rev. Tq2a’s suggestion, we also conducted a small-scale experiment comparing the effect of using mean scaling vs. standardization for normalization. We observed that mean scaling performs better than standardization on 3/4 metrics when controlling for everything else (see table below). This suggests that the solution may not be as straightforward as using standardization during pretraining. A comprehensive analysis of tokenization design choices (normalization, quantization, higher-order tokens, etc.) would constitute interesting future research.
>
> The following table shows the aggregated relative scores (lower is better) for Chronos-T5 (Small) models trained with mean scaling and standardization.
>
> | Benchmark   | Metric   |   Chronos-T5 (Small, Mean) |   Chronos-T5 (Small, Std) |
> |:------------|:---------|---------------------------:|--------------------------:|
> | In Domain   | MASE     |                   **0.776143** |                  0.789948 |
> | In Domain   | WQL      |                   **0.636094** |                  0.660940 |
> | Zero Shot   | MASE     |                   0.895351 |                  **0.876285** |
> | Zero Shot   | WQL      |                   **0.706839** |                  0.712488 |
>
>
> **Comment**: “...would it help to train the model on all tokens, including the context tokens...”
> **Response**: We already report results on a variant of Chronos based on the decoder-only GPT2 model. This model uses all the tokens (including the context tokens) to compute the training loss. This decoder-only variant performed worse than the variants based on the encoder-decoder (T5) architecture (see Figs. 4 & 5 in the paper). Furthermore, we encountered the following issues when conducting experiments with the decoder-only variant.
>
> * The decoder-only model was very brittle and difficult to optimize. Based on our investigation, this was mainly due to the presence of missing values. Unlike causal language modeling, our training corpus also includes missing values. The standard approach of using an attention mask to handle missing values led to severe optimization instabilities in the decoder-only models. As a result, we had to filter out up to 10% of our training corpus (by removing time series with a large proportion of missing values) to be able to stably train the decoder-only model. On the other hand, the T5 model was robust to missing values and did not suffer from any optimization issues.
> * As shown in Fig. 17, the decoder-only model was considerably slower for inference when compared with the T5-based models. The slower wall-clock time was primarily due to larger GPU memory requirements when generating multiple forecast samples.

---

> > ### Comment · Reviewer_B6a4 · 2024-09-09
> > **Thank You for the Response**
> >
> > Thank you for the thorough authors' response! I can confirm that most of my concerns have been addressed. I particularly appreciate the results on the standard scaling (a similar issue was raised by another reviewer) and how it's not necessarily better than mean scaling, although one can still standardise each time series dataset before running the result through Chronos (although the standard deviation normalisation is dataset-dependent in this case).
> >
> > The response has also addressed my other concerns. I would recommend the authors to include these points in a later revision (including the experimental results with standard scaling, the label smoothing results to take into account a notion of "distance" in the loss function, designing other distance-aware loss functions in future work, the fact that the decoder-only models are trained on all the tokens, etc.), whether in the main paper or the Appendix.
> >
> > One remaining question based on the response: I don't understand why the decoder-only model is slower to run when generating multi-forecast samples? In the LLM space, people generate multiple tokens from the LLM just fine (potentially through speculative sampling, etc.).  Why does the encoder-decoder model use up less memory than the decoder-only model? Is it just due to the model size, or because of something else? In my understanding, the sequence length should be exactly the same for the two models?
> >
> > Also, I would encourage the authors to try state space models as baselines, especially in light of their strong performance and fairly recent publication (which means that it outperforms many older baselines). It's not urgent for this work as the current paper is already strong enough, but if the authors are extending the paper, consider running this baseline as it might change the pattern
> > of results. Though of course, one can also run Chronos with state space models as the foundation model. But given the good performance of state space models in dealing with real-valued sequences already, using them might resolve some of the problems with quantising the real values into discrete buckets.

---

> > > ### Author Response · Authors · 2024-09-09
> > >
> > > Thank you for your response. We are glad to know that our response has addressed your concerns, and we appreciate your suggestions, which have greatly contributed to enhancing the clarity of our manuscript. We will ensure that the discussed points are included in the revision.
> > >
> > > > I don't understand why the decoder-only model is slower to run when generating multi-forecast samples? Why does the encoder-decoder model use up less memory than the decoder-only model?
> > >
> > > Apart from the fact that T5 and GPT2 have different architectures, a key reason for larger memory requirements is how multi-sample inference is implemented in the `transformers` library, which we used for training and evaluating Chronos models. In the case of encoder-decoder models, the context is fed into the encoder and then repeated along a new **sample** dimension. This repeated context is then used for cross-attention in the decoder. In contrast, for the decoder-only models, the input context itself is repeated along a new dimension before being fed into the decoder, effectively scaling the batch size with the number of forecast samples. This leads to larger memory requirements and more computation. It may be possible to write a more memory-efficient implementation for decoder-only Chronos models. However, we did not investigate this further, primarily due to the generally worse performance (when compared with T5 models) and training instabilities we encountered with the decoder-only models.
> > >
> > > Thank you also for your suggestion regarding state-space models. These models indeed present a compelling alternative, also  as a backbone for time series foundation models. We will consider including a comparison with state-space models in future benchmarks and follow-up works.

---

> ### Author Response · Authors · 2024-08-29
> **Author Response (2/2)**
>
> **Comment**: Would a distance-aware cost function improve performance?
> **Response**: We show in Fig. 15 that the model learns the topological information directly from the data. Intuitively, providing hints to the model about the topology of the space should help, especially for tokens that may not be well-represented in the training corpus. During the initial phase of this work, we had conducted a preliminary investigation into a distance-aware version of the cross-entropy loss based on label smoothing. The main idea involves assigning non-zero mass to tokens in the vicinity of the correct token instead of using a one-hot encoding. We observed that the performance of such models was not better than the regular Chronos models trained with one-hot encoding. A sample comparison is tabulated below.
>
> | Benchmark   | Metric   |   Chronos-T5 (Small, One-Hot) |   Chronos-T5 (Small, Smoothing) |
> |:------------|:---------|---------------------------:|--------------------------:|
> | In Domain   | MASE     |                   0.726    |                  0.725    |
> | In Domain   | WQL      |                   0.596    |                  0.594    |
> | Zero Shot   | MASE     |                   0.836    |                  0.834    |
> | Zero Shot   | WQL      |                   **0.661**    |                  0.670    |
>
>
> At the same time, Farebrother et al. (2024) [4] have obtained promising results with a distance-aware regression via classification objective, albeit in the context of reinforcement learning. We believe that the regression via classification objective deserves a closer look, both theoretically and empirically, as part of a dedicated work.
>
> **Comment**: How big is the Kernel bank?
> **Response**: The kernel bank comprises 33 kernels. This includes 21 periodic kernels with different commonly-observed seasonalities, 3 linear kernels, 3 RBF kernels, 3 rational quadratic kernels, 2 white noise kernels and 1 constant kernel. We will update the appendix to include details of the exact kernels used in KernelSynth.
>
> **Comment**: What are the deep learning models trained on?
> **Response**: If you’re referring to task-specific deep learning models, these models are trained on the training portion of each dataset, i.e., one model is trained for each dataset/task. We have described this in Section 5.3 and in Figs. 4 & 5. On the other hand, we used the publicly available checkpoints for the baseline pretrained models. The training corpora of these models have been described in their respective papers.
>
> **Comment**: Would adding natural language information help?
> **Response**: For some domains (e.g., finance), adding exogenous natural language information should improve the prediction. However, developing a multimodal forecasting system is a challenging endeavor, particularly due to the scarcity of such multimodal data in the public domain. We hope that our work will accelerate research on these challenging settings.
>
> **Comment**: How does the method compare with state space models?
> **Response**: We did not compare with state space models in the current work. Our comprehensive benchmark includes 17 baselines, spanning different model types, including popular local model (e.g., AutoARIMA) and state-of-the-art global models (e.g., PatchTST, N-BEATS). We believe that inclusion of other baselines will not change the overall conclusion of our paper: Chronos models outperform baselines on in-domain datasets and perform on par with trained task-specific models on unseen (zero-shot) datasets.
>
> **Comment**: Include “lower is better” and mark other pretrained models.
> **Response**: Thank you for your suggestions. We will incorporate them in the manuscript.
>
> **Comment**: Using a "triangular" learning rate would work better for fine-tuning.
> **Response**: We thank the reviewer for this suggestion. The goal of our fine-tuning experiment is to demonstrate that fine-tuning leads to improvements over the zero-shot results. We did not conduct hyperparameter exploration in this context. Indeed, better hyperparameter selection and validation may lead to significant improvements.
>
> We hope that we have satisfactorily answered your questions. If you have any follow-up questions and comments, we would be happy to respond to them.
>
> ### References
>
> [1] Salinas, David, et al. "DeepAR: Probabilistic forecasting with autoregressive recurrent networks." International journal of forecasting 36.3 (2020): 1181-1191.
> [2] Rabanser, Stephan, et al. "The effectiveness of discretization in forecasting: An empirical study on neural time series models." arXiv preprint arXiv:2005.10111 (2020).
> [3] Farebrother, Jesse, et al. "Stop regressing: Training value functions via classification for scalable deep RL." arXiv preprint arXiv:2403.03950 (2024).

---

### Author Response · Authors · 2024-09-05
**Acknowledgement of our responses**

Dear reviewers,

We would like to sincerely thank you for reviewing our paper and providing valuable suggestions for improving the manuscript. We hope you have had the opportunity to review our responses over the past week and trust that we have adequately addressed your concerns. If you find our responses satisfactory, we would greatly appreciate it if you could acknowledge them with a comment. Should you have any further questions or feedback, we would be happy to address them. In particular, we welcome any additional suggestions you may have for enhancing the quality and clarity of the manuscript, especially those you consider critical. Thank you again for your time.

Regards,
The Authors

---

### Decision · Action_Editor_PJUz · 2024-10-30

**Recommendation:** Accept with minor revision

**Comment:**

Really interesting work. This paper is almost ready to be accepted. I have a few minor issues to address.

You say results are averaged over 3 runs. I am trying to understand what that means for Figure 4 and 5.
1. What is the source of stochasticity here?
2. What are the 3 runs averaging over, and how variable are outcomes?
3. Are the differences between algorithms significant?

Minor comment: another recent work found shifting from a regression loss to classification when using Transformers in RL allowed for significant improvements (https://arxiv.org/abs/2403.03950). This could be another useful citation, given the comment from a reviewer about regression as classification.

**Audience:**

There is a clear TMLR audience.

**Claims And Evidence:**

Likely, but one component of the results needs to be clarified

---

> ### Author Response · Authors · 2024-10-31
> **Author Response (1/3)**
>
> Thank you for driving the review process and making the final decision.
>
> In the following, we respond to the questions you have raised above and describe the changes we have made to the manuscript in the camera ready revision that we have just uploaded.
>
> > What is the source of stochasticity here? ... how variable are outcomes?
>
> For Chronos and task-specific deep learning models, the results have been averaged over three training and evaluation runs. Therefore, the primary source of stochasticity is the random initialization of the models. We report the average weighted quantile loss (WQL) and mean absolute scaled error (MASE) over the three runs in Tables 7 - 10 (appendix). These (averaged) scores are then used to compute the final numbers for Figs. 4 and 5 which are geometric means of the relative WQL (and MASE) across the datasets. From prior experience, we know that deep learning-based time series models are sometimes susceptible to training variance. Therefore, to report a robust estimate of Chronos' performance, we trained and evaluated each model (including ablations) 3 times.
>
> Nevertheless, we observed that the performance of Chronos models shows little variance over different training runs, as shown in the following table (see part 2 of the response) which reports the mean and standard deviation of the weighted quantile loss (WQL) on Benchmark II for Chronos and DeepAR. Interestingly, the task-specific DeepAR model exhibits large variance on some datasets (e.g., Australian Electricity and CIF 2016).

---

> ### Author Response · Authors · 2024-10-31
> **Author Response (2/3)**
>
> | Dataset   | Model   |   mean (WQL) |   std (WQL) |
> |:-----------------------|:----------------------|-------:|------:|
> | Dominick               | Chronos-T5 (Large)   |  0.332 | 0.002 |
> | Dominick               | DeepAR                |  0.364 | 0.002 |
> | ERCOT Load             | Chronos-T5 (Large)   |  0.019 | 0.002 |
> | ERCOT Load             | DeepAR                |  0.032 | 0.010 |
> | ETT (15 Min.)          | Chronos-T5 (Large)   |  0.068 | 0.002 |
> | ETT (15 Min.)          | DeepAR                |  0.069 | 0.007 |
> | ETT (Hourly)           | Chronos-T5 (Large)   |  0.073 | 0.002 |
> | ETT (Hourly)           | DeepAR                |  0.081 | 0.005 |
> | Exchange Rate          | Chronos-T5 (Large)   |  0.013 | 0.001 |
> | Exchange Rate          | DeepAR                |  0.009 | 0.001 |
> | M4 (Quarterly)         | Chronos-T5 (Large)   |  0.082 | 0.000 |
> | M4 (Quarterly)         | DeepAR                |  0.080 | 0.001 |
> | M4 (Yearly)            | Chronos-T5 (Large)   |  0.134 | 0.001 |
> | M4 (Yearly)            | DeepAR                |  0.111 | 0.000 |
> | M5                     | Chronos-T5 (Large)   |  0.587 | 0.002 |
> | M5                     | DeepAR                |  0.657 | 0.009 |
> | Australian Electricity | Chronos-T5 (Large)   |  0.067 | 0.006 |
> | Australian Electricity | DeepAR                |  0.087 | **0.042** |
> | Car Parts              | Chronos-T5 (Large)   |  1.060 | 0.010 |
> | Car Parts              | DeepAR                |  0.967 | 0.003 |
> | CIF 2016               | Chronos-T5 (Large)   |  0.014 | 0.001 |
> | CIF 2016               | DeepAR                |  0.136 | **0.039** |
> | Covid Deaths           | Chronos-T5 (Large)   |  0.045 | 0.011 |
> | Covid Deaths           | DeepAR                |  0.108 | **0.029** |
> | FRED-MD                | Chronos-T5 (Large)   |  0.020 | 0.002 |
> | FRED-MD                | DeepAR                |  0.043 | 0.013 |
> | Hospital               | Chronos-T5 (Large)   |  0.056 | 0.001 |
> | Hospital               | DeepAR                |  0.056 | 0.001 |
> | M1 (Monthly)           | Chronos-T5 (Large)   |  0.130 | 0.002 |
> | M1 (Monthly)           | DeepAR                |  0.150 | 0.016 |
> | M1 (Quarterly)         | Chronos-T5 (Large)   |  0.107 | 0.006 |
> | M1 (Quarterly)         | DeepAR                |  0.089 | 0.006 |
> | M1 (Yearly)            | Chronos-T5 (Large)   |  0.183 | 0.018 |
> | M1 (Yearly)            | DeepAR                |  0.139 | 0.015 |
> | M3 (Monthly)           | Chronos-T5 (Large)   |  0.096 | 0.001 |
> | M3 (Monthly)           | DeepAR                |  0.099 | 0.001 |
> | M3 (Quarterly)         | Chronos-T5 (Large)   |  0.074 | 0.001 |
> | M3 (Quarterly)         | DeepAR                |  0.073 | 0.002 |
> | M3 (Yearly)            | Chronos-T5 (Large)   |  0.151 | 0.003 |
> | M3 (Yearly)            | DeepAR                |  0.122 | 0.002 |
> | NN5 (Weekly)           | Chronos-T5 (Large)   |  0.091 | 0.001 |
> | NN5 (Weekly)           | DeepAR                |  0.087 | 0.007 |
> | Tourism (Monthly)      | Chronos-T5 (Large)   |  0.100 | 0.006 |
> | Tourism (Monthly)      | DeepAR                |  0.092 | 0.000 |
> | Tourism (Quarterly)    | Chronos-T5 (Large)   |  0.061 | 0.001 |
> | Tourism (Quarterly)    | DeepAR                |  0.072 | 0.003 |
> | Tourism (Yearly)       | Chronos-T5 (Large)   |  0.183 | 0.008 |
> | Tourism (Yearly)       | DeepAR                |  0.127 | 0.006 |
> | Traffic                | Chronos-T5 (Large)   |  0.256 | 0.003 |
> | Traffic                | DeepAR                |  0.233 | 0.002 |
> | Weather                | Chronos-T5 (Large)   |  0.139 | 0.000 |
> | Weather                | DeepAR                |  0.147 | 0.001 |
> | NN5 (Daily)            | Chronos-T5 (Large)   |  0.156 | 0.001 |
> | NN5 (Daily)            | DeepAR                |  0.155 | 0.002 |
>
> > Are the differences between algorithms significant?
>
> Yes! Ours is one of the most comprehensive evaluations in time series literature with comparisons on 42 datasets. Beyond the aggregated relative scores in the main text, the raw WQL and MASE scores obtained by each model for every dataset have also been reported in Tables 7 - 10 in the appendix for completeness.

---

> ### Author Response · Authors · 2024-10-31
> **Author Response (3/3)**
>
> ### Manuscript Changes
>
> We have added a citation to Farebrother et al. (2024). Including this, we have made the following main changes in the camera ready revision:
>
> 1. We found an off-by-one error in the decoded bin indices for Chronos models which had led to artificially worse results for Chronos models in the previous version. Upon fixing this issue, the results for Chronos models improved significantly. Note that this issue only affected inference and the updated results still refer to the models we had pretrained previously. Further details on this issue can be found in the relevant [Github pull request](https://github.com/amazon-science/chronos-forecasting/pull/73).
> 2. The SCUM ensemble was added as one of the baselines.
> 3. Clarified and polished the text in multiple places. We are thankful to the reviewers for their suggestions. Key changes include:
>     - Added brief reasoning on our use of mean scaling in Section 3.1.
>     - Clarified the notation and discussion on quantization in Section 3.1.
>     - Added a brief discussion on ordinal regression in Section 3.2.
>     - Added a brief discussion on how topological information could potentially be incorporated into the objective function in Section 5.7.
>     - Added details on the kernel bank, $\mathcal{K}$, used in KernelSynth in Table 2.